# DELTA: Diverse Client Sampling for Fasting Federated Learning

**Lin Wang**[1,2]**, Yongxin Guo**[1,2]**, Tao Lin**[4,5]**, Xiaoying Tang**[1,2,3*]
[1]School of Science and Engineering, The Chinese University of Hong Kong (Shenzhen)
[2]The Shenzhen Institute of Artificial Intelligence and Robotics for Society
[3]The Guangdong Provincial Key Laboratory of Future Networks of Intelligence
[4]Research Center for Industries of the Future, Westlake University
[5]School of Engineering, Westlake University.

## Abstract

Partial client participation has been widely adopted in Federated Learning (FL) to reduce the communication burden efficiently. However, an inadequate client sampling scheme can lead to the selection of unrepresentative subsets, resulting in significant variance in model updates and slowed convergence. Existing sampling methods are either biased or can be further optimized for faster convergence. In this paper, we present DELTA, an unbiased sampling scheme designed to alleviate these issues. DELTA characterizes the effects of client diversity and local variance, and samples representative clients with valuable information for global model updates. In addition, DELTA is a proven optimal unbiased sampling scheme that minimizes variance caused by partial client participation and outperforms other unbiased sampling schemes in terms of convergence. Furthermore, to address full-client gradient dependence, we provide a practical version of DELTA depending on the available clients' information, and also analyze its convergence. Our results are validated through experiments on both synthetic and real-world datasets.

## 1 Introduction

Federated Learning (FL) is a distributed learning paradigm that allows a group of clients to collaborate with a central server to train a model. Edge clients can perform local updates without sharing their data, which helps to protect their privacy. However, communication can be a bottleneck in FL, as edge devices often have limited bandwidth and connection availability [58]. To reduce the communication burden, only a subset of clients are typically selected for training. However, an improper client sampling strategy, such as uniform client sampling used in FedAvg [38], can worsen the effects of data heterogeneity in FL. This is because the randomly selected unrepresentative subsets can increase the variance introduced by client sampling and slow down convergence.

Existing sampling strategies can be broadly classified into two categories: biased and unbiased. Unbiased sampling is important because it can preserve the optimization objective. However, only a few unbiased sampling strategies have been proposed in FL, such as multinomial distribution (MD) sampling and cluster sampling. Specifically, cluster sampling can include clustering based on sample size and clustering based on similarity. Unfortunately, these sampling methods often suffer from slow convergence, large variance, and computation overhead issues [2, 13].

To accelerate the convergence of FL with partial client participation, Importance Sampling (IS), an unbiased sampling strategy, has been proposed in recent literature [5, 49]. IS selects clients with a large gradient norm, as shown in Figure 1. Another sampling method shown in Figure 1 is

---

*Corresponding author.

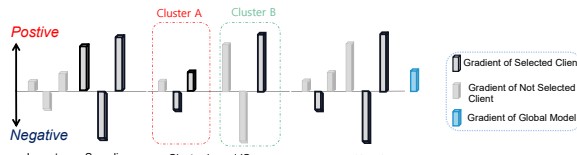

Figure 1: **Client selection illustration of different methods.** IS (left) selects high-gradient clients but faces redundant sampling issues. Cluster-based IS (mid) addresses redundancy, but using small gradients for updating continuously can slow down convergence. In contrast, DELTA (right) selects diverse clients with significant gradients without clustering operations.

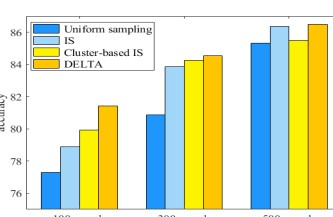

Figure 2: **Comparison of the convergence performance for different sampling methods.** In this example, we use a logistic regression model on non-iid MNIST data and sample 10 out of 200 clients. We run 500 communication rounds and report the average of the best 10 accuracies at 100, 300, and 500 rounds. This shows the accuracy performance from the initial training state to convergence.

cluster-based IS, which first clusters clients according to the gradient norm and then uses IS to select clients with a large gradient norm within each cluster.

Though IS and cluster-based IS have their advantages, **1) IS could be inefficient because it can result in the transfer of excessive similar updates from the clients to the server**. This problem has been pointed out in recent works [52, 63], and efforts are being made to address it. One approach is to use cluster-based IS, which groups similar clients together. This can help, but **2) cluster-based IS has its drawbacks in terms of convergence speed and clustering effect.** Figure 2 illustrates that both of these sampling methods can perform poorly at times. Specifically, compared with cluster-based IS, IS cannot fully utilize the diversity of gradients, leading to redundant sampling and a lack of substantial improvement in accuracy [52, 2]. While the inclusion of clients from small gradient groups in cluster-based IS leads to slow convergence as it approaches convergence, as shown by experimental results in Figure 6 and 7 in Appendix B.2. Furthermore, the clustering algorithm's performance tends to vary when applied to different client sets with varying parameter configurations, such as different numbers of clusters, as observed in prior works [52, 51, 56].

To address the limitations of IS and cluster-based IS, namely excessive similar updates and poor convergence performance, we propose a novel sampling method for Federated Learning termed **D**iv**E**rse c**L**ien**T** s**A**mpling (DELTA). Compared to IS and cluster-based IS methods, DELTA tends to select clients with diverse gradients, as shown in Figure 1. This allows DELTA to utilize the advantages of a large gradient norm for convergence acceleration while also overcoming the issue of gradient similarity.

Additionally, we propose practical algorithms for DELTA and IS that rely on accessible information from partial clients, addressing the limitations of existing analysis based on full client gradients [35, 5]. We also provide convergence rates for these algorithms. We replace uniform client sampling with DELTA in FedAvg, referred to as **FedDELTA**, and replace uniform client sampling with IS in FedAvg, referred to as **FedIS**. Their practical versions are denoted as **FedPracDELTA** and **FedPracIS**.

**Toy Example and Motivation.** We present a toy example to illustrate our motivation, where each client has a regression model. The detailed settings of each model and the calculation of each sampling algorithm's gradient are provided in Appendix B.1. Figure 3 shows that IS deviates from the ideal global model when aggregating gradients from clients with large norms. This motivates us to consider the correlation between local and global gradients in addition to gradient norms when sampling clients. *Compared to IS, DELTA selects clients with large gradient diversities, which exploits the clients' information of both gradient norms and directions, resulting in a closer alignment to the ideal global model.*

**Our contributions.** In this paper, we propose an efficient unbiased sampling scheme in the sense that (i) It effectively addresses the issue of excessive similar gradients without the need for additional clustering, while taking advantage of the accelerated convergence of gradient-norm-based IS and (ii) it is provable better than uniform sampling or gradient norm-based sampling. The sampling scheme is versatile and can be easily integrated with other optimization techniques, such as momentum, to improve convergence further.

As our key contributions,

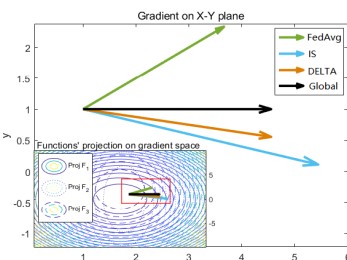

Figure 3: **Model update comparison: The closer to the ideal global update (black arrow), the better the sampling algorithm is.** The small window shows the projection of 3 clients' functions $F_1, F_2, F_3$ in the X-Y plane, where $\nabla F_1 = (2,2), \nabla F_2 = (4,1), \nabla F_3 = (6,-3)$ at $(1,1)$. The enlarged image shows the aggregated gradients of FedAvg, IS, DELTA and ideal global gradient. Each algorithm samples two out of three clients: FedIS tends to select Client 2 and 3 with largeset gradient norms, DELTA tends to select Client 1 and 3 with the largest gradient diversity and FedAvg is more likely to select Client 1 and 2 compared to IS and DELTA. The complete gradient illustration with clients' gradient is shown in Figure 5 in Appendix.

- We present DELTA, an unbiased FL sampling scheme based on gradient diversity and local variance. Our refined analysis shows that FedDELTA surpasses the state-of-the-art FedAvg in convergence rate by eliminating the $\mathcal{O}(1/T^{2/3})$ term and a $\sigma_G^2$-related term of $\mathcal{O}(1/T^{1/2})$.

- We present a novel theoretical analysis of nonconvex FedIS, which yields a superior convergence rate compared to existing works while relying on a more lenient assumption. Moreover, our analysis eliminates the $\mathcal{O}(1/T^{2/3})$ term of the convergence rate, in contrast to FedAvg.

- We present a practical algorithm for DELTA in partial participation settings, utilizing available information to mitigate the reliance on full gradients. We prove that the convergence rates of these practical algorithms can attain the same order as the theoretical optimal sampling probabilities for DELTA and IS.

## 2 Related Work

Client sampling in federated learning (FL) can be categorized into unbiased and biased methods [14]. Unbiased methods, including multinomial sampling and importance sampling [30, 5, 49], ensure that the expected client aggregation is equivalent to the deterministic global aggregation when all clients participate. Unlike unbiased sampling, which has received comparatively little attention, biased sampling has been extensively examined in the context of federated learning, such as selecting clients with higher loss [7] or larger updates [48]. Recently, cluster-based client selection, which involves grouping clients into clusters and sampling from these clusters, has been proposed to sample diverse clients and reduce variance [41, 12, 52]. Nevertheless,the clustering will require extra communication and computational resources. The proposed DELTA algorithm can be seen as a muted version of a diverse client clustering algorithm without clustering operation.

While recent works [57, 28] have achieved comparable convergence rates to ours using variance reduction techniques, it is worth noting that these techniques are orthogonal to ours and can be easily integrated with our approach. Although [60] achieved the same convergence rate as ours, but their method requires dependent sampling and mixing participation conditions, which can lead to security problems and exceed the communication capacity of the server. In contrast, our method avoids these issues by not relying on such conditions.

A more comprehensive discussion of the related work can be found in Appendix A.

## 3 Theoretical Analysis and An Improved FL Sampling Strategy

This section presents FL preliminaries and analyzes sampling algorithms, including the convergence rate of nonconvex FedIS in Section 3.2, improved convergence analysis for FL sampling in Section 3.3, and proposal and convergence rate of the DELTA sampling algorithm in Section 3.4.

In FL, the objective of the global model is a sum-structured optimization problem:

$$f^* = \min_{x \in \mathbb{R}^d} \left[ f(x) \coloneqq \sum_{i=1}^m w_i F_i(x) \right] , \tag{1}$$

where $F_i(x) = \mathbb{E}_{\xi_i \sim D_i} [F_i(x, \xi_i)]$ represents the local objective function of client $i$ over data distribution $D_i$, and $\xi_i$ means the sampled data of client $i$. $m$ is the total number of clients and $w_i$ represents the weight of client $i$. With partial client participation, FedAvg randomly selects $|S_t| = n$ clients ($n \leq m$) to communicate and update model. Then the loss function of actual participating users in each round can be expressed as:

$$f_{S_t}(x_t) = \tfrac{1}{n} \sum_{i \in S_t} F_i(x_t) . \tag{2}$$

Table 1: **Comparison of convergence rate for different sampling algorithms:** Number of communication rounds required to reach $\epsilon$ or $\epsilon + \varphi$ ($\epsilon$ for unbiased sampling and $\epsilon + \varphi$ for biased sampling, where $\varphi$ is a non-convergent constant term) accuracy for FL. $\sigma_L$ is local variance bound, and $G$ bound is $E\|\nabla F_i(x) - \nabla f(x)\|^2 \leq G^2$. $\Gamma$ is the distance of global optimum and the average of local optimum (Heterogeneity bound), $\mu$ corresponds to $\mu$ strongly convex. and $\zeta_G$ is the gradient diversity.

| Algorithm | Convexity | Partial Worker | Unbiasedness | Convergence rate | Assumption |
|---|---|---|---|---|---|
| SGD | S/N | ✓ | ✓ | $\frac{\sigma_L^2}{\mu m K \epsilon} + (\frac{1}{\mu}) / \frac{\sigma_L^2}{m K \epsilon^2} + \frac{1}{\epsilon}$ | $\sigma_L$ bound |
| **FedDELTA** | N | ✓ | ✓ | $\frac{\sigma_L^2}{nK\epsilon^2} + \frac{\dot{M}^2}{K\epsilon}$ | Assumption 3 |
| **FedPracDELTA** | N | ✓ | ✓ | $\frac{\tilde{U}^2 \sigma_L^2}{nK\epsilon^2} + \frac{\tilde{U}^2 \dot{M}^2}{K\epsilon}$ | Assumption 3 and Assumption 4 |
| **FedIS** (ours) | N | ✓ | ✓ | $\frac{\sigma_L^2 + K\sigma_G^2}{nK\epsilon^2} + \frac{M^2}{K\epsilon}$ | Assumption 3 |
| FedIS (others) [5] | N | ✓ | ✓ | $\frac{\dot{M}^2}{nK\epsilon^2} + \frac{A^2+1}{\epsilon} + \frac{\sigma_G}{\epsilon^{3/2}}$ | Assumption 3 and $\rho$ Assumption |
| FedIS (others) [36] | S | ✓ | ✓ | $\frac{\sigma_L^2 + 4nKG^2 + 6n\Gamma}{\mu^2 nK\epsilon} + \frac{K^2 G^2}{\epsilon} + \frac{\|w_0 - w^*\|^2}{\mu K\epsilon}$ | $G$ bound |
| **FedPracIS** (ours) | N | ✓ | ✓ | $\frac{\sigma_L^2 + KU^2\sigma_G^2}{nK\epsilon^2} + \frac{M^2}{K\epsilon}$ | Assumption 3 and Assumption 4 |
| FedAvg [65] | N | ✓ | ✓ | $\frac{\sigma_L^2}{nK\epsilon^2} + \frac{4K\sigma_G^2}{nK\epsilon^2} + \frac{\tilde{M}^2}{K\epsilon} + \frac{K^{1/3}\tilde{M}^2}{n^{1/3}\epsilon^{2/3}}$ | $G$ bound |
| FedAvg [21] | N | ✓ | ✓ | $\frac{\dot{M}^2}{nK\epsilon^2} + \frac{A^2+1}{\epsilon} + \frac{\sigma_G}{\epsilon^{3/2}}$ | Assumption 3 |
| DivFL [2] | S | ✓ | ✗ | $\frac{1}{\epsilon} + \frac{1}{\varphi}$ | Heterogeneity Gap |
| Power-of-Choice [7] | S | ✓ | ✗ | $\frac{\sigma_L^2 + \tilde{G}^2}{\epsilon + \varphi} + \frac{\Gamma}{\mu}$ | Heterogeneity Gap |
| FedAvg [65] | N | ✗ | ✓ | $\frac{\sigma_L^2}{mK\epsilon^2} + \frac{\sigma_L^2/(4K) + \sigma_G^2}{\epsilon}$ | $\sigma_G$ bound |
| Arbitrary Sampling[60] | N | Mix | ✓ | $\frac{\zeta_G^2 + (1+\sigma_L^2)n\rho}{nK\epsilon^2} + \frac{\dot{M}^2}{K\epsilon}$ | Assumption 3 |

$M^2 = \sigma_L^2 + 4K\sigma_G^2$, $\hat{M}^2 = \sigma_L^2 + K(1 - n/m)\sigma_G^2$, $\bar{M}^2 = \sigma_L^2 + 6K\sigma_G^2$, $\dot{M}^2 = \sigma_L^2 + 4K\zeta_G^2$, $\acute{M}^2 = K\zeta_G^2 + K\sigma_L^2$.
Convexity: S and N are abbreviations for strong convex and nonconvex, respectively.    $\rho$ assumption: Bound of the similarity among local gradients.
Mix participation: the number of participating clients is random, from none to full participation.

For ease of theoretical analysis, we make the following commonly used assumptions:

## 3.1 Assumptions

**Assumption 1** (L-Smooth). *There exists a constant $L > 0$, such that $\|\nabla F_i(x) - \nabla F_i(y)\| \leq L\|x - y\|, \forall x, y \in \mathbb{R}^d$, and $i = 1, 2, \ldots, m$.*

**Assumption 2** (Unbiased Local Gradient Estimator and Local Variance). *Let $\xi_t^i$ be a random local data sample in the round $t$ at client $i$: $\mathbb{E}\left[\nabla F_i(x_t, \xi_t^i)\right] = \nabla F_i(x_t), \forall i \in [m]$. The function $F_i(x_t, \xi_t^i)$ has a bounded local variance of $\sigma_{L,i} > 0$, satisfying $\mathbb{E}\left[\left\|\nabla F_i(x_t, \xi_t^i) - \nabla F_i(x_t)\right\|^2\right] = \sigma_{L,i}^2 \leq \sigma_L^2$.*

**Assumption 3** (Bound Dissimilarity). *There exists constants $\sigma_G \geq 0$ and $A \geq 0$ such that $\mathbb{E}\|\nabla F_i(x)\|^2 \leq (A^2 + 1)\|\nabla f(x)\|^2 + \sigma_G^2$. When all local loss functions are identical, $A^2 = 0$ and $\sigma_G^2 = 0$.*

The above assumptions are commonly used in both non-convex optimization and FL literature, see e.g. [21, 27, 60].

We notice that Assumption 3 can be further relaxed by Assumption 2 of [24]. We also provide Proposition C.4 in Appendix C to show all our convergence analysis, including Theorem 3.1,3.5 and Corollary 4.1,4.2 can be easily extended to the relaxed assumption while keeping the order of convergence rate unchanged.

## 3.2 Convergence Analysis of FedIS

As discussed in the introduction, IS faces an excessive gradient similarity problem, necessitating the development of a novel diversity sampling method. Prior to delving into the specifics of our new sampling strategy, we first present the convergence rate of FL under standard IS analysis in this section; this analysis itself is not well explored, particularly in the nonconvex setting. The complete FedIS algorithm is provided in Algorithm 2 of Appendix D, which differs from DELTA only in sampling probability (line 2) by using $p_i \propto \|\sum_{k=0}^{K-1} g_{t,k}^i\|$.

**Theorem 3.1** (Convergence rate of FedIS). *Let constant local and global learning rates $\eta_L$ and $\eta$ be chosen as such that $\eta_L < \min(1/(8LK), C)$, where $C$ is obtained from the condition that $\frac{1}{2} - 10L^2K^2(A^2+1)\eta_L^2 - \frac{L^2\eta K(A^2+1)}{2n}\eta_L > 0$ ,and $\eta \leq 1/(\eta_L L)$. In particular, suppose $\eta_L = \mathcal{O}\left(\frac{1}{\sqrt{T}KL}\right)$ and $\eta = \mathcal{O}\left(\sqrt{Kn}\right)$, under Assumptions 1-3, the expected gradient norm of FedIS algorithm 2 will be bounded as follows:*

$$\min_{t\in[T]} \mathbb{E}\|\nabla f(x_t)\|^2 \leq \mathcal{O}\left(\frac{f^0 - f^*}{\sqrt{nKT}}\right) + \underbrace{\mathcal{O}\left(\frac{\sigma_L^2 + K\sigma_G^2}{\sqrt{nKT}}\right) + \mathcal{O}\left(\frac{M^2}{T}\right)}_{order\ of\ \Phi}. \tag{3}$$

*where $T$ is the total communication round, $K$ is the total local epoch times, $f^0 = f(x_0)$, $f^* = f(x_*)$, $M = \sigma_L^2 + 4K\sigma_G^2$ and the expectation is over the local dataset samples among clients.*

The FedIS sampling probability $p_i^t$ is determined by minimizing the variance of convergence with respect to $p_i^t$. The variance term $\Phi$ is:

$$\Phi = \frac{5\eta_L^2 K L^2}{2} M^2 + \frac{\eta \eta_L L}{2m} \sigma_L^2 + \frac{L\eta\eta_L}{2nK} \text{Var}(\frac{1}{mp_i^t}\hat{g}_i^t),$$
(4)

where $\text{Var}(1/(mp_i^t)\hat{g}_i^t)$ is called *update variance*. By optimizing the update variance, we get the sampling probability FedIS:

$$p_i^t = \frac{\|\hat{g}_i^t\|}{\sum_{j=1}^m \|\hat{g}_j^t\|},$$
(5)

where $\hat{g}_i^t = \sum_{k=0}^{K-1} \nabla F_i(x_{k,t}^i, \xi_{k,t}^i)$ is the sum of the gradient updates of multiple local updates. The proof details of Theorem 3.1 and derivation of sampling probability FedIS are detailed in Appendix D and Appendix F.1.

**Remark 3.2** (Explanation for the convergence rate). *It is worth mentioning that although a few works provide the convergence upper bound of FL with gradient-based sampling, several limitations exist in these analyses and results:*
*1) [49, 35] analyzed FL with IS using a strongly convex condition, whereas we extended the analysis to the non-convex problem.*
*2) Our analysis results, compared to the very recent non-convex analysis of FedIS [5] and FedAvg, remove the term $\mathcal{O}(T^{-\frac{2}{3}})$, although all these works choose a learning rate of $\mathcal{O}(T^{-\frac{1}{2}})$. Thus, our result achieves a tighter convergence rate when we use $\mathcal{O}(1/T + 1/T^{2/3})$ (provided by [43]) as our lower bound of convergence (see Table 1).*
*The comparison results in Table 1 reveal that even when $\sigma_G$ is large and becomes a dependency term for convergence rate, FedIS (ours) is still better than FedAvg and FedIS (others) since our result reduces the coefficient of $\sigma_G$ in the dominant term $\mathcal{O}(T^{-\frac{1}{2}})$.*

**Remark 3.3** (Novelty of our FedIS analysis). *Despite the existence of existing convergence analysis of partial participant FL [65, 47], including FedIS that builds on this analysis [35, 4], none of them take full advantage of the nature of unbiased sampling, and thus yield an imprecise upper bound on convergence. To tighten the FedIS upper bound, we first derive a tighter convergence upper bound for unbiased sampling FL. By adopting uniform sampling for unbiased probability, we achieve a tighter FedAvg convergence rate. Leveraging this derived bound, we optimize convergence variance using IS.*

---

**Algorithm 1** **FedDELTA** and **FedPracDELTA**:
Federated learning with unbiased diverse sampling

**Require:** initial weights $x_0$, global learning rate $\eta$, local learning rate $\eta_l$, number of local epoch $K$, number of training rounds $T$
**Ensure:** trained weights $x_T$
1: **for** round $t = 1, \dots, T$ **do**
2:    **Sampling** clients using **DELTA** (13)
3:    **Sampling** clients using **Practical DELTA** (16)
4:    **for** each worker $i \in S_t$, in parallel **do**
5:      $x_{t,0}^i = x_t$
6:      **for** $k = 0, \cdots, K-1$ **do**
7:        compute $g_{t,k}^i = \nabla F_i(x_{t,k}^i, \xi_{t,k}^i)$
8:        Local update: $x_{t,k+1}^i = x_{t,k}^i - \eta_L g_{t,k}^i$
9:      Let $\Delta_t^i = x_{t,K}^i - x_{t,0}^i = -\eta_L \sum_{k=0}^{K-1} g_{t,k}^i$
10:    At Server:
11:    Receive $\Delta_t^i, i \in S_t$
12:    let $\Delta_t = \frac{1}{|S_t|} \sum_{i \in S_t} \frac{n_i}{np_i^t} \Delta_t^i$
13:    Server update: $x_{t+1} = x_t + \eta\Delta_t$
14:    Broadcast $x_{t+1}$ to clients

---

Compared with existing unbiased sampling FL works, including FedAvg and FedIS (others), our analysis on FedIS entails: (1) **A tighter Local Update Bound Lemma:** We establish Lemma C.3 using Assumption 3, diverging from the stronger assumption $\|\nabla F_i(x_t)) - \nabla f(x_t)\|^2 \le \sigma_G^2$ (used in [65, 47]), and the derived Lemma C.3 achieves a tighter upper bound than other works (Lemma 4 in [47], Lemma 2 in [65]). (2) **A tighter upper bound on aggregated model updates $E\|\Delta_t\|^2$:** By fully utilizing the nature of unbiased sampling, we convert the bound analysis of $A_2 = E\|\Delta_t\|^2$ equally to a bound analysis of participant variance $V\left(\frac{1}{mp_i^t}\hat{g}_i^t\right)$ and aggregated model update with full user participation. In contrast, instead of exploring the property the unbiased sampling, [47] repeats to use Lemma 4 and [65] uses Lemma 2 for bound $A_2$. This inequality transform imposes a loose upper bound for $A_2$, resulting in a convergence variance term determined by $\eta_L^3$, which reacts to the rate order being $\mathcal{O}(T^{-\frac{2}{3}})$. (3) **Relying on a more lenient assumption:** Beyond the aforementioned analytical improvement, our IS analysis obviates the necessity for unusual assumptions in other FedIS analysis such as Mix Participation [35] and $\rho$-Assumption [4].

**Remark 3.4** (Extending FedIS to practical algorithm). *The existing analysis of IS algorithms [35, 5] relies on information from full clients, which is not available in partial participation FL. We propose a practical algorithm for FedIS that only uses information from available clients and provide its convergence rate in Corollary 4.1 in Section 4.*

Despite its success in reducing the variance term in the convergence rate, FedIS is far from optimal due to issues with high gradient similarity and the potential for further minimizing the variance term (i.e., the global variance $\sigma_G$ and local variance $\sigma_L$ in $\Phi$). In the next section, we will discuss how to address this challenging variance term.

### 3.3 An Improved Convergence Analysis for FedDELTA

FedIS and FedDELTA have different approaches to analyzing objectives, with FedIS analyzing the global objective and FedDELTA analyzing a surrogate objective $\tilde{f}(x)$ (cf. (7)). This leads to different convergence variance and sampling probabilities between the two methods. A flowchart (Figure 8 in Appendix E) has been included to illustrate the differences between FedIS and FedDELTA.

**The limitations of FedIS.** As shown in Figure 1, IS may have excessive similar gradient selection. The variance $\Phi$ in (4) reveals that the standard IS strategy can only control the update variance $\mathrm{Var}(1/(mp_i^t)\hat{g}_i^t$, leaving other terms in $\Phi$, namely $\sigma_L$ and $\sigma_G$, untouched. Therefore, the standard IS is ineffective at addressing the excessive similar gradient selection problem, motivating the need for a new sampling strategy to address the issue of $\sigma_L$ and $\sigma_G$.

**The decomposition of the global objective.** As inspired by the proof of Theorem 3.1 as well as the corresponding Lemma C.1 (stated in Appendix) proposed for unbiased sampling, the gradient of global objective can be decomposed into the gradient of surrogate objective $\tilde{f}(x_t)$ and update gap,

$$\mathbb{E}\|\nabla f(x_t)\|^2 = \mathbb{E}\left\|\nabla \tilde{f}_{S_t}(x_t)\right\|^2 + \chi_t^2, \tag{6}$$

where $\chi_t = \mathbb{E}\left\|\nabla \tilde{f}_{S_t}(x_t) - \nabla f(x_t)\right\|$ is the update gap.

Intuitively, the surrogate objective represents the practical objective of the participating clients in each round, while the update gap $\chi_t$ represents the distance between partial client participation and full client participation. The convergence behavior of the update gap $\chi_t^2$ is analogous to the update variance in $\Phi$, and the convergence of the surrogate objective $\mathbb{E}\left\|\nabla \tilde{f}_{S_t}(x_t)\right\|^2$ depends on the other variance terms in $\Phi$, namely the local variance and global variance.

Minimizing the surrogate objective allows us to further reduce the variance of convergence, and we will focus on analyzing surrogate objective below. We first formulate the surrogate objective with an arbitrary unbiased sampling probability.

**Surrogate objective formulation.** The expression of the surrogate objective relies on the property of IS. In particular, IS aims to substitute the original sampling distribution $p(z)$ with another arbitrary sampling distribution $q(z)$ while keeping the expectation unchanged: $\mathbb{E}_{q(z)}[F_i(z)] = \mathbb{E}_{p(z)}[q_i(z)/p_i(z)F_i(z)]$. According to the Monte Carlo method, when $q(z)$ follows the uniform distribution, we can estimate $\mathbb{E}_{q(z)}[F_i(z)]$ by $1/m\sum_{i=1}^m F_i(z)$ and $\mathbb{E}_{p(z)}[q_i(z)/p_i(z)F_i(z)]$ by $1/n\sum_{i\in S_t}1/mp_i F_i(z)$, where $m$ and $|S_t| = n$ are the sample sizes.

Based on IS property, we formulate the surrogate objective:

$$\tilde{f}_{S_t}(x_t) = \tfrac{1}{n}\sum_{i\in S_t}\tfrac{1}{mp_i^t}F_i(x_t), \tag{7}$$

where $m$ is the total number of clients, $|S_t| = n$ is the number of participating clients in each round, and $p_t^i$ is the probability that client $i$ is selected at round $t$.

As noted in Lemma C.2 in the appendix, we have:[2]:

$$\min_{t\in[T]}\mathbb{E}\|\nabla f(x_t)\|^2 = \min_{t\in[T]}\mathbb{E}\|\nabla \tilde{f}(x_t)\|^2 + \mathbb{E}\|\chi_t^2\| \leq \min_{t\in[T]}2\mathbb{E}\|\nabla \tilde{f}(x_t)\|^2. \tag{8}$$

Then the convergence rate of the global objective can be formulated as follows:

**Theorem 3.5** (Convergence upper bound of FedDELTA). *Under Assumption 1–3 and let local and global learning rates $\eta$ and $\eta_L$ satisfy $\eta_L < 1/(2\sqrt{10K}L\sqrt{\frac{1}{n}\sum_{l=1}^m\frac{1}{mp_l^t}})$ and $\eta\eta_L \leq 1/KL$, the minimal gradient norm will be bounded as below:*

$$\min_{t\in[T]}\mathbb{E}\|\nabla f(x_t)\|^2 \leq \frac{f^0 - f^*}{c\eta\eta_L KT} + \frac{\tilde{\Phi}}{c}, \tag{9}$$

---

[2]With slight abuse of notation, we use the $\tilde{f}(x_t)$ for $\tilde{f}_{S_t}(x_t)$ in this paper.

where $f^0 = f(x_0)$, $f^* = f(x_*)$, $c$ is a constant, and the expectation is over the local dataset samples among all workers. The combination of variance $\tilde{\Phi}$ represents combinations of local variance and client gradient diversity.

We derive the convergence rates for both sampling with replacement and sampling without replacement. For sampling without replacement:

$$\tilde{\Phi} = \frac{5L^2 K \eta_L^2}{2mn} \sum_{i=1}^m \frac{1}{p_i^t}(\sigma_{L,i}^2 + 4K\zeta_{G,i,t}^2) + \frac{L\eta_L\eta}{2n} \sum_{i=1}^m \frac{1}{m^2 p_i^t}\sigma_{L,i}^2 \,. \tag{10}$$

For sampling with replacement,

$$\tilde{\Phi} = \frac{5L^2 K \eta_L^2}{2m^2} \sum_{i=1}^m \frac{1}{p_i^t}(\sigma_{L,i}^2 + 4K\zeta_{G,i,t}^2) + \frac{L\eta_L\eta}{2n} \sum_{i=1}^m \frac{1}{m^2 p_i^t}\sigma_{L,i}^2 \,, \tag{11}$$

where $\zeta_{G,i,t} = \|\nabla F_i(x_t) - \nabla f(x_t)\|$ and let $\zeta_G$ be a upper bound for all $i$, i.e., $\zeta_{G,i,t} \le \zeta_G$. The proof details of Theorem 3.5 can be found in Appendix E.

**Remark 3.6** (The novelty of DELTA analysis). *IS focuses on minimizing $V\left(\frac{1}{mp_i^t}\hat{g}_i^t\right)$ in convergence variance $\Phi$ (Eq. (4)), while leaving other terms like $\sigma_L$ and $\sigma_G$ unreduced. Unlike IS roles to reduce the update gap, we propose analyzing the surrogate objective for additional variance reduction.*

*Compared with FedIS, our analysis of DELTA entails: **Focusing on surrogate objective, introducing a novel Lemma and bound:** (1) we decompose global objective convergence into surrogate objective and update gap (6). For surrogate objective analysis, we introduce Lemma E.8 to bound local updates. (2) leveraging the unique surrogate objective expression and Lemma E.8, we link sampling probability with local variance and gradient diversity, deriving novel upper bounds for $A_1$ and $A_2$. (3) by connecting update gap's convergence behavior to surrogate objective through Definition E.1 and Lemma C.2, along with (6), we establish $\tilde{\Phi}$ as the new global objective convergence variance. **Optimizing convergence variance through novel $\tilde{\Phi}$:** FedIS aims to reduce the update variance term $V(\frac{1}{(mp_i^t)}\hat{g}_i^t)$ in $\Phi$, while FedDELTA aims to minimize the entire convergence variance $\tilde{\Phi}$, which is composed of both gradient diversity and local variance. By minimizing $\tilde{\Phi}$, we get the sampling method DELTA, which further reduces the variance terms of $\Phi$ that cannot be minimized through IS.*

### 3.4 Proposed Sampling Strategy: DELTA

The expression of the convergence upper bound suggests that utilizing sampling to optimize the convergence variance can accelerate the convergence. Hence, we can formulate an optimization problem that minimizes the variance $\tilde{\Phi}$ with respect to the proposed sampling probability $p_i^t$:

$$\min_{p_i^t} \tilde{\Phi} \quad \text{s.t.} \quad \sum_{i=1}^m p_i^t = 1 \,, \tag{12}$$

where $\tilde{\Phi}$ is a linear combination of local variance $\sigma_{L,i}$ and gradient diversity $\zeta_{G,i,t}$ (cf. Theorem 3.5).

**Corollary 3.7** (Optimal sampling probability of DELTA). *By solving the above optimization problem, the optimal sampling probability is determined as follows:*

$$p_i^t = \frac{\sqrt{\alpha_1 \zeta_{G,i,t}^2 + \alpha_2 \sigma_{L,i}^2}}{\sum_{j=1}^m \sqrt{\alpha_1 \zeta_{G,j,t}^2 + \alpha_2 \sigma_{L,j}^2}} \,, \tag{13}$$

*where $\alpha_1$ and $\alpha_2$ are constants defined as $\alpha_1 = 20K^2 L\eta_L$ and $\alpha_2 = 5KL\eta_L + \frac{\eta}{n}$.*

**Remark 3.8.** *We note that a tension exists between the optimal sampling probability (13) and the setting of partial participation for FL. Thus, we also provide a practical implementation version for DELTA and analyze its convergence in Section 4. In particular, we will show that the convergence rate of the practical implementation version keeps the same order with a coefficient difference.*

**Corollary 3.9** (Convergence rate of FedDELTA). *Let $\eta_L = \mathcal{O}\left(\frac{1}{\sqrt{T}KL}\right)$, $\eta = \mathcal{O}\left(\sqrt{Kn}\right)$ and substitute the optimal sampling probability (13) back to $\tilde{\Phi}$. Then for sufficiently large T, the expected norm of DELTA algorithm 1 satisfies:*

$$\min_{t \in [T]} \mathbb{E}\|\nabla f(x_t)\|^2 \le \mathcal{O}\left(\frac{f^0 - f^*}{\sqrt{nKT}}\right) + \underbrace{\mathcal{O}\left(\frac{\sigma_L^2}{\sqrt{nKT}}\right) + \mathcal{O}\left(\frac{\sigma_L^2 + 4K\zeta_G^2}{KT}\right)}_{\text{order of } \tilde{\Phi}} \,. \tag{14}$$

**Difference between FedDELTA and FedIS.** The primary distinction between FedDELTA and FedIS lies in the difference between $\tilde{\Phi}$ and $\Phi$. FedIS aims to decrease the update variance term $\mathrm{Var}(1/(mp_i^t)\hat{g}_i^t)$ in $\Phi$, while FedDELTA aims to reduce the entire quantity $\tilde{\Phi}$, which is composed of both gradient diversity and local variance. By minimizing $\tilde{\Phi}$, we can further reduce the terms of $\Phi$ that cannot be minimized through FedIS. This leads to different expressions for the optimal sampling probability. The difference between the two resulting update gradients is discussed in Figure 3. Additionally, as seen in Table 1, FedDELTA achieves a superior convergence rate of $\mathcal{O}(G^2/\epsilon^2)$ compared to other unbiased sampling algorithms.

**Compare DELTA with uniform sampling.** According to the Cauchy-Schwarz inequality, DELTA is at least better than uniform sampling by reducing variance: $\frac{\tilde{\Phi}_{\mathrm{uniform}}}{\tilde{\Phi}_{\mathrm{DELTA}}} = \frac{m\sum_{i=1}^m\left(\sqrt{\alpha_1\sigma_L^2+\alpha_2\zeta_{G,i,t}^2}\right)^2}{\left(\sum_{i=1}^m\sqrt{\alpha_1\sigma_L^2+\alpha_2\zeta_{G,i,t}^2}\right)^2} \geq 1$.

This implies that DELTA does reduce the variance, especially when $\frac{\left(\sum_{i=1}^m\sqrt{\alpha_1\sigma_L^2+\alpha_2\zeta_{G,i,t}^2}\right)^2}{\sum_{i=1}^m\left(\sqrt{\alpha_1\sigma_L^2+\alpha_2\zeta_{G,i,t}^2}\right)^2} \ll m$.

**The significance of DELTA.** (1) DELTA is the first unbiased sampling algorithm, to the best of our knowledge, that considers both gradient diversity and local variance in sampling, accelerating convergence. (2) Developing DELTA inspires an improved convergence analysis by focusing on the surrogate objective, leading to a superior convergence rate for FL. (3) Moreover, DELTA can be seen as an unbiased version with the complete theoretical justification for the existing heuristic or biased diversity sampling algorithm of FL, such as [2].

## 4 FedPracDELTA and FedPracIS: The Practical Algorithms

The gradient-norm-based sampling method necessitates the calculation of the full gradient in every iteration [10, 70]. However, acquiring each client's gradient in advance is generally impractical in FL. To overcome this obstacle, we leverage the gradient from the previous participated round to estimate the gradient of the current round, thus reducing computational resources [49].

For FedPracIS, at round 0, all probabilities are set to $1/m$. Then, during the $i_{th}$ iteration, once participating clients $i \in S_t$ have sent the server their updated gradients, the sampling probabilities are updated as follows:

$$p_{i,t+1}^* = \frac{\|\hat{g}_{i,t}\|}{\sum_{i\in S_t}\|\hat{g}_{i,t}\|}\Big(1 - \sum_{i\in S_t^c}p_{i,t}^*\Big), \tag{15}$$

where the multiplicative factor ensures that all probabilities sum to 1. The FedPracIS algorithm is shown in Algorithm 2 of Appendix D.

For FedPracDELTA, we use the average of the latest participated clients' gradients to approximate the true gradient of the global model. For local variance, it is obtained by the local gradient's variance over local batches. Specifically, $\zeta_{G,i,t} = \|\hat{g}_{i,t} - \nabla\hat{f}(x_t)\|$, where $\nabla\hat{f}(x_t) = \frac{1}{n}\sum_{i\in S_t}\hat{g}_{i,t} = \frac{1}{n}\sum_{i\in S_t}\sum_{k=0}^{K-1}\nabla F_i(x_{k,t}^i,\xi_{k,t}^i)$ and $\sigma_{L,i}^2 = \frac{1}{|B|}\sum_{b\in B}(\hat{g}_{i,t}^b - \frac{1}{|B|}\sum_{b\in B}\hat{g}_{i,t}^b)^2$, where $b \in B$ is the local data batch. Then the sampling probabilities are updated as follows:

$$p_{i,t+1}^* = \frac{\sqrt{\alpha_1\zeta_{G,i,t}^2 + \alpha_2\sigma_{L,i}^2}}{\sum_{i\in S_t}\sqrt{\alpha_1\zeta_{G,i,t}^2 + \alpha_2\sigma_{L,i}^2}}\Big(1 - \sum_{j\in S_t^c}p_{i,t}^*\Big). \tag{16}$$

The FedPracDELTA algorithm is shown in Algorithm 1. Specifically, for $\alpha$, the default value is 0.5, whereas $\zeta_G$ and $\sigma_L$ can be implemented by computing the locally obtained gradients.

**Assumption 4** (Local gradient norm bound). *The gradients $\nabla F_i(x)$ are uniformly upper bounded (by a constant $G > 0$) $\|\nabla F_i(x)\|^2 \leq G^2, \forall i$.*

Assumption 4 is a general assumption in IS community to bound the gradient norm [70, 10, 23], and it is also used in the FL community to analyze convergence [2, 68]. This assumption tells us a useful fact that will be used later: $\|\nabla F_i(x_{t,k},\xi_{t,k})/\nabla F_i(x_{s,k},\xi_{s,k})\| \leq U$. While, for DELTA, the assumption used is a more relaxed version of Assumption 4, namely, $\mathbb{E}\|\nabla F_i(x) - \nabla f(x)\|^2 \leq G^2$ (further details are provided in Appendix G).

**Corollary 4.1** (Convergence rate of FedPracIS). *Under Assumption 1-4, the expected norm of FedPracIS will be bounded as follows:*

$$\min_{t\in[T]} E\|\nabla f(x_t)\|^2 \leq \mathcal{O}\Big(\frac{f^0-f^*}{\sqrt{nKT}}\Big) + \mathcal{O}\Big(\frac{\sigma_L^2}{\sqrt{nKT}}\Big) + \mathcal{O}\Big(\frac{M^2}{T}\Big) + \mathcal{O}\Big(\frac{KU^2\sigma_{G,s}^2}{\sqrt{nKT}}\Big), \tag{17}$$

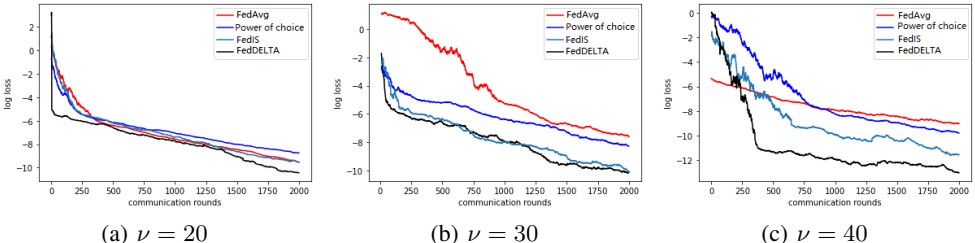

(a) $\nu = 20$          (b) $\nu = 30$          (c) $\nu = 40$

Figure 4: **Performance of different algorithms on the regression model.** The loss is calculated by $f(x,y) = \left\| y - log^{((A_i x - b_i)^2/2)} \right\|^2$, $A_i = 10$, $b_i = 1$. The logarithm of global loss is reported for various degrees of gradient noise, $\nu$, and all methods are well-tuned to yield the best results for each algorithm under each setting.

where $M = \sigma_L^2 + 4K\sigma_{G,s}^2$, $\sigma_{G,s}$ is the gradient dissimilarity bound of round $s$, and $\|\nabla F_i(x_{t,k}, \xi_{t,k})/\nabla F_i(x_{s,k}, \xi_{s,k})\| \leq U$ for all $i$ and $k$.

**Corollary 4.2** (Convergence rate of FedPracDELTA). *Under Assumption 1-4, the expected norm of FedPracDELTA satisfies:*

$$\min_{t \in [T]} \mathbb{E}\|\nabla f(x_t)\|^2 \leq \mathcal{O}\left(\frac{f^0 - f^*}{\sqrt{nKT}}\right) + \mathcal{O}\left(\frac{\tilde{U}^2 \sigma_{L,s}^2}{\sqrt{nKT}}\right) + \mathcal{O}\left(\frac{\tilde{U}^2 \sigma_{L,s}^2 + 4K\tilde{U}^2 \zeta_{G,s}^2}{KT}\right), \quad (18)$$

*where $\tilde{U}$ is a constant that $\|\nabla F_i(x_t) - \nabla f(x_t)\|/\|\nabla F_i(x_s) - \nabla f(x_s)\| \leq \tilde{U}_1 \leq \tilde{U}$ and $\|\sigma_{L,t}/\sigma_{L,s}\| \leq \tilde{U}_2 \leq \tilde{U}$, and $\zeta_{G,s}$ is the gradient diversity bound of round $s$ for all clients.*

**Remark 4.3.** *The analysis of the FedPracIS and FedPracDELTA is independent of the unavailable information in the partial participation setting. The convergence rates are of the same order as that of our theoretical algorithm but with an added coefficient constant term that limits the gradient changing rate, as shown in Table 1.*

The complete derivation and discussion of the practical algorithm can be found in Appendix G.

## 5 Experiments

In this section, we evaluate the efficiency of the theoretical algorithm FedDELTA and the practical algorithm FedPracDELTA on various datasets. Our code is available at https://github.com/L3030/DELTA_FL.

**Datasets.** (1) We evaluate FedDELTA on synthetic data and split-FashionMNIST. The synthetic data follows $y = \log\left((A_i x - b_i)^2/2\right)$ and "split" means letting $10\%$ of clients own $90\%$ of the data. (2) We evaluate FedPracDELTA on non-iid FashionMNIST, CIFAR-10 and LEAF [3]. Details of data generation and partitioning are provided in Appendix H.2.

**Baselines and Models.** We compare our algorithm, Fed(Prac)DELTA (Algorithm 1), with Fed(Prac)IS (Algorithm 2 in Appendix D), FedAVG [38], which uses random sampling, and Power-of-choice [7], which uses loss-based sampling and Cluster-based IS [52]. We utilize the regression model on synthetic date, the CNN model on Fashion-MNIST and Leaf, and the ResNet-18 on CIFAR-10. All algorithms are compared under the same experimental settings, such as lr and batch size. Full details of the sampling process of baselines and the setup of experiments are provided in Appendix H.2.

The maximum values reported in Table 2 are observed during the last 4% of rounds, where these algorithms have already reached convergence. The term 'maximum five accuracies' refers to the mean of the five highest accuracy values obtained within the plateau region of the accuracy curve.

**Figure 4 illustrates the theoretical FedDELTA outperforms other biased and unbiased methods in convergence speed on synthetic datasets.** The superiority of the theoretical DELTA is also confirmed on split-FashionMNIST, as shown in Appendix H in Figure 14(a). Additional experimental results, which include a range of different choices of regression parameters $A_i$, $b_i$, noise $\nu$, and client numbers, are presented in Figure 11, Figure 12, and Figure 13 in Appendix H.3.

**Table 2 shows the FedPracDELTA has better performance in accuracy, communication rounds, and training wall-clock times.** Notably, FedPracDELTA significantly accelerates convergence by requiring fewer training rounds and less time to achieve the threshold accuracy in FashionMNIST,

Table 2: **Performance of algorithms over various datasets.** We run 500 communication rounds on FashionM-NIST, CIFAR-10, FEMNIST, and CelebA for each algorithm. We report the mean of maximum 5 accuracies for test datasets and the average number of communication rounds and time to reach the threshold accuracy.

| Algorithm | FashionMNIST | | | CIFAR-10 | | |
|---|---|---|---|---|---|---|
| | Acc (%) | Rounds for 70% | Time (s) for 70% | Acc (%) | Rounds for 54% | Time (s) for 54% |
| FedAvg | 70.35±0.51 | 426 (1.0×) | 1795.12 (1.0×) | 54.28±0.29 | 338 (1.0×) | 3283.14 (1.0×) |
| Cluster-based IS | 71.21±0.24 | 362 (1.17×) | 1547.41 (1.16×) | 54.83±0.02 | 323 (1.05×) | 3188.54 (1.03×) |
| FedPracIS | 71.69±0.43 | 404 (1.05×) | 1719.26 (1.04×) | 55.05±0.27 | 313 (1.08×) | 3085.05 (1.06×) |
| FedPracDELTA | **72.10±0.49** | **322 (1.32×)** | **1372.33 (1.31×)** | **55.20±0.26** | **303 (1.12×)** | **2989.98 (1.1×)** |

| Algorithm | FEMNIST | | | CelebA | | |
|---|---|---|---|---|---|---|
| | Acc (%) | Rounds for 70% | Time (s) for 70% | Acc (%) | Rounds for 85% | Time (s) for 85% |
| FedAvg | 71.82±0.93 | 164 (1.0×) | 330.02 (1.0×) | 85.92±0.89 | 420 (1.0×) | 3439.81 (1.0×) |
| Cluster-based IS | 70.42±0.66 | 215 (0.76×) | 453.56 (0.73×) | 86.77±0.11 | 395 (1.06×) | 3474.50 (1.01×) |
| FedPracIS | 80.11±0.29 | 110 (1.51×) | 223.27 (1.48×) | 88.12±0.71 | 327 (1.28×) | 2746.82 (1.25×) |
| FedPracDELTA | **81.44±0.28** | **98 (1.67×)** | **198.95 (1.66×)** | **89.67±0.56** | **306 (1.37×)** | **2607.12 (1.32×)** |

Table 3: **Performance of sampling algorithms integration with other optimization methods on FEMNIST.** PracIS and PracDELTA are the sampling methods of Algorithm FedPracIS and FedPracDELTA, respectively, using the sampling probabilities defined in equations (15) and (16). For proximal and momentum methods, we use the default hyperparameter setting $\mu = 0.01$ and $\gamma = 0.9$.

| Backbone with Sampling | Uniform Sampling | | Cluster-based IS | | PracIS | | PracDELTA | |
|---|---|---|---|---|---|---|---|---|
| | Acc (%) | Rounds for 80% | Acc (%) | Rounds for 80% | Acc (%) | Rounds for 80% | Acc (%) | Rounds for 80% |
| FedAvg | 71.82±0.93 | 164 (for 70%) | 70.42±0.66 | 215 (for 70%) | 80.11±0.29 | 110 (for 70%) | **81.44±0.28** | **98** (for 70%) |
| FedAvg + momentum | 80.86±0.49 | 268 | 80.86±0.49 | 281 | 81.80±0.05 | 246 | **82.58±0.44** | **200** |
| FedAvg + proximal | 81.41±0.34 | 313 | 80.88±0.38 | 326 | 81.28±0.25 | 289 | **82.54±0.57** | **245** |

CIFAR-10, FEMNIST, and CelebA. Additionally, on the natural federated dataset LEAF (FEMNIST and CelebA), our results demonstrate that both FedPracDELTA and FedPracIS exhibit substantial improvements over FedAvg. Figure 14(b) in Appendix H.3 illustrates the superior convergence of FedPracDELTA, showcasing the accuracy curves of sampling algorithms on FEMNIST.

**Table 3 demonstrates that when compatible with momentum or proximal regularization, our method keeps its superiority in convergence.** We combine various optimization methods such as proximal regularization [29], momentum [34], and VARP [18] with sampling algorithms to assess their performance on FEMNIST and FashionMNIST. Additional results for proximal and momentum on CIFAR-10, and for VARP on FashionMNIST, are available in Table 4 and Table 5 in Appendix H.3.

**Ablation studies.** We also provide ablation studies of heterogeneity $\alpha$ in Table 9 and the impact of the number of sampled clients on accuracy in Figure 15 in Appendix H.3.

# 6 Conclusions, Limitations, and Future Works

This work studies the unbiased client sampling strategy to accelerate the convergence speed of FL by leveraging diverse clients. To address the prevalent issue of full-client gradient dependence in gradient-based FL [36, 4], we extend the theoretical algorithm DELTA to a practical version that utilizes information from the available clients.

Nevertheless, addressing the backdoor attack defense issue remains crucial in sampling algorithms. Furthermore, there is still significant room for developing an efficient and effective practical algorithm for gradient-based sampling methods. We will prioritize this as a future research direction.

# 7 Acknowledgement

This work is supported in part by the National Natural Science Foundation of China under Grant No. 62001412, in part by the funding from Shenzhen Institute of Artificial Intelligence and Robotics for Society, in part by the Shenzhen Key Lab of Crowd Intelligence Empowered Low-Carbon Energy Network (Grant No. ZDSYS20220606100601002), and in part by the Guangdong Provincial Key Laboratory of Future Networks of Intelligence (Grant No. 2022B1212010001). This work is also supported in part by the Research Center for Industries of the Future (RCIF) at Westlake University, and Westlake Education Foundation.

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

# Contents of Appendix

## A   An Expanded Version of The Related Work

FedAvg is proposed by [38] as a de facto algorithm of FL, in which multiple local SGD steps are executed on the available clients to alleviate the communication bottleneck. While communication efficient, heterogeneity, such as system heterogeneity [29, 31, 59, 39, 9], and statistical/objective heterogeneity [33, 21, 29, 59, 15], results in inconsistent optimization objectives and drifted clients models, impeding federated optimization considerably.

**Objective inconsistency in FL.** Several works also encounter difficulties from the objective inconsistency caused by partial client participation [31, 7, 2]. [31, 7] use the local-global gap $f^* - \frac{1}{m}\sum_{i=1}^{m} F_i^*$ to measure the distance between the global optimum and the average of all local personal optima, where the local-global gap results from objective inconsistency at the final optimal point. In fact, objective inconsistency occurs in each training round, not only at the final optimal point. [2] also encounter objective inconsistency caused by partial client participation. However, they use $|\frac{1}{n}\sum_{i=1}^{n} \nabla F_i(x_t) - \nabla f(x_t)| \le \epsilon$ as an assumption to describe such update inconsistency caused by objective inconsistency without any analysis on it. To date, the objective inconsistency caused by partial client participation has not been fully analyzed, even though it is prevalent in FL, even in homogeneous local updates. Our work provides a fundamental convergence analysis on the influence of the objective inconsistency of partial client participation.

**Client selection in FL.** In general, sampling methods in federated learning (FL) can be classified as biased or unbiased. Unbiased sampling guarantees that the expected value of client aggregation is equal to that of global deterministic aggregation when all clients participate. Conversely, biased sampling may result in suboptimal convergence. A prominent example of unbiased sampling in FL is multinomial sampling (MD), which samples clients based on their data ratio [59, 12]. Additionally, importance sampling (IS), an unbiased sampling method, has been utilized in FL to reduce convergence variance. For instance, [4] use update norm as an indicator of importance to sample clients, [49] sample clients based on data variability, and [40] use test accuracy as an estimation of importance. Meanwhile, various biased sampling strategies have been proposed to speed up training, such as selecting clients with higher loss [7], as many clients as possible under a threshold [45], clients

with larger updates [48], and greedily sampling based on gradient diversity [2]. However, these biased sampling methods can exacerbate the negative effects of objective inconsistency and only converge to a neighboring optimal solution. Another line of research focuses on reinforcement learning for client sampling, treating each client as an agent and aiming to find the optimal action [69, 62, 6, 53, 67]. There are also works that consider online FL, in which the client selection must consider the client's connection ability [44, 17, 26, 71, 46, 8]. Recently, cluster-based client selection has gained some attention in FL [12, 64, 42, 52, 37, 50, 25, 41, 61]. Though clustering adds additional computation and memory overhead, [12, 52] show that it is helpful for sampling diverse clients and reducing variance. Although some studies employ adaptive cluster-based IS to address the issue of slow convergence due to small gradient groups [52, 11], these approaches differ from our method as they still require an additional clustering operation. The proposed DELTA [3] in Algorithm 1 can be viewed as a muted version of the diverse client clustering algorithm, while promising to be unbiased.

**Importance sampling.** Importance sampling is a statistical method that allows for the estimation of certain quantities by sampling from a distribution that is different from the distribution of interest. It has been applied in a wide range of areas, including Monte Carlo integration [10, 70, 1], Bayesian inference [22, 23], and machine learning [54, 19].

In a recent parallel work, [49] demonstrated mean square convergence of strongly convex federated learning under the assumption of a bounded distance between the global optimal model and the local optimal models.[4] analyzed the convergence of strongly convex and nonconvex federated learning by studying the improvement factor, which is the ratio of the participation variance using importance sampling and the participation variance using uniform sampling. This algorithm dynamically selects clients without any constraints on the number of clients, potentially violating the principle of partial user participation. It is worth noting that both of these sampling methods are based on the gradient norm, ignoring the effect of the direction of the gradient. Other works have focused on the use of importance sampling in the context of online federated learning, where the client selection must consider the client's connection ability. For example, [69] proposed an adaptive client selection method based on reinforcement learning, which takes into account the communication cost and the accuracy of the local model when selecting clients to participate in training. [62] also employed reinforcement learning for adaptive client selection, treating each client as an agent and aiming to find the optimal action that maximizes the accuracy of the global model.[6] introduced a bandit-based federated learning algorithm that uses importance sampling to select the most informative clients in a single communication round. [53] considered the problem of federated learning with imperfect feedback, where the global model is updated based on noisy and biased local gradients, and proposed an importance sampling method to adjust for the bias and reduce the variance of convergence.

## B  Toy Example and Experiments for Illustrating Our Observation

### B.1  Toy example

Figure 5 is a separate illustrated version of each sampling algorithm provided in Figure 3.

We consider a regression problem involving three clients, each with a unique square function: $F_1(x, y) = x^2 + y^2$; $F_2(x, y) = 4(x - \frac{1}{2})^2 + \frac{1}{2}y^2$; $F_3(x, y) = 3x^2 + \frac{3}{2}(y - 2)^2$. Suppose $(x_t, y_t) = (1, 1)$ at current round $t$, the gradients of three clients are $\nabla F_1 = (2, 2)$, $\nabla F_2 = (4, 1)$, and $\nabla F_3 = (6, -3)$. Suppose only two clients are selected to participate in training. The closer the selected user's update is to the global model, the better.

*For ideal global model*, $\nabla F_{global} = \frac{1}{3} \sum_{i=1}^{3} \nabla F_i = (4, 0)$, which is the average over all clients.

*For FedIS*, $\nabla F_{FedIS} = \frac{1}{2}(\nabla F_2 + \nabla F_3) = (5, -1)$: It tends to select Client 2 and 3 who have large gradient norms, as $\|\nabla F_3\| > \|\nabla F_2\| > \|\nabla F_1\|$.

*For DELTA*, $\nabla F_{DELTA} = \frac{1}{2}(\nabla F_1 + \nabla F_3) = (4, -\frac{1}{2})$: It tends to select Client 1 and 3 who have the largest gradient diversity than that of other clients pair, where the gradient diversity can be formulated by $div_i = \|\nabla F_i(x_t, y_t) - \nabla F_{global}(x_t, y_t)\|$ [55, 32].

---

[3]With a slight abuse of the name, we use DELTA for the rest of the paper to denote either the sampling probability or the federated learning algorithm with sampling probability DELTA, as does FedIS.

*For FedAvg*, $\nabla F_{FedAvg} = \frac{1}{2}(\nabla F_1 + \nabla F_2) = (3, \frac{3}{2})$: It assigns each client with equal sampling probability. Compared to FedIS and DELTA, FedAvg is more likely to select Client 1 and 2. To facilitate the comparison, FedAvg is assumed to select Client 1 and 2 here.

From Figure 3, we can observe that the gradient produced by DELTA is closest to that of the ideal global model. Specifically, using $L2$ norm as the distance function $\mathcal{D}$, we have $\mathcal{D}(\nabla F_{DELTA}, \nabla F_{global}) < \mathcal{D}(\nabla F_{FedIS}, \nabla F_{global}) < \mathcal{D}(\nabla F_{FedAvg}, \nabla F_{global})$. This illustrates the selection of more diverse clients better approaches the ideal global model, thereby making it more efficient.

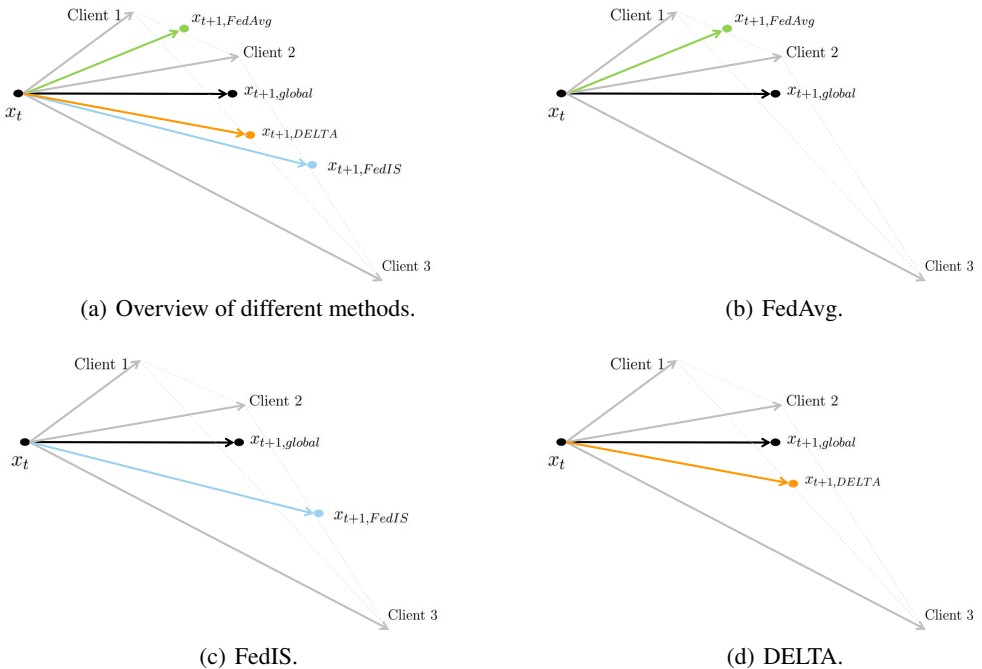

(a) Overview of different methods.          (b) FedAvg.

(c) FedIS.          (d) DELTA.

Figure 5: **Overview of objective inconsistency.** The intuition of objective inconsistency in FL is caused by client sampling. When Client 1 & 2, are selected to participate the training, then the model $x^{t+1}$ becomes $x_{FedAvg}^{t+1}$ instead of $x_{global}^{t+1}$, resulting in *objective inconsistency*. Different sampling strategies can cause different surrogate objectives, thus causing different biases. From Fig 5(a) we can see DELTA achieves minimal bias among the three unbiased sampling methods.

## B.2    Experiments for illustrating our observation.

**Experiment setting.** For the experiments to illustrate our observation in the introduction, we apply a logistic regression model on the non-iid MNIST dataset. 10 clients are selected from 200 clients to participate in training in each round. We set 2 cluster centers for cluster-based IS. And we set the mini batch-size to 32, the learning rate to 0.01, and the local update time to 5 for all methods. We run 500 communication rounds for each algorithm. We report the average of each round's selected clients' gradient norm and the minimum of each round's selected clients' gradient norm.

**Performance of gradient norm.** We report the gradient norm performance of cluster-based IS and IS to show that cluster-based IS selects clients with small gradients. As we mentioned in the introduction, the cluster-based IS always selects some clients from the cluster with small gradients, which will slow the convergence in some cases. We provide the average gradient norm comparison between IS and cluster-based IS in Figure 6(a). In addition, we also provide the minimal gradient norm comparison between IS and cluster-based IS in Figure 6(b).

**Performance of removing small gradient clusters.** We report on a comparison of the accuracy and loss performance between vanilla cluster-based IS and the removal of cluster-based IS with small gradient clusters. Specifically, we consider a setting with two cluster centers. After 250 rounds, we replace the clients in the cluster containing the smaller gradient with the clients in the cluster containing the larger gradient while maintaining the same total number of participating clients. The experimental results are shown in Figure 7. We can observe that vanilla cluster-based IS performs

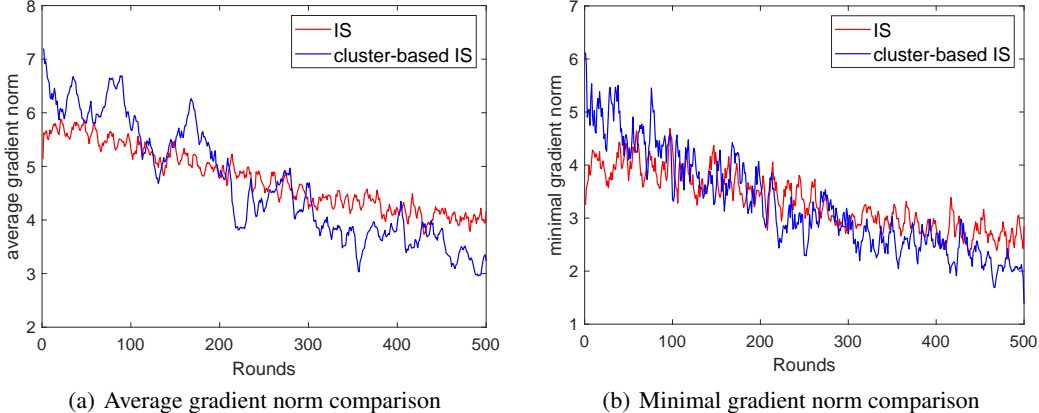

(a) Average gradient norm comparison  (b) Minimal gradient norm comparison

Figure 6: **The gradient norm comparison.** Both results indicate that cluster-based IS selects clients with small gradients after about half of the training rounds compared to IS.

worse than cluster-based IS without small gradients, indicating that small gradients are a contributing factor to poor performance.

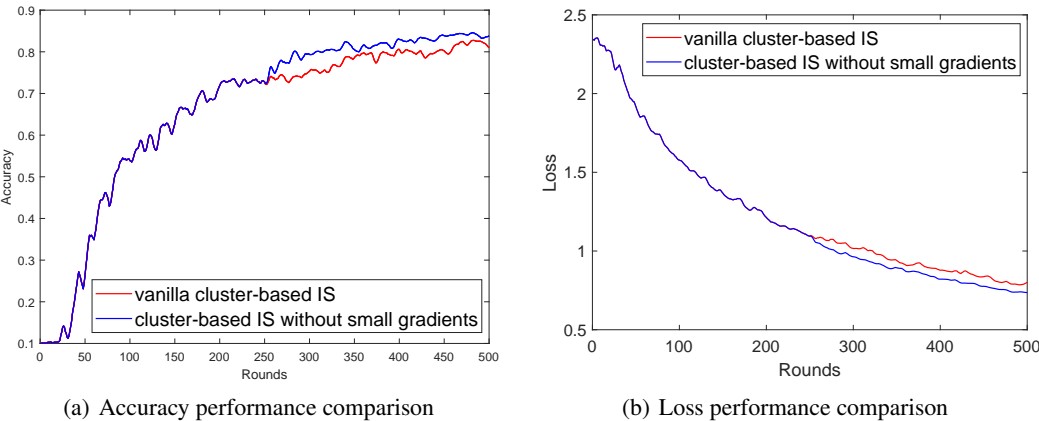

(a) Accuracy performance comparison  (b) Loss performance comparison

Figure 7: **An illustration that cluster-based IS sampling from the cluster with small gradients will slow convergence.** When the small gradient-norm cluster's clients are replaced by the clients from the large gradient-norm cluster, we see the performance improvement of cluster-based IS.

## C Techniques

Here, we present some technical lemmas that are useful in the theoretical proof. We substitute $\frac{1}{m}$ for $\frac{n_i}{N}$ to simplify the writing in all subsequent proofs. $\frac{n_i}{N}$ is the data ratio of client $i$. All of our proofs can be easily extended from $f(x_t) = \frac{1}{m}\sum_{i=1}^{m} F_i(x_t)$ to $f(x_t) = \sum_{i=1}^{m} \frac{n_i}{N} F_i(x_t)$.

**Lemma C.1.** *(Unbiased Sampling).* *Importance sampling is unbiased sampling.* $\mathbb{E}(\frac{1}{n}\sum_{i \in S_t} \frac{1}{mp_i}\nabla F_i(x_t)) = \frac{1}{m}\sum_{i=1}^{m}\nabla F_i(x_t)$ , *no matter whether the sampling is with replacement or without replacement.*

Lemma C.1 proves that the importance sampling is an unbiased sampling strategy, either in sampling with replacement or sampling without replacement.

*Proof.* For with replacement:

$$\mathbb{E}\left(\frac{1}{n}\sum_{i\in S_t}\frac{1}{mp_i^t}\nabla F_i(x_t)\right) = \frac{1}{n}\sum_{i\in S_t}\mathbb{E}\left(\frac{1}{mp_i^t}\nabla F_i(x_t)\right) = \frac{1}{n}\sum_{i\in S_t}\mathbb{E}\left(\mathbb{E}\left(\frac{1}{mp_i^t}\nabla F_i(x_t)\mid S\right)\right)$$

$$= \frac{1}{n}\sum_{i\in S_t}\mathbb{E}\left(\sum_{l=1}^{m}p_l^t\frac{1}{mp_l^t}\nabla F_l(x_t)\right) = \frac{1}{n}\sum_{i\in S_t}\nabla f(x_t) = \nabla f(x_t),$$

(19)

For without replacement:

$$\mathbb{E}\left(\frac{1}{n}\sum_{i\in S_t}\frac{1}{mp_i}\nabla F_i(x_t)\right) = \frac{1}{n}\sum_{l=1}^{m}\mathbb{E}\left(\mathbb{I}_m\frac{1}{mp_l^t}\nabla F_l(x_t)\right) = \frac{1}{n}\sum_{l=1}^{m}\mathbb{E}(\mathbb{I}_m)\times\mathbb{E}(\frac{1}{mp_l^t}\nabla F_l(x_t))$$

$$= \frac{1}{n}\mathbb{E}(\sum_{l=1}^{m}\mathbb{I}_m)\times\mathbb{E}(\frac{1}{mp_l^t}\nabla F_l(x_t)) = \frac{1}{n}n\times\sum_{l=1}^{m}p_l^t\frac{1}{mp_l^t}\nabla F_l(x_t)$$

$$= \frac{1}{n}\sum_{l=1}^{m}np_l^t\times\frac{1}{mp_l^t}\nabla F_l(x_t) = \frac{1}{m}\sum_{l=1}^{m}\nabla F_l(x_t) = \nabla f(x_t),$$

(20)

where $\mathbb{I}_m \triangleq \begin{cases} 1 & if\ x_l\in S_t\,, \\ 0 & \text{otherwise}\,. \end{cases}$

In the expectation, there are three sources of stochasticity. They are client sampling, local SGD, and the filtration of $x_t$. Therefore, the expectation is taken over all of these sources of randomness. Here, $S$ represents the sources of stochasticity other than client sampling. More precisely, $S$ represents the filtration of the stochastic process $x_j, j = 1, 2, 3\dots$ at time $t$ and the stochasticity of local SGD. □

**Lemma C.2** (update gap bound).

$$\chi^2 = \mathbb{E}\|\frac{1}{n}\sum_{i\in S_t}\frac{1}{mp_i^t}\nabla F_i(x_t) - \nabla f(x_t)\|^2 = \mathbb{E}\|\nabla\tilde{f}(x_t)\|^2 - \|\nabla f(x_t)\|^2 \le \mathbb{E}\|\nabla\tilde{f}(x_t)\|^2\,.$$

(21)

*where the first equation follows from* $\mathbb{E}[x-\mathbb{E}(x)]^2 = \mathbb{E}[x^2] - [\mathbb{E}(x)]^2$ *and Lemma C.1.*

For ease of understanding, we give a detailed derivation of the Lemma C.2.

$$\mathbb{E}\left(\|\nabla\tilde{f}(x_t) - \nabla f(x_t)\|^2 \mid S\right) = \mathbb{E}\left(\|\nabla\tilde{f}(x_t)\|^2 \mid S\right) - 2\mathbb{E}\left(\|\nabla\tilde{f}(x_t)\|\|\nabla f(x_t)\| \mid S\right)$$
$$+ \mathbb{E}\left(\|\nabla f(x_t)\|^2 \mid S\right),$$

(22)

where $\mathbb{E}(x \mid S)$ means the expectation on $x$ over the sampling space. We have $\mathbb{E}\left(\|\nabla\tilde{f}(x_t) \mid S\right) = \nabla f(x_t)$ and $\mathbb{E}\left(\|\nabla f(x_t)\|^2 \mid S\right) = \|\nabla f(x_t)\|^2$ ($\|\nabla f(x)\|$ is a constant for stochasticity $S$ and the expectation over a constant is the constant itself.)
Therefore, we conclude

$$\mathbb{E}\left(\|\nabla\tilde{f}(x_t) - \nabla f(x_t)\|^2 \mid S\right) = \mathbb{E}\left(\|\nabla\tilde{f}(x_t)\|^2 \mid S\right) - \|\nabla f(x_t)\|^2 \le \mathbb{E}\left(\|\nabla\tilde{f}(x_t)\|^2 \mid S\right).$$

(23)

We can further take the expectation on both sides of the inequality according to our needs, without changing the relationship.

The following lemma follows from Lemma 4 of [47], but with a looser condition Assumption 3, instead of $\sigma_G^2$ bound. With some effort, we can derive the following lemma:

**Lemma C.3** (Local updates bound.). *For any step-size satisfying* $\eta_L \le \frac{1}{8LK}$, *we can have the following results:*

$$\mathbb{E}\|x_{i,k}^t - x_t\|^2 \le 5K(\eta_L^2\sigma_L^2 + 4K\eta_L^2\sigma_G^2) + 20K^2(A^2 + 1)\eta_L^2\|\nabla f(x_t)\|^2\,.$$

(24)

*Proof.*

$$\mathbb{E}_t \|x_{t,k}^i - x_t\|^2$$

$$= \mathbb{E}_t \|x_{t,k-1}^i - x_t - \eta_L g_{t,k-1}^t\|^2$$

$$= \mathbb{E}_t \|x_{t,k-1}^i - x_t - \eta_L (g_{t,k-1}^t - \nabla F_i(x_{t,k-1}^i) + \nabla F_i(x_{t,k-1}^i) - \nabla F_i(x_t) + \nabla F_i(x_t))\|^2$$

$$\leq (1 + \frac{1}{2K-1}) \mathbb{E}_t \|x_{t,k-1}^i - x_t\|^2 + \mathbb{E}_t \|\eta_L (g_{t,k-1}^t - \nabla F_i(x_{t,k}^i))\|^2$$

$$+ 4K \mathbb{E}_t [\|\eta_L (\nabla F_i(x_{t,K-1}^i) - \nabla F_i(x_t))\|^2] + 4K \eta_L^2 \mathbb{E}_t \|\nabla F_i(x_t)\|^2$$

$$\leq (1 + \frac{1}{2K-1}) \mathbb{E}_t \|x_{t,k-1}^i - x_t\|^2 + \eta_L^2 \sigma_L^2 + 4K \eta_L^2 L^2 \mathbb{E}_t \|x_{t,k-1}^i - x_t\|^2$$

$$+ 4K \eta_L^2 \sigma_G^2 + 4K \eta_L^2 (A^2 + 1) \|\nabla f(x_t)\|^2$$

$$\leq (1 + \frac{1}{K-1}) \mathbb{E} \|x_{t,k-1}^i - x_t\|^2 + \eta_L^2 \sigma_L^2 + 4K \eta_L^2 \sigma_G^2 + 4K(A^2+1) \|\eta_L \nabla f(x_t)\|^2. \qquad (25)$$

Unrolling the recursion, we obtain:

$$\mathbb{E}_t \|x_{t,k}^i - x_t\|^2 \leq \sum_{p=0}^{k-1} (1 + \frac{1}{K-1})^p \left[ \eta_L^2 \sigma_L^2 + 4K \eta_L^2 \sigma_G^2 + 4K(A^2+1) \|\eta_L \nabla f(x_t)\|^2 \right]$$

$$\leq (K-1) \left[ (1 + \frac{1}{K-1})^K - 1 \right] \left[ \eta_L^2 \sigma_L^2 + 4K \eta_L^2 \sigma_G^2 + 4K(A^2+1) \|\eta_L \nabla f(x_t)\|^2 \right]$$

$$\leq 5K(\eta_L^2 \sigma_L^2 + 4K \eta_L^2 \sigma_G^2) + 20K^2(A^2+1) \eta_L^2 \|\nabla f(x_t)\|^2. \qquad (26)$$

$\square$

In the following Proposition, we will demonstrate that the convergence rate in this paper with the relaxed version of Assumption 3 remains unchanged.

**Proposition C.4** (convergence under relaxed Assumption 3 [24])**.** *The relaxed version of Assumption 3 in this paper is:*

$$\mathbb{E} \|\nabla F_i(x)\|^2 \leq 2B(f(x) - f^{inf}) + (A^2+1) \|\nabla f(x)\|^2 + \sigma_G^2. \qquad (27)$$

*Since we have $f(x) - f^{inf} \leq f^0 - f^{inf} \leq F$, where $F$ is a positive constant. This implies that we can substitute $\sigma_g$ with $2BF + \sigma_G$ in all analyses without altering the outcomes (one can directly conclude this from using the above bound in Lemma C.3). In the final convergence rate, it is straightforward to see that the convergence rate remains unchanged, yet the constant term $\sigma_g$ becomes $2BF + \sigma_G$.*

Thus, we can assert that we have furnished the analysis under the relaxed assumption condition.

# D Convergence of FedIS, Proof of Theorem 3.1

The complete version of FedIS algorithm is shown below:

We first restate the convergence theorem (Theorem 3.1) more formally, then prove the result for the nonconvex case.

**Theorem D.1.** *Under Assumptions 1–3 and the sampling strategy FedIS, the expected gradient norm will converge to a stationary point of the global objective. More specifically, if the number of communication rounds T is predetermined and the learning rate $\eta$ and $\eta_L$ are constant, then the expected gradient norm will be bounded as follows:*

$$\min_{t \in [T]} \mathbb{E} \|\nabla f(x_t)\|^2 \leq \frac{F}{c \eta \eta_L K T} + \Phi, \qquad (28)$$

*where $F = f(x_0) - f(x_*)$, $M^2 = \sigma_L^2 + 4K \sigma_G^2$, and the expectation is over the local datasets samples among workers.*

**Algorithm 2** **FedIS** and **FedPracIS** : Federated learning with importance sampling

---

**Require:** initial weights $x_0$, global learning rate $\eta$, local learning rate $\eta_l$, number of training rounds $T$
**Ensure:** trained weights $x_T$
1: **for** round $t = 1, \ldots, T$ **do**
2:     Select clients by using IS (5) or Practical IS (15) .
3:     **for** each worker $i \in S_t$,in parallel **do**
4:        $x_{t,0}^i = x_t$
5:        **for** $k = 0, \cdots, K-1$ **do**
6:           compute $g_{t,k}^i = \nabla F_i(x_{t,k}^i, \xi_{t,k}^i)$
7:           Local update:$x_{t,k+1}^i = x_{t,k}^i - \eta_L g_{t,k}^i$
8:        Let $\Delta_t^i = x_{t,K}^i - x_{t,0}^i = -\eta_L \sum_{k=0}^{K-1} g_{t,k}^i$
9:        Send gradient to server
10:    At Server:
11:    Receive $\Delta_t^i, i \in S_t$
12:    let $\Delta_t = \frac{1}{|S_t|} \sum_{i \in S_t} \frac{n_i}{np_i^t} \Delta_t^i$
13:    Server update: $x_{t+1} = x_t + \eta \Delta_t$
14:    Broadcast $x_{t+1}$ to clients

---

*Let $\eta_L < min\,(1/(8LK), C)$, where $C$ is obtained from the condition that $\frac{1}{2} - 10L^2K^2(A^2 + 1)\eta_L^2 - \frac{L^2\eta K(A^2+1)}{2n}\eta_L > 0$ ,and $\eta \leq 1/(\eta_L L)$, it then holds that:*

$$\Phi = \frac{1}{c}\Big[\frac{5\eta_L^2 L^2 K}{2m} \sum_{i=1}^{m}(\sigma_L^2 + 4K\sigma_G^2) + \frac{\eta\eta_L L}{2m}\sigma_L^2 + \frac{L\eta\eta_L}{2nK}V(\frac{1}{mp_i^t}\hat{g}_i^t)\Big]. \tag{29}$$

*where $c$ is a constant that satisfies $\frac{1}{2} - 10L^2K^2(A^2 + 1)\eta_L^2 - \frac{L^2\eta K(A^2+1)}{2n}\eta_L > c > 0$, and $V(\frac{1}{mp_i^t}\hat{g}_i^t) = E\|\frac{1}{mp_i^t}\hat{g}_i^t - \frac{1}{m}\sum_{i=1}^{m}\hat{g}_i^t\|^2$.*

**Corollary D.2.** *Suppose $\eta_L$ and $\eta$ are such that the conditions mentioned above are satisfied, $\eta_L = \mathcal{O}\left(\frac{1}{\sqrt{T}KL}\right)$ and $\eta = \mathcal{O}\left(\sqrt{Kn}\right)$, and let the sampling probability be FedIS (75). Then for sufficiently large T, the iterates of Theorem 3.1 satisfy:*

$$\min_{t \in [T]} \mathbb{E}\|\nabla f(x_t)\|^2 = \mathcal{O}\left(\frac{\sigma_L^2}{\sqrt{nKT}} + \frac{K\sigma_G^2}{\sqrt{nKT}} + \frac{\sigma_L^2 + 4K\sigma_G^2}{KT}\right). \tag{30}$$

*Proof.*

$$\mathbb{E}_t[f(x_{t+1})] \overset{(a1)}{\leq} f(x_t) + \langle \nabla f(x_t), \mathbb{E}_t[x_{t+1} - x_t]\rangle + \frac{L}{2}\mathbb{E}_t[\|x_{t+1} - x_t\|^2]$$

$$= f(x_t) + \langle \nabla f(x_t), \mathbb{E}_t[\eta\Delta_t + \eta\eta_L K\nabla f(x_t) - \eta\eta_L K\nabla f(x_t)]\rangle + \frac{L}{2}\eta^2\mathbb{E}_t[\|\Delta_t\|^2]$$

$$= f(x_t) - \eta\eta_L K\|\nabla f(x_t)\|^2 + \eta\underbrace{\langle \nabla f(x_t), \mathbb{E}_t[\Delta_t + \eta_L K\nabla f(x_t)]\rangle}_{A_1} + \frac{L}{2}\eta^2\underbrace{\mathbb{E}_t\|\Delta_t\|^2}_{A_2},$$

$$\tag{31}$$

where (a1) follows from the Lipschitz continuous condition. The expectation is conditioned on everything prior to the current step $k$ of round $t$. Specifically, it is taken over the sampling of clients, the sampling of local data, and the current round's model $x_t$.

Firstly we consider $A_1$:

$$A_1 = \langle \nabla f(x_t), \mathbb{E}_t[\Delta_t + \eta_L K \nabla f(x_t)] \rangle$$

$$= \left\langle \nabla f(x_t), \mathbb{E}_t[-\frac{1}{|S_t|} \sum_{i \in S_t} \frac{1}{mp_i^t} \sum_{k=0}^{K-1} \eta_L g_{t,k}^i + \eta_L K \nabla f(x_t)] \right\rangle$$

$$\overset{(a2)}{=} \left\langle \nabla f(x_t), \mathbb{E}_t[-\frac{1}{m} \sum_{i=1}^{m} \sum_{k=0}^{K-1} \eta_L \nabla F_i(x_{t,k}^i) + \eta_L K \nabla f(x_t)] \right\rangle$$

$$= \left\langle \sqrt{\eta_L K} \nabla f(x_t), -\frac{\sqrt{\eta_L}}{\sqrt{K}} \mathbb{E}_t[\frac{1}{m} \sum_{i=1}^{m} \sum_{k=0}^{K-1} (\nabla F_i(x_{t,k}^i) - \nabla F_i(x_t))] \right\rangle$$

$$\overset{(a3)}{=} \frac{\eta_L K}{2} \|\nabla f(x_t)\|^2 + \frac{\eta_L}{2K} \mathbb{E}_t \left\| \frac{1}{m} \sum_{i=1}^{m} \sum_{k=0}^{K-1} (\nabla F_i(x_{t,k}^i) - \nabla F_i(x_t)) \right\|^2$$

$$- \frac{\eta_L}{2K} \mathbb{E}_t \| \frac{1}{m} \sum_{i=1}^{m} \sum_{k=0}^{K-1} \nabla F_i(x_{t,k}^i) \|^2$$

$$\overset{(a4)}{\leq} \frac{\eta_L K}{2} \|\nabla f(x_t)\|^2 + \frac{\eta_L L^2}{2m} \sum_{i=1}^{m} \sum_{k=0}^{K-1} \mathbb{E}_t \left\| x_{t,k}^i - x_t \right\|^2 - \frac{\eta_L}{2K} \mathbb{E}_t \| \frac{1}{m} \sum_{i=1}^{m} \sum_{k=0}^{K-1} \nabla F_i(x_{t,k}^i) \|^2$$

$$\leq \left( \frac{\eta_L K}{2} + 10K^3 L^2 \eta_L^3 (A^2 + 1) \right) \|\nabla f(x_t)\|^2 + \frac{5L^2 \eta_L^3}{2} K^2 \sigma_L^2 + 10\eta_L^3 L^2 K^3 \sigma_G^2$$

$$- \frac{\eta_L}{2K} \mathbb{E}_t \| \frac{1}{m} \sum_{i=1}^{m} \sum_{k=0}^{K-1} \nabla F_i(x_{t,k}^i) \|^2, \tag{32}$$

where (a2) follows from Assumption 2 and LemmaC.1. (a3) is due to $\langle x, y \rangle = \frac{1}{2} \left[ \|x\|^2 + \|y\|^2 - \|x - y\|^2 \right]$ and (a4) comes from Assumption 1.

Then we consider $A_2$. Let $\hat{g}_i^t = \sum_{k=0}^{K-1} g_{i,k}^t = \sum_{k=0}^{K-1} \nabla F_i(x_{t,k}^i, \xi_{t,k}^i)$

$$A_2 = \mathbb{E}_t \|\Delta_t\|^2$$

$$= \mathbb{E}_t \left\| \eta_L \frac{1}{n} \sum_{i \in S_t} \frac{1}{mp_i^t} \sum_{k=0}^{K-1} g_{t,k}^i \right\|^2$$

$$= \eta_L^2 \frac{1}{n} \mathbb{E}_t \left\| \frac{1}{mp_i^t} \sum_{k=0}^{K-1} g_{t,k}^i - \frac{1}{m} \sum_{i=1}^{m} \sum_{k=0}^{K-1} g_{t,k}^i \right\|^2$$

$$+ \eta_L^2 \mathbb{E}_t \left\| \frac{1}{m} \sum_{i=1}^{m} \sum_{k=0}^{K-1} g_i(x_{t,k}^i) \right\|^2$$

$$= \frac{\eta_L^2}{n} V(\frac{1}{mp_i^t} \hat{g}_i^t)$$

$$+ \eta_L^2 \mathbb{E} \| \frac{1}{m} \sum_{i=1}^{m} \sum_{k=0}^{K-1} [g_i(x_{t,k}^i) - \nabla F_i(x_{t,k}^i) + \nabla F_i(x_{t,k}^i)] \|^2$$

$$\leq \frac{\eta_L^2}{n} V(\frac{1}{mp_i} \hat{g}_i^t)$$

$$+ \eta_L^2 \frac{1}{m^2} \sum_{i=1}^{m} \sum_{k=0}^{K-1} \mathbb{E}\|g_i(x_{t,k}^i) - \nabla F_i(x_{t,k}^i)\|^2 + \eta_L^2 \mathbb{E} \| \frac{1}{m} \sum_{i=1}^{m} \sum_{k=0}^{K-1} \nabla F_i(x_{t,k}^i) \|^2$$

$$\leq \frac{\eta_L^2}{n} V(\frac{1}{mp_i^t} \hat{g}_i^t) + \eta_L^2 \frac{K}{m} \sigma_L^2 + \eta_L^2 \mathbb{E} \| \frac{1}{m} \sum_{i=1}^{m} \sum_{k=0}^{K-1} \nabla F_i(x_{t,k}^i) \|^2. \tag{33}$$

The third equality follows from independent sampling.

Specifically, for sampling with replacement, due to every index being independent, we utilize $\mathbb{E}\|x_1^2 + ... + x_n\|^2 = \mathbb{E}[\|x_1\|^2 + ... + \|x_n\|^2]$.

For sampling without replacement:

$$\mathbb{E}\|\frac{1}{n}\sum_{i \in S_t}(\frac{1}{mp_i^t}\hat{g}_i^t - \frac{1}{m}\sum_{i=1}^m \hat{g}_i^t)\|^2$$

$$= \frac{1}{n^2}\mathbb{E}\|\sum_{i=1}^m \mathbb{I}_i(\frac{1}{mp_i^t}\hat{g}_i^t - \frac{1}{m}\sum_{i=1}^m \hat{g}_i^t)\|^2$$

$$= \frac{1}{n^2}\mathbb{E}\left(\|\sum_{i=1}^m \mathbb{I}_i(\frac{1}{mp_i^t}\hat{g}_i^t - \frac{1}{m}\sum_{i=1}^m \hat{g}_i^t)\|^2 \mid \mathbb{I}_i = 1\right) \times \mathbb{P}(\mathbb{I}_i = 1)$$

$$+ \frac{1}{n^2}\mathbb{E}\left(\|\sum_{i=1}^m \mathbb{I}_i(\frac{1}{mp_i^t}\hat{g}_i^t - \frac{1}{m}\sum_{i=1}^m \hat{g}_i^t)\|^2 \mid \mathbb{I}_i = 0\right) \times \mathbb{P}(\mathbb{I}_i = 0)$$

$$= \frac{1}{n}\sum_{i=1}^m p_i^t\|\frac{1}{mp_i^t}\hat{g}_i^t - \frac{1}{m}\sum_{i=1}^m \hat{g}_i^t\|^2$$

$$= \frac{1}{n}E\|\frac{1}{mp_i^t}\hat{g}_i^t - \frac{1}{m}\sum_{i=1}^m \hat{g}_i^t\|^2. \tag{34}$$

From the above, we observe that it is possible to achieve a speedup by sampling from the distribution that minimizes $V(\frac{1}{mp_i^t}\hat{g}_i^t)$. Furthermore, as we discussed earlier, the optimal sampling probability is $p_i^* = \frac{\|\hat{g}_i^t\|}{\sum_{i=1}^m \|\hat{g}_i^t\|}$. For MD sampling [31], which samples according to the data ratio, the optimal sampling probability is $p_{i,t}^* = \frac{q_i\|\hat{g}_i^t\|}{\sum_{i=1}^m q_i\|\hat{g}_i^t\|}$, where $q_i = \frac{n_i}{N}$.

Now we substitute the expressions of $A_1$ and $A_2$:

$$\mathbb{E}_t[f(x_{t+1})] \leq f(x_t) - \eta\eta_L K\|\nabla f(x_t)\|^2 + \eta\langle\nabla f(x_t), \mathbb{E}_t[\Delta_t + \eta_L K\nabla f(x_t)]\rangle + \frac{L}{2}\eta^2\mathbb{E}_t\|\Delta_t\|^2$$

$$\leq f(x_t) - \eta\eta_L K\left(\frac{1}{2} - 10L^2 K^2\eta_L^2(A^2+1)\right)\|\nabla f(x_t)\|^2 + \frac{5\eta\eta_L^3 L^2 K^2}{2}(\sigma_L^2 + 4K\sigma_G^2)$$

$$+ \frac{\eta^2\eta_L^2 KL}{2m}\sigma_L^2 + \frac{L\eta^2\eta_L^2}{2n}V(\frac{1}{mp_i^t}\hat{g}_i^t) - \left(\frac{\eta\eta_L}{2K} - \frac{L\eta^2\eta_L^2}{2}\right)\mathbb{E}_t\left\|\frac{1}{m}\sum_{i=1}^m\sum_{k=0}^{K-1}\nabla F_i(x_{t,k}^i)\right\|^2$$

$$\leq f(x_t) - c\eta\eta_L K\|\nabla f(x_t)\|^2 + \frac{5\eta\eta_L^3 L^2 K^2}{2}(\sigma_L^2 + 4K\sigma_G^2) + \frac{\eta^2\eta_L^2 KL}{2m}\sigma_L^2 + \frac{L\eta^2\eta_L^2}{2n}V(\frac{1}{mp_i^t}\hat{g}_i^t), \tag{35}$$

where the last inequality follows from $\left(\frac{\eta\eta_L}{2K} - \frac{L\eta^2\eta_L^2}{2}\right) \geq 0$ if $\eta\eta_l \leq \frac{1}{KL}$, and (a9) holds because there exists a constant $c > 0$ (for some $\eta_L$) satisfying $\frac{1}{2} - 10L^2\frac{1}{m}\sum_{i-1}^m K^2\eta_L^2(A^2+1) > c > 0$.
Rearranging and summing from $t = 0, \ldots, T-1$, we have:

$$\sum_{t=1}^{T-1} c\eta\eta_L K\mathbb{E}\|\nabla f(x_t)\|^2 \leq f(x_0) - f(x_T) + T(\eta\eta_L K)\Phi. \tag{36}$$

Which implies:

$$\min_{t\in[T]}\mathbb{E}\|\nabla f(x_t)\|^2 \leq \frac{f_0 - f_*}{c\eta\eta_L KT} + \Phi, \tag{37}$$

where

$$\Phi = \frac{1}{c} \left[ \frac{5\eta\eta_L^2 K L^2}{2}(\sigma_L^2 + 4K\sigma_G^2) + \frac{\eta\eta_L L}{2m}\sigma_L^2 + \frac{L\eta\eta_L}{2nK}V(\frac{1}{mp_i^t}\hat{g}_i^t) \right]. \tag{38}$$

$\square$

### D.1 Proof for convergence rate of FedIS (Theorem 3.1) under Assumption 1–3.

In this section, we compare the convergence rate of FedIS with and without Assumption 4. For comparison, we first provide the convergence result under Assumption 4.

First we show Assumption 4 can be used to bound the update variance $V\left(\frac{1}{mp_i^t}\hat{g}_i^t\right)$, and under the sampling probability FedIS (73):

$$V\left(\frac{1}{mp_i^t}\hat{g}_i^t\right) \leq \frac{1}{m^2}\mathbb{E}\|\sum_{i=1}^{m}\sum_{k=1}^{K}\nabla F_i(x_{t,k},\xi_{k,t})\|^2 \leq \frac{1}{m}\sum_{i=1}^{m}K\sum_{k=1}^{K}\mathbb{E}\|\nabla F_i(x_{t,k},\xi_{k,t})\|^2 \leq K^2 G^2 \tag{39}$$

While for using Assumption 3 instead of additional Assumption 4, we can also bound the update variance:

$$V\left(\frac{1}{mp_i^t}\hat{g}_i^t\right) \leq \frac{1}{m^2}\mathbb{E}\|\sum_{i=1}^{m}\sum_{k=1}^{K}\nabla F_i(x_{t,k},\xi_{k,t})\|^2 \leq \frac{1}{m}\sum_{i=1}^{m}K\sum_{k=1}^{K}\mathbb{E}\|\nabla F_i(x_{t,k},\xi_{k,t})\|^2$$
$$\leq K^2\sigma_G^2 + K^2(A^2+1)\|\nabla f(x_t)\|^2 \tag{40}$$

We replace the variance back to equation (35):

$$\mathbb{E}_t[f(x_{t+1})] \leq f(x_t) - \eta\eta_L K \|\nabla f(x_t)\|^2 + \eta\langle\nabla f(x_t), \mathbb{E}_t[\Delta_t + \eta_L K\nabla f(x_t)]\rangle + \frac{L}{2}\eta^2\mathbb{E}_t\|\Delta_t\|^2$$

$$\leq f(x_t) - \eta\eta_L K \left(\frac{1}{2} - 10L^2K^2\eta_L^2(A^2+1)\right)\|\nabla f(x_t)\|^2 + \frac{5\eta\eta_L^3 L^2 K^2}{2}(\sigma_L^2 + 4K\sigma_G^2)$$

$$+ \frac{\eta^2\eta_L^2 K L}{2m}\sigma_L^2 + \frac{L\eta^2\eta_L^2}{2n}V(\frac{1}{mp_i^t}\hat{g}_i^t) - \left(\frac{\eta\eta_L}{2K} - \frac{L\eta^2\eta_L^2}{2}\right)\mathbb{E}_t\left\|\frac{1}{m}\sum_{i=1}^{m}\sum_{k=0}^{K-1}\nabla F_i(x_{t,k}^i)\right\|^2$$

$$\leq f(x_t) - \eta\eta_L K \left(\frac{1}{2} - 10L^2K^2\eta_L^2(A^2+1) - \frac{L\eta\eta_L K(A^2+1)}{2n}\right)\|\nabla f(x_t)\|^2$$

$$+ \frac{5\eta\eta_L^3 L^2 K^2}{2}(\sigma_L^2 + 4K\sigma_G^2) + \frac{\eta^2\eta_L^2 K L}{2m}\sigma_L^2 + \frac{L\eta^2\eta_L^2}{2n}K^2\sigma_G^2$$

$$- \left(\frac{\eta\eta_L}{2K} - \frac{L\eta^2\eta_L^2}{2}\right)\mathbb{E}_t\left\|\frac{1}{m}\sum_{i=1}^{m}\sum_{k=0}^{K-1}\nabla F_i(x_{t,k}^i)\right\|^2. \tag{41}$$

This shows that the requirement for $\eta_L$ is different. It needs that there exists a constant $c > 0$ (for some $\eta_L$) satisfying $\frac{1}{2} - 10L^2K^2\eta_L^2(A^2+1) - \frac{L\eta\eta_L K(A^2+1)}{2n} > c > 0$. One can still guarantee that there exists a constant for $\eta_L$ to satisfy this inequality according to the properties of quadratic functions. Specifically, for the quadratic equation $-10L^2K^2(A^2+1)\eta_L^2 - \frac{L\eta K(A^2+1)}{2n}\eta_L + \frac{1}{2}$, we know that $-10L^2K^2(A^2+1) < 0$, $-\frac{L\eta K(A^2+1)}{2n}$ and $\frac{1}{2} > 0$. Based on the solution of quadratic equations, we can ensure that there exists a $\eta_L > 0$ solution.

Then we can substitute equation (35) with equation (41) and let $\eta_L = \mathcal{O}\left(\frac{1}{\sqrt{TKL}}\right)$ and $\eta = \mathcal{O}\left(\sqrt{Kn}\right)$, yielding the convergence rate of FedIS under Assumptions 1– 3:

$$\min_{t\in[T]}E\|\nabla f(x_t)\|^2 \leq \mathcal{O}\left(\frac{f^0 - f^*}{\sqrt{nKT}}\right) + \underbrace{\mathcal{O}\left(\frac{\sigma_L^2}{\sqrt{nKT}}\right) + \mathcal{O}\left(\frac{M^2}{T}\right) + \mathcal{O}\left(\frac{K\sigma_G^2}{\sqrt{nKT}}\right)}_{\text{order of }\Phi}. \tag{42}$$

# E  Convergence of DELTA. Proof of Theorem 3.5

## E.1  Convergence rate with improved analysis method for getting DELTA

As we see FedIS can only reduce the update variance term in $\Phi$. Since we want to reduce the convergence variance as much as possible, the other term $\sigma_L$ and $\sigma_G$ still needs to be optimized. However, it is not straightforward to derive the optimization problem from $\Phi$. In order to further reduce the variance in $\Phi$ (cf. 4), i.e., local variance ($\sigma_L$) and global variance ($\sigma_G$), we divide the convergence of the global objective into a surrogate objective and an update gap and analyze them separately. The analysis framework is shown in Figure 8.

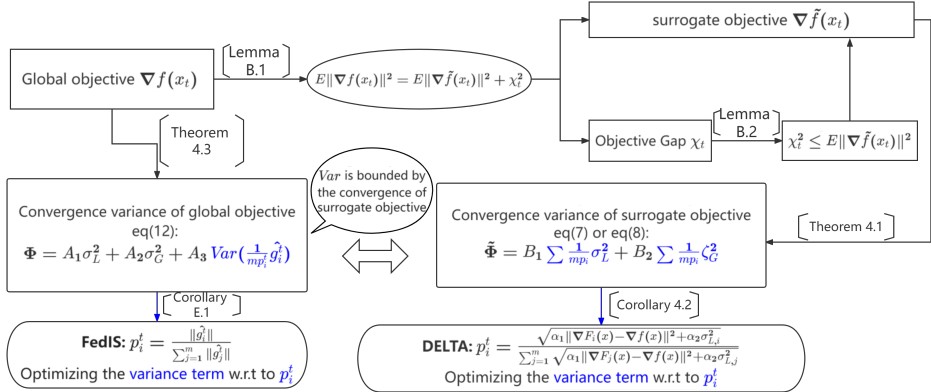

Figure 8: **Theoretical analysis flow.** The figure shows the theoretical analysis flow of FedIS (left) and DELTA (right), highlighting the differences in sampling probability due to variance.

As for the update gap, inspired by the expression form of the update variance, we formally define it as follows:

**Definition E.1** (Update gap). *In order to measure the update inconsistency, we define the update gap:*

$$\chi_t = \mathbb{E}\left[\left\|\nabla\tilde{f}(x_t) - \nabla f(x_t)\right\|\right] . \tag{43}$$

*Here, the expectation is taken over the distribution of all clients. When all clients participate, we have $\chi_t^2 = 0$. The update inconsistency exists as long as only a partial set of clients participate.*

The update gap is a direct manifestation of the objective inconsistency in the update process. The presence of an update gap makes the analysis of the global objective different from the analysis of the surrogate objective. However, by ensuring the convergence of the update gap, we can re-derive the convergence result for the global objective. Formally, the update gap allows us to connect global objective convergence and surrogate objective convergence as follows:

$$\mathbb{E}\|\nabla f(x_t)\|^2 = \mathbb{E}\|\nabla\tilde{f}(x_t)\|^2 + \chi_t^2 . \tag{44}$$

The equation follows from the property of unbiasedness, as shown in Lemma C.1.

To deduce the convergence rate of the global objective, we begin by examining the convergence analysis of the surrogate objective.

**Theorem E.2** (Convergence rate of surrogate objective). *Under Assumption 1–3 and let local and global learning rates $\eta$ and $\eta_L$ satisfy $\eta_L < 1/(\sqrt{40K}L\sqrt{\frac{1}{n}\sum_{l=1}^{m}\frac{1}{mp_l^t}})$ and $\eta\eta_L \leq 1/KL$, the minimal gradient norm of surrogate objective will be bounded as below:*

$$\min_{t\in[T]}\mathbb{E}\left\|\nabla\tilde{f}\left(x_t\right)\right\|^2 \leq \frac{f^0 - f^*}{\tilde{c}\eta\eta_L KT} + \frac{\tilde{\Phi}}{\tilde{c}} , \tag{45}$$

*where $f^0 = f(x_0)$, $f^* = f(x_*)$, the expectation is over the local dataset samples among workers.*

$\tilde{\Phi}$ **is the new combination of variance**, representing combinations of local variance and client gradient diversity.

For sampling without replacement:

$$\tilde{\Phi} = \frac{5L^2K\eta_L^2}{2mn}\sum_{i=1}^{m}\frac{1}{p_i^t}(\sigma_{L,i}^2 + 4K\zeta_{G,i}) + \frac{L\eta_L\eta}{2n}\sum_{i=1}^{m}\frac{1}{m^2p_i^t}\sigma_{L,i}^2 , \tag{46}$$

For sampling with replacement:

$$\tilde{\Phi} = \frac{5L^2 K \eta_L^2}{2m^2} \sum_{i=1}^{m} \frac{1}{p_i^t}(\sigma_{L,i}^2 + 4K\zeta_{G,i}^2) + \frac{L\eta_L \eta}{2n} \sum_{i=1}^{m} \frac{1}{m^2 p_i^t}\sigma_{L,i}^2 \tag{47}$$

where $\zeta_{G,i}$ represents client gradient diversity: $\zeta_{G,i} = \|\nabla F_i(x_t) - \nabla f(x_t)\|$ [4], and $\tilde{c}$ is a constant. The proof of Theorem E.2 is provided in Appendix E.2.1 and Appendix E.2.2. Specifically, the proof for sampling with replacement is shown in Appendix E.2.1, while the proof for sampling without replacement is shown in Appendix E.2.2.

**Remark E.3.** *We observe that there is no update variance in $\tilde{\Phi}$, but the local variance and global variance are still present. Additionally, the new combination of variance $\tilde{\Phi}$ can be minimized by optimizing the sampling probability, as will be shown later.*

**Derive the convergence from surrogate objective to global objective.** As shown in Lemma C.1, unbiased sampling guarantees that the expected partial client updates are equal to the participation of all clients. With sufficient training rounds, unbiased sampling can ensure that the update gap $\chi^2$ will converge to zero. However, we still need to know the convergence speed of $\chi_t^2$ to recover the convergence rate of the global objective. Fortunately, we can bound the convergence behavior of $\chi_t^2$ by the convergence rate of the surrogate objective according to Definition E.1 and Lemma C.2. This means that the update gap can achieve at least the same convergence rate as the surrogate objective.

**Corollary E.4** (New convergence rate of global objective). *Under Assumption 1–3 and based on the above analysis that update variance is bounded, the global objective will converge to a stationary point. Its gradient is bounded as:*

$$\min_{t\in[T]} \mathbb{E}\|\nabla f(x_t)\|^2 = \min_{t\in[T]} \mathbb{E}\|\nabla \tilde{f}(x_t)\|^2 + \mathbb{E}\|\chi_t^2\| \le \min_{t\in[T]} 2\mathbb{E}\|\nabla \tilde{f}(x_t)\|^2 \le \frac{f^0 - f^*}{c\eta\eta_L KT} + \frac{\tilde{\Phi}}{c}. \tag{48}$$

**Theorem E.5** (Restate of Theorem 3.5). *Under Assumptions 1-3 and the same conditions as in Theorem 3.1, the minimal gradient norm of the surrogate objective will be bounded as follows by setting $\eta_L = \frac{1}{\sqrt{T}KL}$ and $\eta\sqrt{Kn}$. Let the local and global learning rates $\eta$ and $\eta_L$ satisfy $\eta_L < \frac{1}{\sqrt{40K}L\sqrt{\frac{1}{n}\sum_{l=1}^{m}\frac{1}{mp_l^t}}}$ and $\eta\eta_L \le \frac{1}{KL}$. Under Assumptions 1-3 and with partial worker participation, the sequence of outputs $x_k$ generated by Algorithm 1 satisfies:*

$$\min_{t\in[T]} \mathbb{E}\|\nabla f(x_t)\|^2 \le \frac{F}{c\eta\eta_L KT} + \frac{1}{c}\tilde{\Phi}, \tag{49}$$

*where $F = f(x_0) - f(x_*)$, and the expectation is over the local dataset samplings among workers. $c$ is a constant. $\zeta_{G,i}$ is defined as client gradient diversity: $\zeta_{G,i} = \|\nabla F_i(x_t) - \nabla f(x_t)\|$.*

*For sample with replacement:* $\tilde{\Phi} = \frac{5L^2 K \eta_L^2}{2m^2} \sum_{l=1}^{m} \frac{1}{p_l^t}(\sigma_{L,l}^2 + 4K\zeta_{G,l}^2) + \frac{L\eta_L \eta}{2n} \sum_{l=1}^{m} \frac{1}{m^2 p_l^t}\sigma_{L,i}^2.$

*For sampling without replacement:* $\tilde{\Phi} = \frac{5L^2 K \eta_L^2}{2mn} \sum_{l=1}^{m} \frac{1}{p_l^t}(\sigma_{L,l}^2 + 4K\zeta_{G,l}^2) + \frac{L\eta_L \eta}{2n} \sum_{l=1}^{m} \frac{1}{m^2 p_l^t}\sigma_{L,l}^2.$

**Remark E.6** (Condition of $\eta_L$). *Here, though the condition expression for $\eta_L$ relies on a dynamic sampling probability $p_l^t$, we can still guarantee that there a constant $\eta_L$ satisfies this condition.*

*Specifically, one can substitute the optimal sampling probability $\frac{1}{p_i^t} = \frac{\sum_{j=1}^{m}\sqrt{\alpha_1\zeta_{G,j}^2 + \alpha_2\sigma_{L,j}^2}}{\sqrt{\alpha_1\zeta_{G,i}^2 + \alpha_2\sigma_{L,i}^2}}$ back to the above inequality condition. As long as the gradient $\nabla F_i(x_t)$ is bounded, we can ensure $\frac{1}{m^2}\sum_{i=1}^{m}\frac{\sum_{j=1}^{m}\sqrt{\alpha_1\zeta_{G,j}^2 + \alpha_2\sigma_{L,j}^2}}{\sqrt{\alpha_1\zeta_{G,i}^2 + \alpha_2\sigma_{L,i}^2}} \le \frac{\max_j \sqrt{\alpha_1\zeta_{G,j}^2 + \alpha_2\sigma_{L,j}^2}}{\min_i \sqrt{\alpha_1\zeta_{G,i}^2 + \alpha_1\sigma_{L,i}^2}} \le \tilde{G}$, therefore $\frac{1}{2\sqrt{10(A^2+1)}KL\sqrt{\frac{1}{m^2}\sum_{i=1}^{m}\frac{\sum_{j=1}^{m}\sqrt{\alpha_1\zeta_{G,j}^2 + \alpha_2\sigma_{L,j}^2}}{\sqrt{\alpha_1\zeta_{G,i}^2 + \alpha_2\sigma_{L,i}^2}}}} \ge \frac{1}{2\sqrt{10(A^2+1)}KL\sqrt{\tilde{G}}} \ge C$, where $\tilde{G}$ and $C$ are positive constants. Thus, we can always find a constant $\eta_L$ to satisfy this inequality under dynamic sampling probability $p_i^t$.*

---

[4]In the Appendix, we abbreviate $\zeta_{G,i,t}$ to $\zeta_{G,i}$ for the sake of simplicity in notation, without any loss of generality.

**Corollary E.7** (Convergence rate of DELTA). *Suppose $\eta_L$ and $\eta$ are such that the conditions mentioned above are satisfied, $\eta_L = \mathcal{O}\left(\frac{1}{\sqrt{T}KL}\right)$ and $\eta = \mathcal{O}\left(\sqrt{Kn}\right)$. Then for sufficiently large T, the iterates of Theorem 3.5 satisfy:*

$$\min_{t\in[T]} \mathbb{E}\|\nabla f(x_t)\|^2 \leq \mathcal{O}\left(\frac{F}{\sqrt{nKT}}\right) + \mathcal{O}\left(\frac{\sigma_L^2}{\sqrt{nKT}}\right) + \mathcal{O}\left(\frac{\sigma_L^2 + 4K\zeta_G^2}{KT}\right). \tag{50}$$

**Lemma E.8.** *For any step-size satisfying $\eta_L \leq \frac{1}{8LK}$, we can have the following results:*

$$\mathbb{E}\|x_{i,k}^t - x_t\|^2 \leq 5K(\eta_L^2\sigma_L^2 + 4K\eta_L^2\zeta_{G,i}^2) + 20K^2(A^2+1)\eta_L^2\|\nabla f(x_t)\|^2. \tag{51}$$

*where $\zeta_{G,i} = \|\nabla F(x_t) - \nabla f(x_t)\|$, and the expectation is over local SGD and filtration of $x_t$, without the stochasticity of client sampling.*

*Proof.*

$$\begin{aligned}
&\mathbb{E}_t\|x_{t,k}^i - x_t\|^2 \\
&= \mathbb{E}_t\|x_{t,k-1}^i - x_t - \eta_L g_{t,k-1}^t\|^2 \\
&= \mathbb{E}_t\|x_{t,k-1}^i - x_t - \eta_L(g_{t,k-1}^t - \nabla F_i(x_{t,k-1}^i) + \nabla F_i(x_{t,k-1}^i) - \nabla F_i(x_t) + \nabla F_i(x_t))\|^2 \\
&\leq (1 + \frac{1}{2K-1})\mathbb{E}_t\|x_{t,k-1}^i - x_t\|^2 + \mathbb{E}_t\|\eta_L(g_{t,k-1}^t - \nabla F_i(x_{t,k}^i))\|^2 \\
&\quad + 4K\mathbb{E}_t[\|\eta_L(\nabla F_i(x_{t,K-1}^i) - \nabla F_i(x_t))\|^2] + 4K\eta_L^2\mathbb{E}_t\|\nabla F_i(x_t)\|^2 \\
&\leq (1 + \frac{1}{2K-1})\mathbb{E}_t\|x_{t,k-1}^i - x_t\|^2 + \eta_L^2\sigma_L^2 + 4K\eta_L^2L^2\mathbb{E}_t\|x_{t,k-1}^i - x_t\|^2 \\
&\quad + 4K\eta_L^2\zeta_{G,i}^2 + 4K\eta_L^2(A^2+1)\|\nabla f(x_t)\|^2 \\
&\leq (1 + \frac{1}{K-1})\mathbb{E}\|x_{t,k-1}^i - x_t\|^2 + \eta_L^2\sigma_L^2 + 4K\eta_L^2\zeta_{G,i}^2 + 4K(A^2+1)\|\eta_L\nabla f(x_t)\|^2. \tag{52}
\end{aligned}$$

Unrolling the recursion, we get:

$$\begin{aligned}
\mathbb{E}_t\|x_{t,k}^i - x_t\|^2 &\leq \sum_{p=0}^{k-1}(1 + \frac{1}{K-1})^p\left[\eta_L^2\sigma_L^2 + 4K\eta_L^2\zeta_{G,i}^2 + 4K(A^2+1)\|\eta_L\nabla f(x_t)\|^2\right] \\
&\leq (K-1)\left[(1 + \frac{1}{K-1})^K - 1\right]\left[\eta_L^2\sigma_L^2 + 4K\eta_L^2\zeta_{G,i}^2 + 4K(A^2+1)\|\eta_L\nabla f(x_t)\|^2\right] \\
&\leq 5K(\eta_L^2\sigma_L^2 + 4K\eta_L^2\zeta_{G,i}^2) + 20K^2(A^2+1)\eta_L^2\|\nabla f(x_t)\|^2. \tag{53}
\end{aligned}$$

$\square$

## E.2 Proof for Theorem E.2.

In Section E.2.1 and Section E.2.2, we provide the proof for Theorem E.2. Specifically, the proof for sampling with replacement is shown in Appendix E.2.1, while the proof for sampling without replacement is shown in Appendix E.2.2.

### E.2.1 Sample with replacement

$$\min_{t\in[T]} \mathbb{E}\|\nabla\tilde{f}(x_t)\|^2 \leq \frac{f_0 - f_*}{c\eta\eta_L KT} + \frac{1}{c}\tilde{\Phi}, \tag{54}$$

where $\tilde{\Phi} = \frac{5L^2K\eta_L^2}{2m^2}\sum_{l=1}^m \frac{1}{p_l^t}(\sigma_L^2 + 4K\zeta_{G,i}^2) + \frac{L\eta_L\eta}{2n}\sum_{l=1}^m \frac{1}{m^2 p_l^t}\sigma_L^2$.

*Proof.*

$$\mathbb{E}_t[\tilde{f}(x_{t+1})] \overset{(a1)}{\leq} \tilde{f}(x_t) + \left\langle \nabla \tilde{f}(x_t), \mathbb{E}_t[x_{t+1} - x_t] \right\rangle + \frac{L}{2} \mathbb{E}_t[\|x_{t+1} - x_t\|^2]$$

$$= \tilde{f}(x_t) + \left\langle \nabla \tilde{f}(x_t), \mathbb{E}_t[\eta \Delta_t + \eta \eta_L K \nabla \tilde{f}(x_t) - \eta \eta_L K \nabla \tilde{f}(x_t)] \right\rangle + \frac{L}{2} \eta^2 \mathbb{E}_t[\|\Delta_t\|^2]$$

$$= \tilde{f}(x_t) - \eta \eta_L K \left\| \nabla \tilde{f}(x_t) \right\|^2 + \eta \underbrace{\left\langle \nabla \tilde{f}(x_t), \mathbb{E}_t[\Delta_t + \eta_L K \nabla \tilde{f}(x_t)] \right\rangle}_{A_1} + \frac{L}{2} \eta^2 \underbrace{\mathbb{E}_t \|\Delta_t\|^2}_{A_2}.$$

$$(55)$$

Where (a1) follows from the Lipschitz continuity condition. Here, the expectation is over the local data SGD and the filtration of $x_t$. However, in the next analysis, the expectation is over all randomness, including client sampling .This is achieved by taking expectation on both sides of the above equation over client sampling.

To begin, let us consider $A_1$:

$$A_1 = \left\langle \nabla \tilde{f}(x_t), \mathbb{E}_t[\Delta_t + \eta_L K \nabla \tilde{f}(x_t)] \right\rangle$$

$$= \left\langle \nabla \tilde{f}(x_t), \mathbb{E}_t[-\frac{1}{|S_t|} \sum_{i \in S_t} \frac{1}{m p_i^t} \sum_{k=0}^{K-1} \eta_L g_{t,k}^i + \eta_L K \nabla \tilde{f}(x_t)] \right\rangle$$

$$\overset{(a2)}{=} \left\langle \nabla \tilde{f}(x_t), \mathbb{E}_t[-\frac{1}{|S_t|} \sum_{i \in S_t} \frac{1}{m p_i^t} \sum_{k=0}^{K-1} \eta_L \nabla F_i(x_{t,k}^i) + \eta_L K \nabla \tilde{f}(x_t)] \right\rangle$$

$$= \left\langle \sqrt{K \eta_L} \nabla \tilde{f}(x_t), \frac{\sqrt{\eta_L}}{\sqrt{K}} \mathbb{E}_t[-\frac{1}{n} \sum_{i \in S_t} \frac{1}{m p_i^t} \sum_{k=0}^{K-1} \nabla F_i(x_{t,k}^i) + K \nabla \tilde{f}(x_t)] \right\rangle$$

$$\overset{(a3)}{=} \frac{K \eta_L}{2} \|\nabla \tilde{f}(x_t)\|^2 + \frac{\eta_L}{2K} \mathbb{E}_t \left( \| -\frac{1}{n} \sum_{i \in S_t} \frac{1}{m p_i^t} \sum_{k=0}^{K-1} \nabla F_i(x_{t,k}^i) + K \nabla \tilde{f}(x_t) \|^2 \right)$$

$$- \frac{\eta_L}{2K} \mathbb{E}_t \| -\frac{1}{n} \sum_{i \in S_t} \frac{1}{m p_i^t} \sum_{k=0}^{K-1} \nabla F_i(x_{t,k}^i) \|^2, \qquad (56)$$

where (a2) follows from Assumption 2, and (a3) is due to $\langle x, y \rangle = \frac{1}{2} \left[ \|x\|^2 + \|y\|^2 - \|x - y\|^2 \right]$ for $x = \sqrt{K \eta_L} \nabla \tilde{f}(x_t)$ and $y = \frac{\sqrt{\eta_L}}{K} [-\frac{1}{n} \sum_{i \in S_t} \frac{1}{m p_i^t} \sum_{k=0}^{K-1} \nabla F_i(x_{t,k}^i) + K \nabla \tilde{f}(x_t)]$.

To bound $A_1$, we need to bound the following part:

$$\mathbb{E}_t\|\frac{1}{n}\sum_{i\in S_t}\frac{1}{mp_i^t}\sum_{k=0}^{K-1}\nabla F_i(x_{t,k}^i) - K\nabla\tilde{f}(x_t)\|^2$$

$$= \mathbb{E}_t\|\frac{1}{n}\sum_{i\in S_t}\frac{1}{mp_i^t}\sum_{k=0}^{K-1}\nabla F_i(x_{t,k}^i) - \frac{1}{n}\sum_{i\in S_t}\frac{1}{mp_i^t}\sum_{k=0}^{K-1}\nabla F_i(x_t)\|^2$$

$$\overset{(a4)}{\leq} \frac{K}{n}\sum_{i\in S_t}\sum_{k=0}^{K-1}\mathbb{E}_t\|\frac{1}{mp_i^t}(\nabla F_i(x_{t,k}^i) - \nabla F_i(x_t))\|^2$$

$$= \frac{K}{n}\sum_{i\in S_t}\sum_{k=0}^{K-1}\mathbb{E}_t\{\mathbb{E}_t(\|\frac{1}{mp_i^t}(\nabla F_i(x_{t,k}^i) - \nabla F_i(x_t))\|^2 \mid S)\}$$

$$= \frac{K}{n}\sum_{i\in S_t}\sum_{k=0}^{K-1}\mathbb{E}_t(\sum_{l=1}^{m}\frac{1}{m^2p_l^t}\|\nabla F_l(x_{t,k}^l) - \nabla F_l(x_t)\|^2)$$

$$= K\sum_{k=0}^{K-1}\sum_{l=1}^{m}\frac{1}{m^2p_l^t}\mathbb{E}_t\|\nabla F_l(x_{t,k}^l) - \nabla F_l(x_t)\|^2$$

$$\overset{(a5)}{\leq} \frac{K^2}{m^2}\sum_{l=1}^{m}\frac{L^2}{p_l^t}\mathbb{E}\|x_{t,k}^l - x_t\|^2$$

$$\overset{(a6)}{\leq} \frac{L^2K^2}{m^2}\sum_{l=1}^{m}\frac{1}{p_l^t}\left(5K(\eta_L^2\sigma_L^2 + 4K\eta_L^2\zeta_{G,i}^2) + 20K^2(A^2+1)\eta_L^2\|\nabla f(x_t)\|^2\right)$$

$$= \frac{5L^2K^3\eta_L^2}{m^2}\sum_{l=1}^{m}\frac{1}{p_l^t}(\sigma_L^2 + 4K\sigma_G^2) + \frac{20L^2K^4\eta_L^2(A^2+1)}{m^2}\sum_{l=1}^{m}\frac{1}{p_l^t}\|\nabla f(x_t)\|^2, \qquad (57)$$

where (a4) follows from the fact that $\mathbb{E}|x_1 + \cdots + x_n|^2 \leq n\mathbb{E}(|x_1|^2 + \cdots + |x_n|^2)$, (a5) is a consequence of Assumption 1, and (a6) is a result of Lemma E.8.

Combining the above expressions, we obtain:

$$A_1 \leq \frac{K\eta_L}{2}\|\nabla\tilde{f}(x_t)\|^2 + \frac{\eta_L}{2K}\left[\frac{5L^2K^3\eta_L^2}{m^2}\sum_{l=1}^{m}\frac{1}{p_l^t}(\sigma_L + 4K\zeta_{G,i}^2)\right.$$

$$\left.+\frac{20L^2K^4\eta_L^2(A^2+1)}{m^2}\sum_{l=1}^{m}\frac{1}{p_l^t}\|\nabla f(x_t)\|^2\right] - \frac{\eta_L}{2K}\mathbb{E}_t\| - \frac{1}{n}\sum_{i\in S_t}\frac{1}{mp_i^t}\sum_{k=0}^{K-1}\nabla F_i(x_{t,k}^i)\|^2. \quad (58)$$

Next, we consider bounding $A_2$:

$$A_2 = \mathbb{E}_t \|\Delta_t\|^2$$

$$= \mathbb{E}_t \left\| -\eta_L \frac{1}{n} \sum_{i \in S_t} \frac{1}{mp_i^t} \sum_{k=0}^{K-1} g_{t,k}^i \right\|^2$$

$$= \eta_L^2 \mathbb{E}_t \left\| \frac{1}{n} \sum_{i \in S_t} \sum_{k=0}^{K-1} \left( \frac{1}{mp_i^t} g_{t,k}^i - \frac{1}{mp_i^t} \nabla F_i(x_{t,k}^i) \right) \right\|^2 + \eta_L^2 \mathbb{E}_t \left\| -\frac{1}{n} \sum_{i \in S_t} \frac{1}{mp_i^t} \sum_{k=0}^{K-1} \nabla F_i(x_{t,k}^i) \right\|^2$$

$$= \eta_L^2 \frac{1}{n^2} \sum_{i \in S_t} \sum_{k=0}^{K-1} \mathbb{E}_t \left\| \frac{1}{mp_i^t} g_{t,k}^i - \frac{1}{mp_i^t} \nabla F_i(x_{t,k}^i) \right\|^2 + \eta_L^2 \mathbb{E}_t \left\| -\frac{1}{n} \sum_{i \in S_t} \frac{1}{mp_i^t} \sum_{k=0}^{K-1} \nabla F_i(x_{t,k}^i) \right\|^2$$

$$= \eta_L^2 \frac{1}{n^2} \sum_{k=0}^{K-1} \mathbb{E}_t \left( \mathbb{E} \left\| \frac{1}{mp_i^t} (g_{t,k}^i - \nabla F_i(x_{t,k}^i)) \right\|^2 \mid S \right) + \eta_L^2 \mathbb{E}_t \left\| -\frac{1}{n} \sum_{i \in S_t} \frac{1}{mp_i^t} \sum_{k=0}^{K-1} \nabla F_i(x_{t,k}^i) \right\|^2$$

$$= \eta_L^2 \frac{1}{n^2} \sum_{k=0}^{K-1} \mathbb{E}_t \left( \sum_{l=1}^m \frac{1}{m^2 p_l^t} \left\| g_{t,k}^i - \nabla F_i(x_{t,k}^i) \right\|^2 \right) + \eta_L^2 \mathbb{E}_t \left\| -\frac{1}{n} \sum_{i \in S_t} \frac{1}{mp_i^t} \sum_{k=0}^{K-1} \nabla F_i(x_{t,k}^i) \right\|^2$$

$$\overset{(a7)}{\leq} \eta_L^2 \frac{K}{n} \sum_{l=1}^m \frac{1}{m^2 p_l^t} \sigma_L^2 + \eta_L^2 \mathbb{E}_t \left\| -\frac{1}{n} \sum_{i \in S_t} \frac{1}{mp_i^t} \sum_{k=0}^{K-1} \nabla F_i(x_{t,k}^i) \right\|^2, \tag{59}$$

where $S$ represents the whole sample space and (a7) is due to Assumption 2.

Now we substitute the expressions for $A_1$ and $A_2$ and take the expectation over the client sampling distribution on both sides. It should be noted that the derivation of $A_1$ and $A_2$ above is based on considering the expectation over the sampling distribution:

$$f(x_{t+1}) \leq f(x_t) - \eta \eta_L K \mathbb{E}_t \left\| \nabla \tilde{f}(x_t) \right\|^2 + \eta \mathbb{E}_t \left\langle \nabla \tilde{f}(x_t), \Delta_t + \eta_L K \nabla \tilde{f}(x_t) \right\rangle + \frac{L}{2} \eta^2 \mathbb{E}_t \|\Delta_t\|^2$$

$$\overset{(a8)}{\leq} f(x_t) - K \eta \eta_L \left( \frac{1}{2} - \frac{20 K^2 \eta_L^2 L^2 (A^2+1)}{m^2} \sum_{l=1}^m \frac{1}{p_l^t} \right) \mathbb{E}_t \left\| \nabla \tilde{f}(x_t) \right\|^2$$

$$+ \frac{5L^2 K^2 \eta_L^3 \eta}{2m^2} \sum_{l=1}^m \frac{1}{p_l^t} \left( \sigma_L + 4K \zeta_{G,i}^2 \right)$$

$$+ \frac{L \eta_L^2 \eta^2 K}{2n} \sum_{l=1}^m \frac{1}{m^2 p_l^t} \sigma_L^2 - \left( \frac{\eta \eta_L}{2K} - \frac{L \eta^2 \eta_L^2}{2} \right) \mathbb{E}_t \left\| -\frac{1}{n} \sum_{i \in S_t} \frac{1}{mp_i^t} \sum_{k=0}^{K-1} \nabla f_i(x_{t,k}^i) \right\|^2$$

$$\overset{(a9)}{\leq} f(x_t) - K \eta \eta_L \left( \frac{1}{2} - \frac{20 K^2 \eta_L^2 L^2 (A^2+1)}{m^2} \sum_{l=1}^m \frac{1}{p_l^t} \right) \mathbb{E}_t \| \nabla \tilde{f}(x_t) \|^2$$

$$+ \frac{5L^2 K^2 \eta_L^3 \eta}{2m^2} \sum_{l=1}^m \frac{1}{p_l^t} (\sigma_L + 4K \zeta_{G,i}^2) + \frac{L \eta_L^2 \eta^2 K}{2n} \sum_{l=1}^m \frac{1}{m^2 p_l^t} \sigma_L^2$$

$$\overset{(a10)}{\leq} f(x_t) - cK \eta \eta_L \mathbb{E}_t \| \nabla \tilde{f}(x_t) \|^2 + \frac{5L^2 K^2 \eta_L^3 \eta}{2m^2} \sum_{l=1}^m \frac{1}{p_l^t} (\sigma_L^2 + 4K \zeta_{G,i}^2) + \frac{L \eta_L^2 \eta^2 K}{2n} \sum_{l=1}^m \frac{1}{m^2 p_l^t} \sigma_L^2, \tag{60}$$

where (a8) comes from Lemma C.2, (a9) follows from $\left( \frac{\eta \eta_L}{2K} - \frac{L \eta^2 \eta_L^2}{2} \right) \geq 0$ if $\eta \eta_l \leq \frac{1}{KL}$, and (a10) holds because there exists a constant $c > 0$ satisfying $\left( \frac{1}{2} - \frac{20 K^2 \eta_L^2 L^2 (A^2+1)}{m^2} \sum_{l=1}^m \frac{1}{p_l^t} \right) > c > 0$ if $\eta_L < \frac{1}{2\sqrt{10(A^2+1)} KL \sqrt{\frac{1}{m} \sum_{l=1}^m \frac{1}{mp_l^t}}}$.

Rearranging and summing from $t = 0, \ldots, T-1$, we have:

$$\sum_{t=1}^{T-1} c\eta\eta_L K \mathbb{E}\|\nabla \tilde{f}(x_t)\|^2 \leq f(x_0) - f(x_T)$$

$$+ T(\eta\eta_L K)\left(\frac{5L^2 K\eta_L^2}{2m^2}\sum_{l=1}^{m}\frac{1}{p_l^t}(\sigma_L^2 + 4K\zeta_{G,i}^2) + \frac{L\eta_L\eta}{2n}\sum_{l=1}^{m}\frac{1}{m^2 p_l^t}\sigma_L^2\right). \quad (61)$$

Which implies:

$$\min_{t\in[T]} \mathbb{E}\|\nabla \tilde{f}(x_t)\|^2 \leq \frac{f_0 - f_*}{c\eta\eta_L KT} + \frac{1}{c}\tilde{\Phi}, \quad (62)$$

where $\tilde{\Phi} = \frac{5L^2 K\eta_L^2}{2m^2}\sum_{l=1}^{m}\frac{1}{p_l^t}(\sigma_L^2 + 4K\zeta_{G,i}^2) + \frac{L\eta_L\eta}{2n}\sum_{l=1}^{m}\frac{1}{m^2 p_l^t}\sigma_L^2$.

$\square$

### E.2.2 Sample without replacement

$$\min_{t\in[T]} \mathbb{E}\|\nabla \tilde{f}(x_t)\|^2 \leq \frac{f_0 - f_*}{c\eta\eta_L KT} + \frac{1}{c}\tilde{\Phi}, \quad (63)$$

where $\tilde{\Phi} = \frac{5L^2 K\eta_L^2}{2mn}\sum_{l=1}^{m}\frac{1}{p_l^t}(\sigma_L^2 + 4K\zeta_{G,i}^2) + \frac{L\eta_L\eta}{2n}\sum_{l=1}^{m}\frac{1}{m^2 p_l^t}\sigma_L^2$.

*Proof.*

$$\mathbb{E}[\tilde{f}(x_{t+1})] \leq \tilde{f}(x_t) + \left\langle \nabla \tilde{f}(x_t), \mathbb{E}[x_{t+1} - x_t]\right\rangle + \frac{L}{2}\mathbb{E}_t[\|x_{t+1} - x_t\|]$$

$$= \tilde{f}(x_t) + \left\langle \nabla \tilde{f}(x_t), \mathbb{E}_t[\eta\Delta_t + \eta\eta_L K\nabla\tilde{f}(x_t) - \eta\eta_L K\nabla\tilde{f}(x_t)]\right\rangle + \frac{L}{2}\eta^2 \mathbb{E}_t[\|\Delta_t\|^2]$$

$$= \tilde{f}(x_t) - \eta\eta_L K\left\|\nabla\tilde{f}(x_t)\right\|^2 + \eta\underbrace{\left\langle \nabla\tilde{f}(x_t), \mathbb{E}_t[\Delta_t + \eta_L K\nabla\tilde{f}(x_t)]\right\rangle}_{A_1} + \frac{L}{2}\eta^2\underbrace{\mathbb{E}_t\|\Delta_t\|^2}_{A_2}. \quad (64)$$

Where the first inequality follows from Lipschitz continuous condition. The expectation here is taken over both the local SGD and the filtration of $x_t$. However, in the subsequent analysis, the expectation is taken over all sources of randomness, including client sampling.

Similarly, we consider $A_1$ first:

$$A_1 = \left\langle \nabla\tilde{f}(x_t), \mathbb{E}_t[\Delta_t + \eta_L K\nabla\tilde{f}(x_t)]\right\rangle$$

$$= \left\langle \nabla\tilde{f}(x_t), \mathbb{E}_t\left[-\frac{1}{|S_t|}\sum_{i\in S_t}\frac{1}{mp_i^t}\sum_{k=0}^{K-1}\eta_L g_{t,k}^i + \eta_L K\nabla\tilde{f}(x_t)\right]\right\rangle$$

$$= \left\langle \nabla\tilde{f}(x_t), \mathbb{E}_t\left[-\frac{1}{|S_t|}\sum_{i\in S_t}\frac{1}{mp_i^t}\sum_{k=0}^{K-1}\eta_L \nabla F_i(x_{t,k}^i) + \eta_L K\nabla\tilde{f}(x_t)\right]\right\rangle$$

$$= \left\langle \sqrt{K\eta_L}\nabla\tilde{f}(x_t), \frac{\sqrt{\eta_L}}{\sqrt{K}}\mathbb{E}_t\left[-\frac{1}{n}\sum_{i\in S_t}\frac{1}{mp_i^t}\sum_{k=0}^{K-1}\nabla F_i(x_{t,k}^i) + K\nabla\tilde{f}(x_t)\right]\right\rangle$$

$$= \frac{K\eta_L}{2}\left\|\nabla\tilde{f}(x_t)\right\|^2 + \frac{\eta_L}{2K}\mathbb{E}_t\left\|-\frac{1}{n}\sum_{i\in S_t}\frac{1}{mp_i^t}\sum_{k=0}^{K-1}\nabla F_i(x_{t,k}^i) + K\nabla\tilde{f}(x_t)\right\|^2$$

$$- \frac{\eta_L}{2K}\mathbb{E}_t\left\|-\frac{1}{n}\sum_{i\in S_t}\frac{1}{mp_i^t}\sum_{k=0}^{K-1}\nabla F_i(x_{t,k}^i)\right\|^2. \quad (65)$$

Since $x_i$ are sampled from $S_t$ without replacement, this causes pairs $x_{i1}$ and $x_{i2}$ to no longer be independent. We introduce the activation function as follows:

$$\mathbb{I}_m \triangleq \begin{cases} 1 & if \ x \in S_t \,, \\ 0 & \text{otherwise} \,. \end{cases} \tag{66}$$

Then we obtain the following bound:

$$\mathbb{E}_t \left\| \frac{1}{n} \sum_{i \in S_t} \frac{1}{mp_i^t} \sum_{k=0}^{K-1} \nabla F_i(x_{t,k}^i) - K\nabla \tilde{f}(x_t) \right\|^2$$

$$= \mathbb{E}_t \left\| \frac{1}{n} \sum_{l=1}^{m} \mathbb{I}_m \frac{1}{mp_l^t} \sum_{k=0}^{K-1} \nabla F_l(x_{t,k}^l) - \frac{1}{n} \sum_{l=1}^{m} \mathbb{I}_m \frac{1}{mp_l^t} \sum_{k=0}^{K-1} \nabla F_l(x_t) \right\|^2$$

$$\overset{(b1)}{\leq} \frac{m}{n^2} \sum_{l=1}^{m} \mathbb{E}_t \left\| \mathbb{I}_m \frac{1}{mp_l^t} \sum_{k=0}^{K-1} \left( \nabla F_l(x_{t,k}^l) - \nabla F_l(x_t) \right) \right\|^2$$

$$- \frac{1}{n^2} \sum_{l_1 \neq l_2} \mathbb{E}_t \left\| \left\{ \mathbb{I}_m \frac{1}{mp_{l_1}} \sum_{k=0}^{K-1} \left( \nabla F_{l_1}(x_{t,k}^{l_1}) - \nabla F_{l_1}(x_t) \right) \right\} \right.$$

$$\left. - \left\{ \mathbb{I}_m \frac{1}{mp_{l_2}} \sum_{k=0}^{K-1} \left( \nabla F_{l_2}(x_{t,k}^{l_2}) - \nabla F_{l_2}(x_t) \right) \right\} \right\|^2$$

$$\leq \frac{m}{n^2} \sum_{l=1}^{m} \mathbb{E}_t \left\| \mathbb{I}_m \frac{1}{mp_l^t} \sum_{k=0}^{K-1} \left( \nabla F_l(x_{t,k}^l) - \frac{1}{mp_l^t} \nabla F_l(x_t) \right) \right\|^2$$

$$= \frac{m}{n^2} \sum_{l=1}^{m} \mathbb{E}_t \left\{ \left\| \mathbb{I}_m \frac{1}{mp_l^t} \sum_{k=0}^{K-1} \left( \nabla F_l(x_{t,k}^l) - \frac{1}{mp_l^t} \nabla F_l(x_t) \right) \right\|^2 \mid \mathbb{I}_m = 1 \right\} \times P(\mathbb{I}_m = 1)$$

$$+ \mathbb{E}_t \left\{ \left\| \mathbb{I}_m (\frac{1}{mp_l^t} \sum_{k=0}^{K-1} \nabla F_l(x_{t,k}^l) - \frac{1}{mp_l^t} \nabla F_l(x_t) \right\|^2 \mid \mathbb{I}_m = 0 \right\} \times P(\mathbb{I}_m = 0))$$

$$= \frac{m}{n^2} \sum_{l=1}^{m} np_l^t \mathbb{E} \left\| \frac{1}{mp_l^t} \sum_{k=0}^{K-1} \nabla F_l(x_{t,k}^l) - \frac{1}{mp_l^t} \sum_{k=0}^{K-1} \nabla F_l(x_t) \right\|^2$$

$$\overset{(b2)}{\leq} \frac{L^2 K}{mn} \sum_{k=0}^{K-1} \sum_{l=1}^{m} \frac{1}{p_l^t} \mathbb{E} \| x_{t,k}^l - x_t \|^2$$

$$\overset{(b3)}{\leq} \frac{L^2 K^2}{n} \left( 5K \frac{\eta_L^2}{m} \sum_{l=1}^{m} \frac{1}{p_l^t} (\sigma_L^2 + 4K\zeta_{G,i}^2) + 20K^2(A^2+1)\eta_L^2 \|\nabla f(x_t)\|^2 \frac{1}{m} \sum_{l=1}^{m} \frac{1}{p_l^t} \right), \tag{67}$$

where (b1) follows from $\| \sum_{i=1}^{m} t_i \|^2 = \sum_{i \in [m]} \|t_i\|^2 + \sum_{i \neq j} \langle t_i, t_j \rangle \overset{c1}{=} \sum_{i \in [m]} m\|t_i\|^2 - \frac{1}{2} \sum_{i \neq j} \|t_i - t_j\|^2$ ((c1) here is due to $\langle x, y \rangle = \frac{1}{2} \left[ \|x\|^2 + \|y\|^2 - \|x - y\|^2 \right]$), (b2) is due to $\mathbb{E}\|x_1 + \cdots + x_n\|^2 \leq n\mathbb{E} \left( \|x_1\|^2 + \cdots + \|x_n\|^2 \right)$, and (b3) comes from Lemma E.8.

Therefore, we have the bound of $A_1$:

$$A_1 \leq \frac{K\eta_L}{2} \|\nabla \tilde{f}(x_t)\|^2 + \frac{\eta_L L^2 K}{2n} \left( 5K \frac{\eta_L^2}{m} \sum_{l=1}^{m} \frac{1}{p_l^t} (\sigma_L^2 + 4K\zeta_{G,i}^2) \right.$$

$$\left. + 20K^2(A^2+1)\eta_L^2 \|\nabla f(x_t)\|^2 \frac{1}{m} \sum_{l=1}^{m} \frac{1}{p_l^t} \right) - \frac{\eta_L}{2K} \mathbb{E}_t \left\| -\frac{1}{n} \sum_{i \in S_t} \frac{1}{mp_i^t} \sum_{k=0}^{K-1} \nabla F_i(x_{t,k}^i) \right\|^2. \tag{68}$$

The expression for $A_2$ is as follows:

$$A_2 = \mathbb{E}_t \|\Delta_t\|^2$$

$$= \mathbb{E}_t \left\| -\eta_L \frac{1}{n} \sum_{i \in S_t} \frac{1}{m p_i^t} \sum_{k=0}^{K-1} g_{t,k}^i \right\|^2$$

$$= \eta_L^2 \mathbb{E}_t \left\| \frac{1}{n} \sum_{i \in S_t} \sum_{k=0}^{K-1} \left( \frac{1}{m p_i^t} g_{t,k}^i - \frac{1}{m p_i^t} \nabla F_i(x_{t,k}^i) \right) \right\|^2 + \eta_L^2 \mathbb{E}_t \left\| -\frac{1}{n} \sum_{i \in S_t} \frac{1}{m p_i^t} \sum_{k=0}^{K-1} \nabla F_i(x_{t,k}^i) \right\|^2$$

$$= \eta_L^2 \frac{1}{n^2} \mathbb{E}_t \left\| \sum_{l=1}^{m} \mathbb{I}_m \sum_{k=0}^{K-1} \frac{1}{m p_l^t} (g_{t,k}^l - \nabla F_i(x_{t,k}^i)) \right\|^2 + \eta_L^2 \mathbb{E}_t \left\| -\frac{1}{n} \sum_{i \in S_t} \frac{1}{m p_i^t} \sum_{k=0}^{K-1} \nabla F_i(x_{t,k}^i) \right\|^2$$

$$= \eta_L^2 \frac{1}{n^2} \sum_{l=1}^{m} \mathbb{E}_t \left\| \sum_{l=1}^{m} \mathbb{I}_m \sum_{k=0}^{K-1} \frac{1}{m p_l^t} (g_{t,k}^l - \nabla F_i(x_{t,k}^i)) \right\|^2 + \eta_L^2 \mathbb{E}_t \left\| -\frac{1}{n} \sum_{i \in S_t} \frac{1}{m p_i^t} \sum_{k=0}^{K-1} \nabla F_i(x_{t,k}^i) \right\|^2$$

$$= \eta_L^2 \frac{1}{n^2} \sum_{l=1}^{m} n p_l^t \mathbb{E}_t \left\| \sum_{k=0}^{K-1} \frac{1}{m p_l^t} (g_{t,k}^l - \nabla F_i(x_{t,k}^i)) \right\|^2 + \eta_L^2 \mathbb{E}_t \left\| -\frac{1}{n} \sum_{i \in S_t} \frac{1}{m p_i^t} \sum_{k=0}^{K-1} \nabla F_i(x_{t,k}^i) \right\|^2$$

$$\leq \eta_L^2 \frac{K}{n} \sum_{l=1}^{m} \frac{1}{m^2 p_l^t} \sigma_L^2 + \eta_L^2 \mathbb{E}_t \left\| -\frac{1}{n} \sum_{i \in S_t} \frac{1}{m p_i^t} \sum_{k=0}^{K-1} \nabla F_i(x_{t,k}^i) \right\|^2 . \tag{69}$$

Now we substitute the expressions for $A_1$ and $A_2$ and take the expectation over the client sampling distribution on both sides. It should be noted that the derivation of $A_1$ and $A_2$ above is based on considering the expectation over the sampling distribution:

$$f(x_{t+1}) \leq f(x_t) - \eta \eta_L K \mathbb{E}_t \left\| \nabla \tilde{f}(x_t) \right\|^2 + \eta \mathbb{E}_t \left\langle \nabla \tilde{f}(x_t), \Delta_t + \eta_L K \nabla \tilde{f}(x_t) \right\rangle + \frac{L}{2} \eta^2 \mathbb{E}_t \|\Delta_t\|^2$$

$$\overset{(b4)}{\leq} f(x_t) - \eta \eta_L K \left( \frac{1}{2} - \frac{20 L^2 K^2 (A^2+1) \eta_L^2}{nm} \sum_{l=1}^{m} \frac{1}{p_l^t} \right) \mathbb{E}_t \|\nabla \tilde{f}(x_t)\|^2 + \frac{2K^2 \eta \eta_L^3 L^2}{2nm} \sum_{l=1}^{m} \frac{1}{p_l^t} \big( \sigma_L^2$$

$$+ 4K \zeta_{G,i}^2 \big) + \frac{L \eta^2 \eta_L^2 K}{2n} \sum_{l=1}^{m} \frac{1}{p_l^t} \sigma_L^2 - \left( \frac{\eta \eta_L}{2K} - \frac{L \eta^2 \eta_L^2}{2} \right) \mathbb{E}_t \left\| -\frac{1}{n} \sum_{i \in S_t} \frac{1}{m p_i^t} \sum_{k=0}^{K-1} \nabla F_i(x_{t,k}^i) \right\|^2$$

$$\leq f(x_t) - c \eta \eta_L K \mathbb{E}_t \|\nabla \tilde{f}(x_t)\|^2 + \frac{2K^2 \eta \eta_L^3 L^2}{2nm} \sum_{l=1}^{m} \frac{1}{p_l^t} (\sigma_L^2 + 4K \zeta_{G,i}^2) + \frac{L \eta^2 \eta_L^2 K}{2n} \sum_{l=1}^{m} \frac{1}{p_l^t} \sigma_L^2 . \tag{70}$$

Also, for (b4), step sizes need to satisfy $\left( \frac{\eta \eta_L}{2K} - \frac{L \eta^2 \eta_L^2}{2} \right) \geq 0$ if $\eta \eta_l \leq \frac{1}{KL}$, and there exists a constant $c > 0$ satisfying $\left( \frac{1}{2} - \frac{20 K^2 \eta_L^2 L^2 (A^2+1)}{mn} \sum_{l=1}^{m} \frac{1}{p_l^t} \right) > c > 0$ if $\eta_L < \frac{1}{2 \sqrt{10(A^2+1)} KL \sqrt{\frac{1}{n} \sum_{l=1}^{m} \frac{1}{m p_l^t}}}$.

Rearranging and summing from $t = 0, \ldots, T - 1$, we have:

$$\sum_{t=1}^{T-1} c \eta \eta_L K \mathbb{E} \|\nabla \tilde{f}(x_t)\|^2 \leq f(x_0) - f(x_T) + T(\eta \eta_L K) \tilde{\Phi} . \tag{71}$$

Which implies:

$$\min_{t \in [T]} \mathbb{E} \|\nabla \tilde{f}(x_t)\|^2 \leq \frac{f_0 - f_*}{c \eta \eta_L KT} + \frac{1}{c} \tilde{\Phi} , \tag{72}$$

where $\tilde{\Phi} = \frac{5 L^2 K \eta_L^2}{2mn} \sum_{l=1}^{m} \frac{1}{p_l^t} (\sigma_L^2 + 4K \zeta_{G,i}^2) + \frac{L \eta_L \eta}{2n} \sum_{l=1}^{m} \frac{1}{m^2 p_l^t} \sigma_L^2$.

$$\square$$

**Remark E.9.** $\zeta_G$ *used in DELTA can be easily transformed into a* $\sigma_G$ *related term, thus it is fair to compare DELTA with FedIS. In particular, by taking the expectation on* $\zeta_G$*, it equates to* $E\|\nabla F_i(x_t) - \nabla f(x_t)\|^2$*. As demonstrated in [1], one can derive* $E\|\nabla F_i(x_t) - \nabla f(x_t)\|^2 = E\|\nabla F_i(x_t)\|^2 - \|\nabla f(x_t)\|^2 \le A\|\nabla f(x_t)\|^2 + \sigma_G^2$*. Shifting* $A\|\nabla f(x_t)\|^2$ *to the left side of the convergence result,* $\zeta_G$ *can be directly transformed into* $\sigma_G$*.*

# F   Proof of the Optimal Sampling Probability

## F.1   Sampling probability FedIS

**Corollary F.1** (Optimal sampling probability for FedIS)**.**

$$\min_{p_l^t} \Phi \qquad s.t. \sum_{l=1}^{m} p_l^t = 1 \,.$$

*Solving the above optimization problem, we obtain the expression for the optimal sampling probability:*

$$p_i^t = \frac{\|\hat{g}_i^t\|}{\sum_{j=1}^{m} \|\hat{g}_j^t\|} \,, \tag{73}$$

*where* $\hat{g}_i^t = \sum_{k=0}^{K-1} g_k^i$ *is the sum of the gradient updates across multiple updates.*

Recall Theorem 3.1; only the last variance term in the convergence term $\Phi$ is affected by sampling. In other words, we need to minimize the variance term with respect to probability. We formalize this as follows:

$$\min_{p_i^t \in [0,1], \sum_{i=1}^{m} p_i^t = 1} V\left(\frac{1}{mp_i^t} \hat{g}_i^t\right) \Leftrightarrow \min_{p_i^t \in [0,1], \sum_{i=1}^{m} p_i^t = 1} \frac{1}{m^2} \sum_{i=1}^{m} \frac{1}{p_i^t} \|\hat{g}_i^t\|^2 \,. \tag{74}$$

This optimization problem can be solved in closed form using the KKT conditions. It is straightforward to verify that the solution to the optimization problem is:

$$p_{i,t}^* = \frac{\| \sum_{k=0}^{K-1} g_{t,k}^i \|}{\sum_{i=1}^{m} \| \sum_{k=0}^{K-1} g_{t,k}^i \|}, \forall i \in 1, 2, ..., m \,. \tag{75}$$

Under the optimal sampling probability, the variance will be:

$$V\left(\frac{1}{mp_i^t} \hat{g}_i^t\right) \le \mathbb{E}\left\| \frac{\sum_{i=1}^{m} \hat{g}_i^t}{m} \right\|^2 = \frac{1}{m^2} \mathbb{E}\| \sum_{i=1}^{m} \sum_{k=1}^{K} \nabla F_i(x_{t,k}, \xi_{k,t}) \|^2 \tag{76}$$

Therefore, the variance term is bounded by:

$$V\left(\frac{1}{mp_i^t} \hat{g}_i^t\right) \le \frac{1}{m} \sum_{i=1}^{m} K \sum_{k=1}^{K} \mathbb{E}\|\nabla F_i(x_{t,k}, \xi_{k,t})\|^2 \le K^2 G^2 \tag{77}$$

**Remark:** If the uniform distribution is adopted with $p_i^t = \frac{1}{m}$, it is easy to observe that the variance of the stochastic gradient is bounded by $\frac{\sum_{i=1}^{m} |g_i|^2}{m}$.

According to Cauchy-Schwarz inequality,

$$\frac{\sum_{i=1}^{m} \|\hat{g}_i^t\|^2}{m} \Big/ \left(\frac{\sum_{i=1}^{m} \|\hat{g}_i\|}{m}\right)^2 = \frac{m \sum_{i=1}^{m} \|\hat{g}_i\|^2}{\left(\sum_{i=1}^{m} \|\hat{g}_i\|\right)^2} \ge 1 \,, \tag{78}$$

this implies that importance sampling does improve convergence rate, especially when $\frac{\left(\sum_{i=1}^{m} \|g_i\|\right)^2}{\sum_{i=1}^{m} \|g_i\|^2} << m$.

### F.2 Sampling probability of DELTA

Our result is of the following form:

$$\min_{t \in [T]} \mathbb{E}\|\nabla f(x_t)\|^2 \leq \frac{f_0 - f_*}{c \eta \eta_L K T} + \tilde{\Phi}, \tag{79}$$

It is easy to see that the sampling strategy only affects $\tilde{\Phi}$. To enhance the convergence rate, we need to minimize $\tilde{\Phi}$ with respect to $p_l^t$. As shown, the expression for $\tilde{\Phi}$ with and without replacement is similar, and only differs in the values of $n$ and $m$. Here, we will consider the case with replacement. Specifically, we need to solve the following optimization problem:

$$\min_{p_l^t} \tilde{\Phi} = \frac{1}{c}\left(\frac{5L^2 K \eta_L^2}{2m^2} \sum_{l=1}^{m} \frac{1}{p_l^t}(\sigma_{L,l}^2 + 4K\zeta_{G,i}^2) + \frac{L\eta_L \eta}{2n} \sum_{l=1}^{m} \frac{1}{m^2 p_l^t}\sigma_{L,i}^2\right) \qquad s.t. \sum_{l=1}^{m} p_l^t = 1.$$

Solving this optimization problem, we find that the optimal sampling probability is:

$$p_{i,t}^* = \frac{\sqrt{5KL\eta_L(\sigma_{L,i}^2 + 4K\zeta_{G,i}^2) + \frac{\eta}{n}\sigma_{L,l}^2}}{\sum_{l=1}^{m} \sqrt{5KL\eta_L(\sigma_{L,l}^2 + 4K\zeta_{G,l}^2) + \frac{\eta}{n}\sigma_{L,l}^2}}. \tag{80}$$

For simplicity, we rewrite the optimal sampling probability as:

$$p_{i,t}^* = \frac{\sqrt{\alpha_1 \zeta_{G,i}^2 + \alpha_2 \sigma_{L,i}^2}}{\sum_{l=1}^{m} \sqrt{\alpha_1 \zeta_{G,l}^2 + \alpha_2 \sigma_{L,l}^2}}, \tag{81}$$

where $\alpha_1 = 20K^2 L \eta_L$, $\alpha_2 = 5KL\eta_L + \frac{\eta}{n}$.

**Remark:** Now, we will compare this result with the uniform sampling strategy:

$$\Phi_{DELTA} = \frac{L\eta_L}{2c} \left(\frac{\sum_{l=1}^{m} \sqrt{\alpha_1 \zeta_{G,l}^2 + \alpha_2 \sigma_{L,l}^2}}{m}\right)^2. \tag{82}$$

For uniform $p_l = \frac{1}{m}$:

$$\Phi_{uniform} = \frac{L\eta_L}{2c} \frac{\sum_{l=1}^{m} \left(\sqrt{\alpha_1 \zeta_{G,l}^2 + \alpha_2 \sigma_{L,l}^2}\right)^2}{m}. \tag{83}$$

According to Cauchy-Schwarz inequality:

$$\frac{\sum_{l=1}^{m} \left(\sqrt{\alpha_1 \zeta_{G,l}^2 + \alpha_2 \sigma_{L,l}^2}\right)^2}{m} \Bigg/ \left(\frac{\sum_{l=1}^{m} \sqrt{\alpha_1 \zeta_{G,l}^2 + \alpha_2 \sigma_{L,l}^2}}{m}\right)^2 = \frac{m \sum_{l=1}^{m} \left(\sqrt{\alpha_1 \zeta_{G,l}^2 + \alpha_2 \sigma_{L,l}^2}\right)^2}{\left(\sum_{l=1}^{m} \sqrt{\alpha_1 \zeta_{G,l}^2 + \alpha_2 \sigma_{L,l}^2}\right)^2} \geq 1, \tag{84}$$

this implies that our sampling method does improve the convergence rate (our sampling approach might be $n$ times faster in convergence than uniform sampling), especially when $\frac{\left(\sum_{l=1}^{m} \sqrt{\alpha_1 \zeta_{G,l}^2 + \alpha_2 \sigma_{L,l}^2}\right)^2}{\sum_{l=1}^{m} \left(\sqrt{\alpha_1 \zeta_{G,l}^2 + \alpha_2 \sigma_{L,l}^2}\right)^2} << m$.

## G   Convergence Analysis of The Practical Algorithm

In order to provide the convergence rate of the practical algorithm, we need an additional Assumption 4 ($\|\nabla F_i(x)\|^2 \leq G^2, \forall i$). This assumption tells us a useful fact that will be used later:

It can be shown that $\|\nabla F_i(x_{t,k}, \xi_{t,k})/\nabla F_i(x_{s,k}, \xi_{s,k})\| \leq U$ for all $i$ and $k$, where the subscript $s$ refers to the last round in which client $i$ participated, and $U$ is a constant upper bound. This tells us

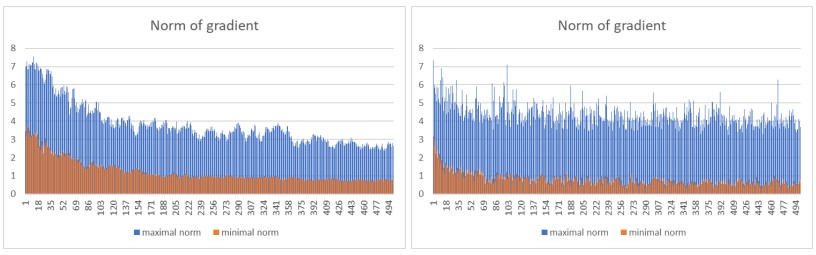

(a) FedIS's gradient norm on MNIST    (b) FedIS' gradient norm on FashionM-
NIST

Figure 9: **Performance of gradient norm of FedIS.** We evaluate the performance of FedIS on MNIST and FashionMNIST datasets. In each round, we report the maximal and minimal gradient norm among all clients.

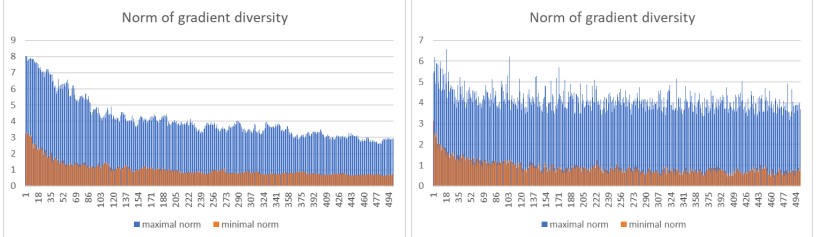

(a) Gradient diversity norm of DELTA on    (b) Gradient diversity norm of DELTA on
MNIST                                        FashionMNIST

Figure 10: **Performance of gradient diversity norm of DELTA.** We evaluate the performance of DELTA on MNIST and FashionMNIST datasets. In each round, we report the maximal and minimal gradient diversity norm among all clients.

that the change in the norm of the client's gradient is bounded. $U$ comes from the following inequality constraint procedure:

$$
V\left(\frac{1}{mp_i^s}\hat{g}_i^t\right) = E\|\frac{1}{mp_i^s}\hat{g}_i^t - \frac{1}{m}\sum_{i=1}^m \hat{g}_i^t\|^2 \leq E\|\frac{1}{mp_i^t}\hat{g}_i^t\|^2 = E\left\|\frac{1}{m}\frac{\hat{g}_i^t}{\|\hat{g}_i^s\|}\sum_{j=1}^m \|\hat{g}_j^s\|\right\|^2
$$

$$
\leq E\left(\|\frac{1}{m}\|^2 \frac{\|\hat{g}_i^t\|^2}{\|\hat{g}_i^s\|^2}\left\|\sum_{j=1}^m \|\hat{g}_j^s\|\right\|^2\right) \leq \frac{1}{m^2}U^2 m \sum_{j=1}^m K \sum_{k=1}^K E\|\nabla F_j(x_{k,s},\xi_{k,s})\|^2. \tag{85}
$$

We establish the upper bound $U$ based on two factors: (1) Assumption 4, and (2) the definition of importance sampling $E_{q(z)}(F_i(z)) = E_{p(z)}\left(q_i(z)/p_i(z)F_i(z)\right)$, where there exists a positive constant $\gamma$ such that $p_i(z) \geq \gamma > 0$. Thus, for $p_i^s = \frac{\hat{g}_i^s}{\sum_j \hat{g}_j^s} \geq \gamma$, we can easily ensure $\frac{\|g_i^t\|}{\|g_i^s\|} \leq U$ since $\hat{g}_i^s > 0$ is consistently bounded.

In general, the gradient norm tends to become smaller as training progresses, which leads to $\|\nabla F_i(x_{t,k},\xi_{t,k})/\nabla F_i(x_{s,k},\xi_{s,k})\|$ going to zero. Even if there are some oscillations in the gradient norm, the gradient will vary within a limited range and will not diverge to infinity. Figures 9 and Figure 10 depict the norms of gradients and gradient diversity across all clients in each round. Notably, these figures demonstrate that in the case of practical IS and practical DELTA, the change ratio of both gradient and gradient diversity remains limited, with the maximum norm being under 8 and the minimum norm exceeding 0.5.

Based on Assumption 4 and Assumption 3, we can re-derive the convergence analysis for both convergence variance $\Phi$ (4) and $\tilde{\Phi}$ (46). In particular, for Assumption 3 ($\mathbb{E}\|\nabla F_i(x)\|^2 \leq (A^2 + 1)\|\nabla f(x)\|^2 + \sigma_G^2$), we use $\sigma_{G,s}$ and $\sigma_{G,t}$ instead of a unified $\sigma_G$ for the sake of comparison.

Specifically, $\Phi = \frac{1}{c}[\frac{5\eta_L^2 L^2 K}{2m}\sum_{i=1}^m(\sigma_L^2 + 4K\sigma_G^2) + \frac{\eta\eta_L L}{2m}\sigma_L^2 + \frac{L\eta\eta_L}{2nK}V(\frac{1}{mp_i^t}\hat{g}_i^t)]$, where $\hat{g}_i^t = \sum_{k=1}^K \nabla F_i(x_{k,s},\xi_{k,s})$. With the practical sampling probability $p_i^s$ of FedIS:

$$
V\left(\frac{1}{mp_i^s}\hat{g}_i^t\right) = E\|\frac{1}{mp_i^s}\hat{g}_i^t - \frac{1}{m}\sum_{i=1}^m \hat{g}_i^t\|^2 \leq E\|\frac{1}{mp_i^t}\hat{g}_i^t\|^2 = E\|\frac{1}{m}\frac{\hat{g}_i^t}{\hat{g}_i^s}\sum_{j=1}^m \hat{g}_j^s\|^2. \tag{86}
$$

According to Assumption 4, we know $\|\frac{\hat{g}_i^t}{\hat{g}_i^s}\|^2 = \|\frac{\sum_{k=1}^K \nabla F_i(x_{t,k}^i, \xi_{t,k}^i)}{\sum_{k=1}^K \nabla F_i(x_{s,k}^i, \xi_{s,k}^i)}\| \le U^2$. Then we get

$$V\left(\frac{1}{mp_i^s}\hat{g}_i^t\right) \le E\left(\|\frac{1}{m}\|^2\|\|\frac{\hat{g}_i^t}{\hat{g}_i^s}\|^2\|\sum_{j=1}^m \hat{g}_j^s\|^2\right) \le \frac{1}{m^2}U^2 E\|\sum_{i=1}^m\sum_{k=1}^K \nabla F_i(x_{k,s}, \xi_{k,s})\|^2$$

$$\le \frac{1}{m^2}U^2 m\sum_{i=1}^m K\sum_{k=1}^K E\|\nabla F_i(x_{k,s}, \xi_{k,s})\|^2 \tag{87}$$

Similar to the previous proof, based on Assumption 3. we can get the new convergence rate:

$$\min_{t\in[T]} E\|\nabla f(x_t)\|^2 \le \mathcal{O}\left(\frac{f^0-f^*}{\sqrt{nKT}}\right) + \underbrace{\mathcal{O}\left(\frac{\sigma_L^2}{\sqrt{nKT}}\right) + \mathcal{O}\left(\frac{M^2}{T}\right) + \mathcal{O}\left(\frac{KU^2\sigma_{G,s}^2}{\sqrt{nKT}}\right)}_{\text{order of } \Phi}. \tag{88}$$

where $M = \sigma_L^2 + 4K\sigma_{G,s}^2$.

**Remark G.1** (Discussion on $U$ and convergence rate.). *It is worth noting that $\|\nabla F_i(x_{t,k}, \xi_{t,k})/\nabla F_i(x_{s,k}, \xi_{s,k})\|$ is typically relatively small because the gradient tends to go to zero as the training process progresses. This means that $U$ can be relatively small, more specifically, $U < 1$ in the upper bound term $\mathcal{O}\left(\frac{KU^2\sigma_{G,s}^2}{\sqrt{nKT}}\right)$. However, this does not necessarily mean that the practical algorithm is better than the theoretical algorithm because the values of $\sigma_G$ are different, as we stated at the beginning. Typically, the value of $\sigma_{G,s}$ for the practical algorithm is larger than the value of $\sigma_{G,t}$, which also comes from the fact that the gradient tends to go to zero as the training process progresses. Additionally, due to the presence of the summation over both $i$ and $k$, the gap between $\sigma_{G,s}$ and $\sigma_{G,t}$ is multiplied, and $\sigma_{G,s}/\sigma_{G,t} \sim m^2 K^2 \frac{1}{U^2}$. Thus, the practical algorithm leads to a slower convergence than the theoretical algorithm.*

Similarly, as long as the gradient is consistently bounded, we can assume that $\|\nabla F_i(x_t) - \nabla f(x_t)\|/\|\nabla F_i(x_s) - \nabla f(x_s)\| \le \tilde{U}_1 \le \tilde{U}$ and $\|\sigma_{L,t}/\sigma_{L,s}\| \le \tilde{U}_2 \le \tilde{U}$ for all $i$, where $\sigma_{L,s}^2 = \mathbb{E}\left[\|\nabla F_i(x_s, \xi_s^i) - \nabla F_i(x_s)\|\right]$. Then, we can obtain a similar conclusion by following the same analysis on $\tilde{\Phi}$.

Specifically, we have $\tilde{\Phi} = \frac{L\eta_L}{2m^2c}\sum_{i=1}^m \frac{1}{p_i^s}\left(\alpha_1\zeta_{G,i}^2 + \alpha_2\sigma_{L,i}^2\right)$, where $\alpha_1$ and $\alpha_2$ are constants defined in (13). For the sake of comparison of different participation rounds $s$ and $t$, we rewrite the symbols as $\zeta_{G,s}^i$ and $\sigma_{L,s}^i$. Then, using the practical sampling probability $p_i^s$ of DELTA, and letting $R_i^s = \sqrt{\alpha_1{\zeta_{G,s}^i}^2 + \alpha_2{\sigma_{L,s}^i}^2}$, we have:

$$\tilde{\Phi} = \frac{L\eta_L}{2m^2c}\sum_{i=1}^m \frac{1}{p_i^s}(R_i^t)^2 = \frac{L\eta_L}{2m^2c}\sum_{i=1}^m \frac{(R_i^t)^2}{R_i^s}\sum_{j=1}^m(R_j^s)^2 = \frac{L\eta_L}{2m^2c}\sum_{i=1}^m\left(\frac{R_i^t}{R_i^s}\right)^2 R_i^s\sum_{j=1}^m R_j^s$$

$$\le \frac{L\eta_L}{2m^2c}\tilde{U}^2\sum_{i=1}^m R_i^s\sum_{j=1}^m R_j^s = \frac{L\eta_L}{2m^2c}\tilde{U}^2\left(\sum_{i=1}^m R_i^s\right)^2 \le \frac{L\eta_L}{2m^2c}\tilde{U}^2 m\sum_{i=1}^m(R_i^s)^2$$

$$\le \frac{L\eta_L}{2c}\tilde{U}^2(5KL\eta_L(\sigma_{L,s}^2 + 4K\zeta_{G,s}^2) + \frac{\eta}{n}\sigma_L^2) \tag{89}$$

Therefore, compared to the theoretical algorithm of DELTA, the practical algorithm of DELTA has the following convergence rate:

$$\min_{t\in[T]}\mathbb{E}\|\nabla f(x_t)\|^2 \le \mathcal{O}\left(\frac{f^0-f^*}{\sqrt{nKT}}\right) + \underbrace{\mathcal{O}\left(\frac{\tilde{U}^2\sigma_{L,s}^2}{\sqrt{nKT}}\right) + \mathcal{O}\left(\frac{\tilde{U}^2\sigma_{L,s}^2 + 4K\tilde{U}^2\zeta_{G,s}^2}{KT}\right)}_{\text{order of } \tilde{\Phi}}. \tag{90}$$

This discussion of the effect of $\tilde{U}$ on the convergence rate is similar to the discussion of $U$ in Remark G.1.

# H   Additional Experiment Results and Experiment Details.

## H.1   Experimental Environment

For all experiments, we use NVIDIA GeForce RTX 3090 GPUs. Each simulation trail with 500 communication rounds and three random seeds.

## H.2   Experiment setup

**Setup for the synthetic dataset.**   To demonstrate the validity of our theoretical results, we first conduct experiments using logistic regression on synthetic datasets. Specifically, we randomly generate $(x, y)$ pairs using the equation $y = \log\left(\frac{(Ax-b)^2}{2}\right)$ with given values for $A_i$ and $b_i$ as training data for clients. Each client's local dataset contains 1000 samples. In each round, we select 10 out of 20 clients to participate in training (we also provide the results of 10 out of 200 clients in Figure 13).

To simulate gradient noise, we calculate the gradient for each client $i$ using the equation $g_i = \nabla f_i(A_i, b_i, D_i) + \nu_i$, where $A_i$ and $b_i$ are the model parameters, $D_i$ is the local dataset for client $i$, and $\nu_i$ is a zero-mean random variable that controls the heterogeneity of client $i$. The larger the value of $\mathbb{E}\|\nu_i\|^2$, the greater the heterogeneity of client $i$.

We demonstrate the experiment on different functions with different values of $A$ and $b$. Each function is set with noise levels of 20, 30, and 40 to illustrate our theoretical results. To construct different functions, we set $A = 8, 10$ and $b = 2, 1$, respectively, to observe the convergence behavior of different functions.

All the algorithms run in the same environment with a fixed learning rate of $0.001$. We train each experiment for 2000 rounds to ensure that the global loss has a stable convergence performance.

**Setup for FashionMNIST and CIFAR-10.**   To evaluate the performance of DELTA and FedIS, we train a two-layer CNN on the non-iid FashionMNIST dataset and a ResNet-18 on the non-iid CIFAR-10 dataset, respectively. CIFAR-10 is composed of $32 \times 32$ images with three RGB channels, belonging to 10 different classes with 60000 samples.

The "non-iid" follows the idea introduced in [66, 16], where we leverage Latent Dirichlet Allocation (LDA) to control the distribution drift with the Dirichlet parameter $\alpha$. Larger $\alpha$ indicates smaller drifts. Unless otherwise stated, we set the Dirichlet parameter $\alpha = 0.5$.

Unless specifically mentioned otherwise, our studies use the following protocol: all datasets are split with a parameter of $\alpha = 0.5$, the server chooses $n = 20$ clients according to our proposed probability from the total of $m = 300$ clients, and each is trained for $T = 500$ communication rounds with $K = 5$ local epochs. The default local dataset batch size is 32. The learning rates are set the same for all algorithms, specifically $lr_{global} = 1$ and $lr_{local} = 0.01$.

All algorithms use FedAvg as the backbone. We compare DELTA, FedIS and Cluster-based IS with FedAvg on different datasets with different settings.

**Setup for Split-FashionMNIST.**   In this section, we evaluate our algorithms on the split-FashionMNIST dataset. In particular, we let $10\%$ clients own $90\%$ of the data, and the detailed split data process is shown below:

- Divide the dataset by labels. For example, divide FashionMNIST into 10 groups, and assign each client one label
- Random select one client
- Reshuffle the data in the selected client
- Equally divided into 100 clients

**Setup for LEAF.**   To test our algorithm's efficiency on diverse real datasets, we use the non-IID FEMNIST dataset and non-IID CelebA dataset in LEAF, as given in [3]. All baselines use a 4-layer CNN for both datasets with a learning rate of $lr_{local} = 0.1$, batch size of 32, sample ratio of $20\%$ and

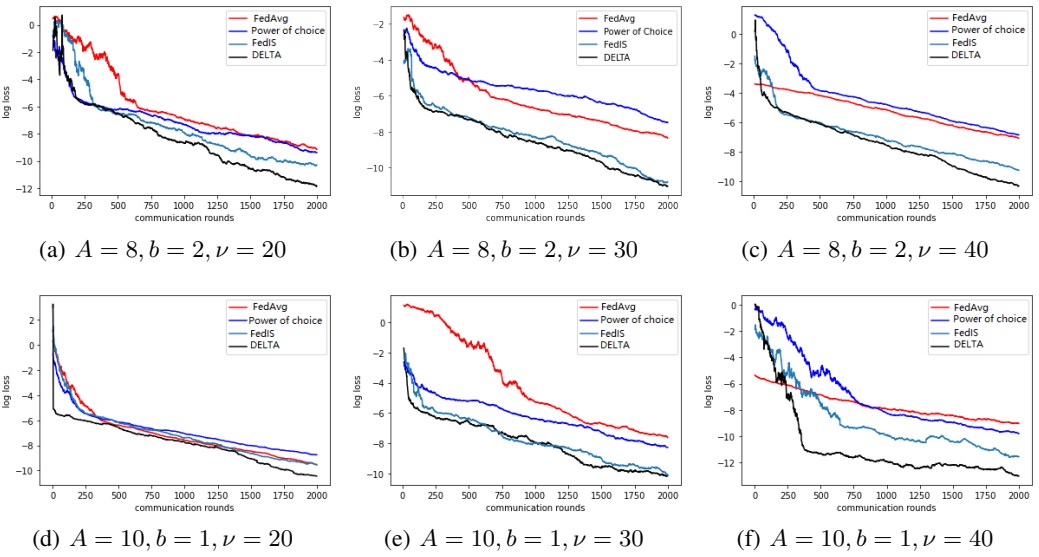

(a) $A = 8, b = 2, \nu = 20$  (b) $A = 8, b = 2, \nu = 30$  (c) $A = 8, b = 2, \nu = 40$

(d) $A = 10, b = 1, \nu = 20$  (e) $A = 10, b = 1, \nu = 30$  (f) $A = 10, b = 1, \nu = 40$

Figure 11: Performance of different algorithms on the regression model. The loss is calculated by $f(x, y) = \left\| y - \log\left(\frac{(A_i x - b_i)^2}{2}\right) \right\|^2$, we report the logarithm of the global loss with different degrees of gradient noise $\nu$. All methods are well-tuned, and we report the best result of each algorithm under each setting.

communication round of $T = 500$. The reported results are averaged over three runs with different random seeds.

**The implementation detail of different sampling algorithms.** The power-of-choice sampling method is proposed by [7]. The sampling strategy is to first sample 20 clients randomly from all clients, and then choose 10 of the 20 clients with the largest loss as the selected clients. FedAvg samples clients according to their data ratio. Thus, FedAvg promises to be unbiased, which is given in [12, 31] to be an unbiased sampling method. As for FedIS, the sampling strategy follows Equation (5). For cluster-based IS, it first clusters clients following the gradient norm and then uses the importance sampling strategy similar to FedIS in each cluster. And for DELTA, the sampling probability follows Equation (13). For the practical implementation of FedIS and DELTA, the sampling probability follows the strategy described in Section 4.

### H.3 Additional Experimental Results

**Performance of algorithms on the synthetic dataset.** We display the log of the global loss of different sampling methods on synthetic dataset in Figure 11, where the Power-of-Choice is a biased sampling strategy that selects clients with higher loss [7].

We also show the convergence behavior of different sampling algorithms under small noise, as shown in Figure 12.

To simulate a large number of clients, we increased the client number from 20 to 200, with only 10 clients participating in each round. The results in Figure 13 demonstrate the effectiveness of DELTA.

**Convergence performance of theoretical DELTA on split-FashionMNIST and practical DELTA on FEMNIST.** Figure 14(a) illustrates the theoretical DELTA outperforms other methods in convergence speed. Figure 14(b) indicates that cluster-based IS and practical DELTA exhibit rapid initial accuracy improvement, while practical DELTA and practical IS achieve higher accuracy in the end.

**Ablation study for DELTA with different sampled numbers.** Figure 15 shows the accuracy performance of practical DELTA algorithms on FEMNIST with different sampled numbers of clients. In particular, the larger number of sampled clients, the faster the convergence speed is. This is consistent with our theoretical result (Corollary 4.2).

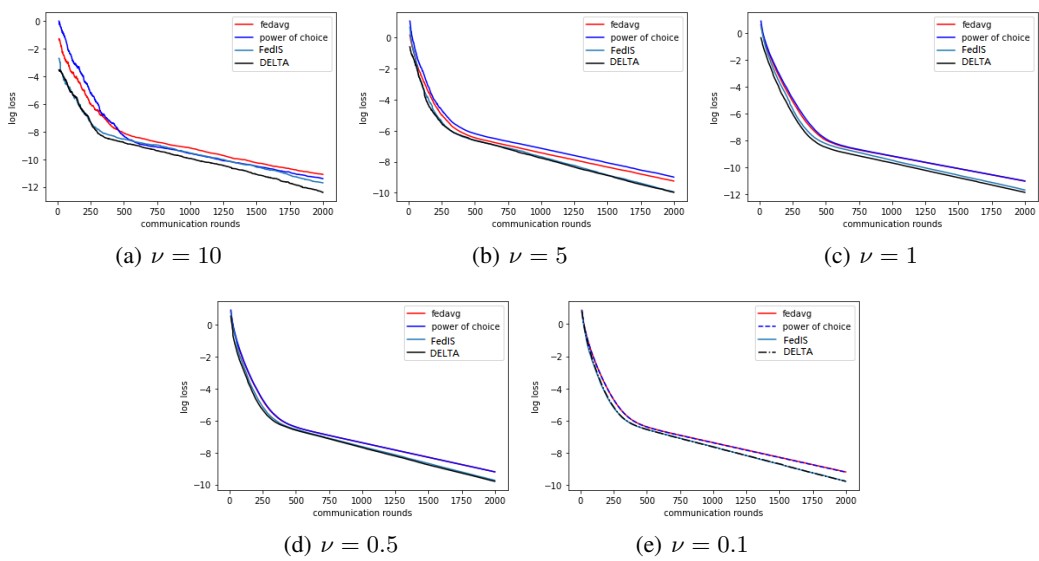

Figure 12: Performance of different algorithms on the regression model with different (small) noise settings.

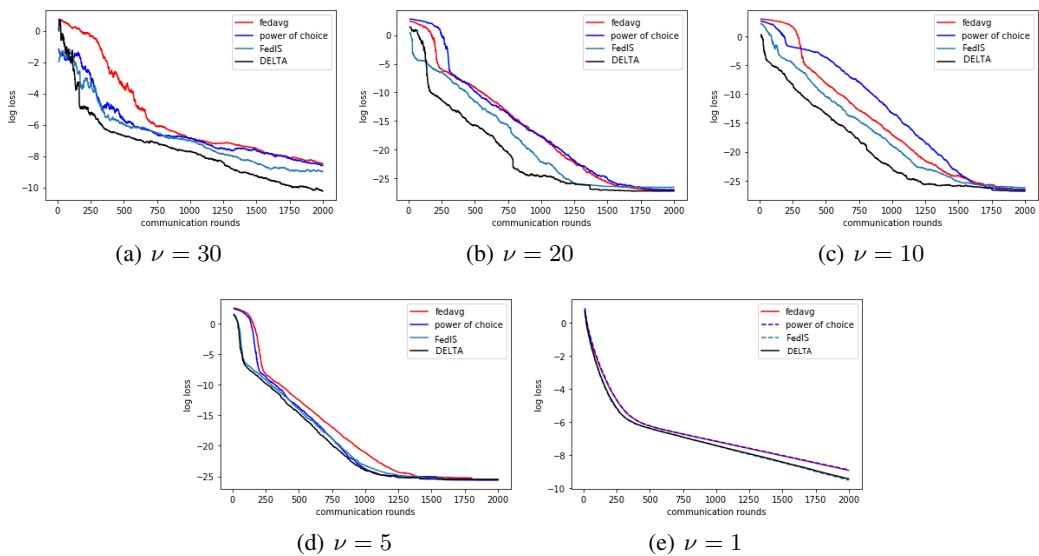

Figure 13: Performance of different algorithms on synthetic data with different noise settings. Specifically, for testing the large client number setting, in each round, 10 out of 200 clients are selected to participate in training.

**Performance on FashionMNIST and CIFAR-10.** For CIFAR-10, we report the mean of the best 10 test accuracies on test data. In Table 2, we compare the performance of DELTA, FedIS, and FedAvg on non-IID FashionMNIST and CIFAR-10 datasets. Specifically, we use $\alpha = 0.1$ for FashionMNIST and $\alpha = 0.5$ for CIFAR-10 to split the datasets. As for Multinomial Distribution (MD) sampling [29], it samples based on the clients' data ratio and average aggregates. It is symmetric in sampling and aggregation with FedAvg, with similar performance. It can be seen that DELTA has better accuracy than FedIS, while both DELTA and FedIS outperform FedAvg with the same number of communication rounds.

**Assessing the Compatibility of FedIS with Other Optimization Methods.** In Table 4, we demonstrate that DELTA and FedIS are compatible with other FL optimization algorithms, such as Fedprox [29] and FedMIME [20]. Furthermore, DELTA maintains its superiority in this setting.

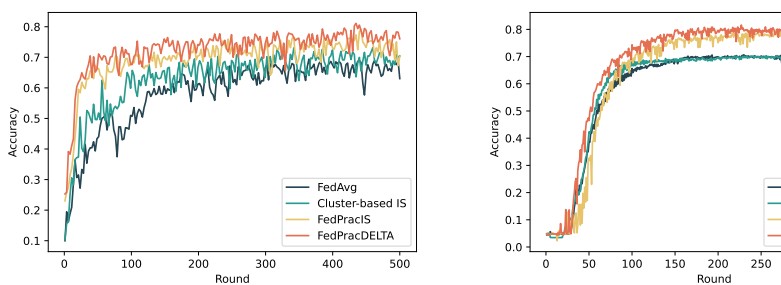

(a) Performance of algorithms on split-FashionMNIST

(b) Performance of algorithms on FEMNIST

Figure 14: Performance comparison of accuracy using different sampling algorithms.

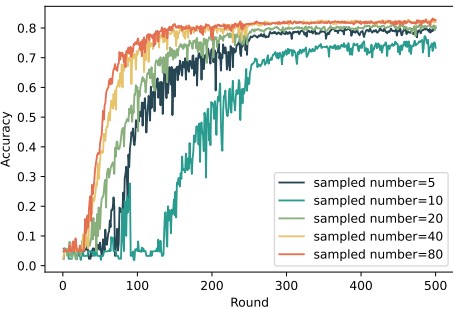

Figure 15: Ablation study of the number of sampled clients.

Table 4: **Performance of sampling algorithms integrated with momentum and prox.** We run 500 communication rounds on CIFAR-10 for each algorithm. We report the mean of maximum 5 accuracies for test datasets and the number of communication rounds to reach the threshold accuracy.

| Algorithm | Sampling + momentum | | Sampling + proximal | |
|---|---|---|---|---|
| | Acc (%) | Rounds for 65% | Acc (%) | rounds for 65% |
| FedAvg (w/ uniform sampling) | 0.6567 | 390 | 0.6596 | 283 |
| FedIS | 0.6571 | **252** | 0.661 | 266 |
| DELTA | **0.6604** | 283 | **0.6677** | **252** |

In Table 5, we demonstrate that DELTA and FedIS are compatible with other variance reduction algorithms, like FedVARP [18].

It is worth noting that FedVARP utilizes the historic update to approximate the unparticipated clients' updates. However, in this setting, the improvement of the sampling strategy on the results is somewhat reduced. This is because the sampling strategy is slightly redundant when all users are involved. Thus, when VARP and DELTA/FedIS are combined, instead of reassigning weights in the aggregation step, we use (75) or (13) to select the current round update clients and then average aggregate the updates of all clients. One can see that the combination of DELTA/FedIS and VARP can still show the advantages of sampling.

Table 5: **Performance of DELTA/FedIS in combination with FedVARP.** We run 500 communication rounds on FashionMNIST with $\alpha = 0.1$ for each algorithm. We report the mean of maximum 5 accuracies for test datasets and the number of communication rounds to reach the threshold accuracy.

| Algorithm | FashionMNIST | |
|---|---|---|
| | Acc (%) | Rounds for 73% |
| FedVARP | $73.81 \pm 0.18$ | 470 |
| FedIS + FedVARP | $73.96 \pm 0.14$ | 452 |
| DELTA +FedVARP | $\mathbf{74.22} \pm 0.14$ | **436** |

In addition to the above optimization methods like VARP, we also conduct experiments on LEAF (FEMNIST and CelebA) with other non-vanilla SGD algorithms, namely Adagrad and Adam, to show that the proposed client selection framework be applied to federated learning algorithms other than vanilla SGD. The results are shown in Table 6 and Table 7.

Table 6: **Performance of sampling algorithms integrated with Adagrad and Adam on FEMNIST.** We run 1000 communication rounds on FEMNIST for each algorithm. In particular, we set the global learning rate $\eta = 0.01$ to ensure convergence of Adagrad and Adam. We report the mean of the maximum of 5 accuracies for test datasets and the average number of communication rounds that reach the threshold accuracy $80\%$.

| Algorithm | Sampling + Adagrad | | Sampling + Adam | |
|---|---|---|---|---|
| | Acc (%) | Rounds for 80% | Acc (%) | rounds for 80% |
| FedAvg | 80.93 ± 0.08 | 893 (1.0×) | 80.04 ± 0.85 | 882 (1.0×) |
| Cluster-based IS | 80.69±0.42 | 760 (1.17×) | 79.11± 0.18 | - |
| FedIS | 80.96 ± 0.31 | 723 (1.24×) | 80.10 ±0.25 | 787 (1.12×) |
| DELTA | 81.79 ± 0.09 | 612 (1.46×) | 80.92 ±0.27 | 600 (1.47×) |

Table 7: **Performance of sampling algorithms integrated with Adagrad and Adam on CelebA.** We run 1000 communication rounds on CelebA for each algorithm. In particular, we set the global learning rate $\eta = 0.01$ to ensure convergence of Adagrad and Adam. We report the mean of the maximum of 5 accuracies for test datasets and the average number of communication rounds that reach the threshold accuracy $80\%$.

| Algorithm | Sampling + Adagrad | | Sampling + Adam | |
|---|---|---|---|---|
| | Acc (%) | Rounds for 80% | Acc (%) | rounds for 80% |
| FedAvg | 88.92 ± 0.08 | 329 (1.0×) | 89.04 ± 0.22 | 244 (1.0×) |
| Cluster-based IS | 89.71±0.10 | 329 (1.0×) | 89.26± 0.19 | 164 (1.49×) |
| FedIS | 90.14 ± 0.01 | 243 (1.35×) | 89.92 ±0.05 | 140 (1.74×) |
| DELTA | 90.38 ± 0.02 | 214 (1.54×) | 90.58 ±0.07 | 109 (2.24×) |

Table 8 provide results under some common thresholds, including 50% for CIFAR-10 and 80% for CelebA, to replace 54% for CIFAR-10 and 85% for CelebA in Table 2.

Table 8: **Performance of sampling algorithms under the common thresholds.** We report the results of algorithms on CIFAR-10 with $50\%$ threshold and on CelebA with $80\%$ threshold. We run 500 communication rounds for each algorithm. We report the mean of the maximum of 5 accuracies for test datasets and the average number of communication rounds that reach the threshold accuracy.

| Algorithm | CIFAR-10 | | CelebA | |
|---|---|---|---|---|
| | Acc (%) | Rounds for 50% | Acc (%) | rounds for 80% |
| FedAvg | 54.28 ± 0.29 | 181 (1.0×) | 85.92 ± 0.89 | 339 (1.0×) |
| Cluster-based IS | 54.83± 0.02 | 187 (0.91×) | 86.77± 0.11 | 303 (1.11×) |
| FedIS | 55.05 ± 0.27 | 168 (1.07×) | 89.12 ±0.71 | 261 (1.29×) |
| DELTA | 55.20 ± 0.26 | 151 (1.20×) | 89.67 ±0.56 | 257 (1.32×) |

**Albation study for $\alpha$.** In Table 9, we experiment with different choices of heterogeneity $\alpha$ in the CIFAR-10 dataset. The parameter of heterogeneity $\alpha$ changes from 0.1 to 0.5 to 1. We observe a consistent improvement of DELTA compared to the other algorithms. This shows that DELTA is robust to changes in the level of heterogeneity in the data distribution.

Table 9: **Performance of algorithms under different $\alpha$.** We run 500 communication rounds on CIFAR10 for each algorithm (with momentum). We report the mean of maximum 5 accuracies for test datasets and the number of communication rounds to reach the threshold accuracy.

| Algorithm | $\alpha = 0.1$ | | $\alpha = 0.5$ | | $\alpha = 1.0$ | |
|---|---|---|---|---|---|---|
| | Acc (%) | Rounds for 42% | Acc (%) | rounds for 65% | Acc (%) | rounds for 71% |
| FedAvg (w/ uniform sampling) | 0.4209 | 263 | 0.6567 | 283 | 0.7183 | 246 |
| FedIS | 0.427 | 305 | 0.6571 | **252** | 0.7218 | 239 |
| DELTA | **0.4311** | **209** | **0.6604** | 283 | **0.7248** | **221** |

