# OpenReview forum: "DELTA: Diverse Client Sampling for Fasting Federated Learning"
_NeurIPS.cc/2023/Conference — NeurIPS 2023 poster_

### Official Review · Reviewer_9Bi1 · 2023-07-04

**Soundness:** 2 fair
**Presentation:** 2 fair
**Contribution:** 2 fair
**Rating:** 5
**Confidence:** 2

**Summary:**

The paper proposed an unbiased client sampling method in Federated Learning. The proposed method is motivated by considering both variance and gradient diversity when doing client sampling. The authors theoretically proved that the proposed sampling method can achieve better convergence rate for non-convex objective functions. Besides, the authors also demonstrated its practical advantages by numerical experiments.

**Strengths:**

Considering gradient diversity is a novel idea in FL client sampling research. The paper demonstrates its proposed method by both theory and experiments. The motivation is clear, and the paper is in general good writing. Given that client sampling is an important problem of federated learning and the claimed advantage of the proposed method, the paper can have significant impact on the area.

**Weaknesses:**

Assumption 4 is a very strong assumption. It can be violated in many problems. For example, the simple least square regression problem can violate this assumption.

Besides, the paper seems to have too much content for a nine-page paper. As a result, the key theoretical result is lack of intuition.

**Questions:**

1.	May the authors discuss what will happen if Assumption 4 does not hold?
2.	How large is U used in the theory of practical IS and practical DELTA?
3.	May the authors discuss what is the key intuition of theoretical advantage of the new IS analysis? In my understanding, the importance sampling can only improve the constants rather than the convergence order. Thus, I find the theoretical result quite surprising. Unfortunately, I do not have enough time to check the correction of the theory.

**Limitations:**

The authors mentioned that their proposed method does not address the backdoor attack concern.

---

> ### Author Rebuttal · Authors · 2023-08-09
>
> Dear reviewer 9Bi1, thank you for providing constructive feedback. We have fully revised our manuscript and have addressed all of the comments, as well as added new experiments to further strengthen our work. Please find our responses to your raised questions below:
>
> > Assumption 4 is a very strong assumption. It can be violated in many problems. For example, the simple least square regression problem can violate this assumption.
> >
>
> Thank you for your suggestion. We would like to explain the soundness of Assumption 4:
>
> - This assumption is essential for IS methods [1,2,3]. Furthermore, it finds common usage in the FL community for convergence analysis [4,5,6,7].
> In particular, according to the definition of IS, $E_{q(x)}[h(x)]=E_{p(x)}\left[\frac{q(x)}{p(x)} h(x)\right]$, where $q(x)$ is the given sampling distribution, $p(x)$  is our proposed sampling distribution and $h(x)$  is the value function. With a little work (similar to our Corollary F.1), one can prove the variance is minimized when $p^*(x) \propto q(x)\|h(x)\|$. If $\|h(x)\|$ is not consistently bounded, $p(x)$ is meaningless.
> - We would like to clarify that in our practical algorithm of DELTA, the used assumption is gradient heterogeneity bound: $E|| \nabla F_i(x_t)-\nabla f(x_t)||^2 \leq G^2$ instead of Assumption 4, as used in eq (89) of the convergence analysis of the practical algorithm. This is a looser assumption than Assumption 4.
> - Figure 1 in the one-page response PDF demonstrates the gradient norm of FedIS on MNIST and FashionMNIST datasets, suggesting gradient can be bounded.
>
> For the least square regression problem, the gradient becomes unbounded only if the model parameters are at infinity in the parameter space. In other words, when the model parameters are normally distributed in the parameter space, the gradient is always finite-large and therefore bounded.
>
> > May the authors discuss what will happen if Assumption 4 does not hold?
> >
>
> We would like to clarify the consequence/meaning of Assumption 4 not holding and explain it :
>
> - If Assumption 4 doesn’t hold:
>     1. All IS-type methods become inapplicable in FL, given that this assumption forms an inherent prerequisite for utilizing IS to attribute meaning to the expression $p^*(x)\propto ||\nabla F_i(x)||$.
>     2. FL algorithms may falter in training since a lack of gradient bounding implies that gradients can assume "nan" values, thereby disrupting the deep learning training process.
>     3. For the practical FedIS algorithm, employing the user's last round participation gradient to approximate the current gradient will introduce an unbounded, significant noise.
> - Figure 1 in the one-page response PDF showcases the gradient norm of FedIS on MNIST and FashionMNIST datasets, indicating that the gradient can be bounded, as its norm remains below 8.
>
> > How large is U used in the theory of practical IS and practical DELTA?
> >
>
> We would like to clarify $U$ in practical IS and $\tilde{U}$ in practical DELTA possess limited magnitudes. We also provide the experimental results to illustrate it.
>
> - Intuitively, $U=||\nabla F_i(x_{t,k},\xi_{t,k}) / \nabla F_i(x_{s,k},\xi_{s,k})||$ (same to $\tilde{U}$) tends to be smaller than 1 since the gradient tends to be zero as training processing, where index $t$ represent the current communication round and $s$ represent the last participated round of client $i$. The discussion of $U$ is provided in Remark G.1, Appendix G.
> - Figures 1 and 2 in the one-page response PDF depict the norms of gradients and gradient diversity across all clients in each round. Notably, these figures demonstrate that in the case of practical IS and practical DELTA, the change ratio of both gradient and gradient diversity remains limited, with the maximum norm being under 8 and the minimum norm exceeding 0.5.
>
> > May the authors discuss what is the key intuition of theoretical advantage of the new IS analysis?
> >
>
> Thank you for your comment. We are willing to share with you the intuition of our theoretical advantage of IS analysis.
>
> First, we clarify that the rate improvement comes from our improved analysis of unbiased sampling in federated learning, including FedAvg. IS then reduces variance.
>
> ***Intuition:***
>
> - The intuition of unbiased sampling FL: Though the existing analysis for partial user participation in FL (including FedAvg) can be extended to unbiased sampling scenarios, *the failure to leverage the unbiased property leads to imprecise upper bounds on convergence for unbiased sampling FL [1,2,3].*
> - The intuition of FedIS: Following the unbiased sampling FL analysis, we optimize the convergence variance with respect to sampling probability.
>
> For details explanation and evidence, please see “Reply to All Reviewers, Elaborate on the novelty and contribution of our analysis below”.
>
> [1]Stochastic optimization with importance sampling for regularized loss minimization, ICML 2015
>
> [2]Not all samples are created equal: Deep learning with importance sampling. ICML 2018
>
> [3]Low-Cost Lipschitz-Independent Adaptive Importance Sampling of Stochastic Gradients. ICPR, 2021
>
> [4]Diverse client selection for federated learning: Submodularity and convergence analysis. ICML 2021
>
> [5]On the effectiveness of partial variance reduction in federated learning with heterogeneous data. CVPR. 2023
>
> [6]Sharper convergence guarantees for asynchronous sgd for distributed and federated learning. NIPS, 2022
>
> [7]Optimal client sampling for federated learning. TMLR, 2022.

---

> ### Author Response · Authors · 2023-08-18
> **To Reviewer 9Bi1**
>
> Dear Reviewer 9Bi1,
>
> I hope all is well.
>
> We are sincerely grateful for your valuable time and expertise devoted to assessing our submission. Your insights and comments have greatly aided in refining our work.
>
> Regarding your concerns, we hope that our rebuttal was able to shed light on them. In summary, we prove the practical DELTA requires a looser assumption($E\|\nabla F_i(x_t)-\nabla f(x_t)\|^2 \leq G^2$) instead of Assumption 4, while the practical FedIS relies on Assumption 4 since it constitutes a necessary requirement for IS in deep learning. Furthermore, we offer experimental results demonstrating bounded gradient norms and diversity. Moreover, we explain the key intuition behind our advanced FedIS analysis: rate improvement via refined unbiased sampling in FL with partial client participation, and variance reduction via IS, as you understand.
>
> Your valuable suggestions have greatly improved our paper. We deeply value your expertise, and we have integrated your feedback to refine our work. Thank you once again for your invaluable time and consideration.
>
> Best Regards,
>
> authors

---

> > ### Comment · Reviewer_9Bi1 · 2023-08-18
> > **Reply to Rebuttal**
> >
> > Thank you for your reply. Most of my concerns are addressed. However, I still have doubts on how IS can improve the order of convergence rate rather than the constant. I do not find the intuition for theory very convincing on this achievement. I would like to keep my score due to lack of confidence.

---

> > > ### Author Response · Authors · 2023-08-19
> > > **To Reviewer 9Bi1**
> > >
> > > Thank you very much for your feedback and the time you've dedicated to reviewing our rebuttal. There might be some misunderstanding regarding your questions on the IS analysis. We would like to offer further elaborations as follows:
> > >
> > > - **`[IS improves the constant]`** First of all, we would like to reiterate that the role of IS is to reduce convergence variance, i.e., improve the constant of term **$\mathcal{O}(\frac{1}{\sqrt{T}})$**,  as per your understanding.
> > > - **`[Enhanced analysis of FL under unbiased sampling improves the order]`** As for the order improvement of term $\mathcal{O}(\frac{1}{T^{\frac{2}{3}}})$, it results from our improved analysis of FL under unbiased sampling. The intuition behind why the improved analysis of FL under unbiased sampling surpasses the existing analysis of FL with partial client participation (vanilla FedAvg with random sampling)[1,2,3] is:
> > >     - `[Unbiased sampling bridges the convergence rate gap between partial and full user participation]` **Unbiased sampling allows for the equitable transformation of the aggregated model updating bounds with partial user participation into model updating bounds with full user participation**, such as $E\|\Delta_t\|^2$ in Eq. (33), thereby improving the convergence rate order from vanilla FedAvg with random sampling $\mathcal{O}(\frac{1}{T}+\frac{1}{\sqrt{T}} + \frac{1}{T^{\frac{2}{3}}})$ to full client participation $\mathcal{O}(\frac{1}{T}+\frac{1}{\sqrt{T}})$. Comprehensive information on the utilization of unbiased sampling in the derivation process can be found in our "Reply to All Reviewers.”
> > >     - `[Evidence of same rate as full participation]` When incorporating full client participation into our analysis, **the upper bound of our convergence recovers the convergence rate of FedAvg with full client participation**, i.e., $\mathcal{O}(\frac{1}{T}+\frac{1}{\sqrt{nKT}})$ [2].
> > >     - `[An improved FedAvg analysis]` Our improved analysis of FL under unbiased sampling can also be viewed as an improved convergence analysis for vanilla FedAvg with random sampling.
> > > - **`[Comparison between ours and existing IS]`** Since IS is one kind of unbiased sampling, **we establish the convergence analysis of IS based on our improved analysis of FL under unbiased sampling.** Thus, our FedIS analysis can achieve an order of $\mathcal{O}(\frac{1}{T}+\frac{1}{\sqrt{T}})$ , in which IS helps reduce the constant of term $\mathcal{O}(\frac{1}{\sqrt{T}})$.  In contrast, **existing IS analysis is established on the analysis of vanilla FedAvg with partial client participation** with order $\mathcal{O}(\frac{1}{T}+\frac{1}{\sqrt{T}} + \frac{1}{T^{\frac{2}{3}}})$, thus our FedIS analysis yields superior outcomes compared to prior research efforts.
> > > - **`[Same upper bound in recent studies]`** Some very recent studies [4,5,6] have similarly noted that the convergence rate of vanilla FedAvg is not tight. These studies have employed various techniques to attain a convergence rate of $\mathcal{O}(1/\epsilon+1/\epsilon^2)$, as our rate order, rather than the rate associated with vanilla FedAvg. This achievement gets a tighter convergence result as the mini-batch SGD convergence is lower bounded by $\mathcal{O}(1/\epsilon+1/\epsilon^2)$ [7]. Though the used techniques are different and orthogonal to ours (unbiased sampling), these variance reduction methods all achieve the same order.
> > >
> > > In summary, we present a novel theoretical analysis of nonconvex FedIS. The improved analysis of FL under unbiased sampling establishes a convergence rate of order $\mathcal{O}(\frac{1}{T}+\frac{1}{\sqrt{T}})$, and the role of IS contributes to reducing the constant of term $\mathcal{O}(\frac{1}{\sqrt{T}})$.
> > >
> > > [1]Scaffold: Stochastic controlled averaging for federated learning. ICML, 2020.\
> > > [2]Achieving linear speedup with partial worker participation in non-iid federated learning. ICLR 2021.\
> > > [3]Reddi S J, Charles Z, Zaheer M, et al. Adaptive Federated Optimization[C]//International Conference on Learning Representations. 2020.\
> > > [4] Li B, Schmidt M N, Alstrøm T S, et al. Partial Variance Reduction improves Non-Convex Federated learning on heterogeneous data[J]. arXiv preprint arXiv:2212.02191, 2022.\
> > > [5] Koloskova A, Stich S U, Jaggi M. Sharper convergence guarantees for asynchronous sgd for distributed and federated learning[J]. arXiv preprint arXiv:2206.08307, 2022.\
> > > [6] Wang S, Ji M. A Unified Analysis of Federated Learning with Arbitrary Client Participation[C]//Advances in Neural Information Processing Systems.2022\
> > > [7]Arjevani Y, Carmon Y, Duchi J C, et al. Lower bounds for non-convex stochastic optimization[J]. Mathematical Programming, 2023, 199(1-2): 165-214.

---

> ### Author Response · Authors · 2023-08-21
> **To reviewer 9Bi1**
>
> Dear reviewer 9Bi1,
>
> We greatly appreciate your insightful review, which plays a crucial role in enhancing the quality of our manuscript. Furthermore, we've incorporated comments from other reviewers to further enhance our paper. We summarize the revisions in the "Author Rebuttal by Authors" global comment.
>
> As the discussion period is ending soon, we want to confirm that our responses, including the second-round discussion on the intuition of FedIS analysis, have adequately addressed your concerns.
> We are happy to take any follow-up questions and look forward to discussing with you.
>
> We sincerely appreciate your expertise and dedicated time in reviewing our manuscript.
>
> Warmest regards,
>
> Authors

---

### Official Review · Reviewer_yPdX · 2023-07-06

**Soundness:** 3 good
**Presentation:** 4 excellent
**Contribution:** 3 good
**Rating:** 7
**Confidence:** 4

**Summary:**

The authors introduce DELTA (Diverse Client Sampling) which is an unbiased method for client selection in Federated Learning (FL). DELTA is  heavily inspired by Importance Sampling (IS) and cluster-based IS, resulting in sampling diverse clients with significant gradients however without the clustering. They provide convergence rate analysis of FL under IS scheme (FedIS) and then of FL under their DELTA scheme (FedDELTA) and closed form expressions for the sampling probabilities are given and discussed. Practical versions of those algorithms FedPracIS and FedPracDELTA are also analyzed. In the experiments loss profiles, accuracies and timings of proposed algorithms against baselines (including FedAvg, Power-of-Choice and cluster-based IS) and over a collection of real and synthetic datasets are presented and discussed, exhibiting the empirical advantages of the sampling scheme.


**Strengths:**

- Comprehensive and nicely structured, easy to follow presentation, with a good balance of theoretical, practical and intuitive/explanatory information.

- Rich empirical results also communicating the benefits of proposed, analyzed sampling approaches.

- Simplicity of resulting FL algorithm: Basically the well-known, seminal FedAvg idea/template with a novel client sampling scheme (DELTA, PracticalDELTA) on top (Algorithm 1).


**Weaknesses:**

- Minor: "developing an efficient and effective practical algorithm for gradient-based sampling methods" in future work section (Section 6) could confuse the reader (since DELTA is exactly such an algorithm). Perhaps  a different wording, emphasizing room of improving over this new/existing development?


**Questions:**

- In the experiments, there seems to be a variety of thresholds (70%, 54%, 85%, 80%) for different experiments and metrics in Tables 2 and 3. I would expect a more uniform way of presentation, some common threshold.

- On a related topic: Since for Table 2, 500 communication rounds were completed for all (dataset, algorithm) pairs why mean of maximum five accuracies? Are all these maximum values observed within the last x% of rounds? How do these accuracy profiles evolve? Have they saturated after 500 rounds or there is still room for improvements? I mean some short comment summarizing the full profile plots (similar to Figures 12a and 12b in the appendix) would be more than sufficient.

- How do you implement a sampling strategy like that of Equation (16) for FedPracDELTA in the experiments. There are parameters like alpha's, sigma's and zeta's in those: it would be great to include or refer to brief hints on how you choose values for those for conducting the empirical evaluation.

---

> ### Author Rebuttal · Authors · 2023-08-09
>
> Dear reviewer yPdX, thank you very much for your appreciation of our work. Please find our responses to your raised questions below:
>
> > Minor: "developing an efficient and effective practical algorithm for gradient-based sampling methods" in future work section (Section 6) could confuse the reader (since DELTA is exactly such an algorithm). Perhaps a different wording, emphasizing room of improving over this new/existing development?
> >
>
> Thanks for your suggestion, we would like to use the statement “further improve the efficiency of the existing gradient-based sampling methods” to enhance clarity.
>
> > In the experiments, there seems to be a variety of thresholds (70%, 54%, 85%, 80%) for different experiments and metrics in Tables 2 and 3. I would expect a more uniform way of presentation, some common threshold.
> >
>
> Thanks for your suggestion, we are willing to provide results under some common thresholds in the below table, including 50% for CIFAR-10 and 80% for CelebA, to replace 54% for CIFAR-10 and 85% for CelebA.
>
> In addition, we would like to clarify that we use these thresholds (like 54%) in our paper because they are the best integer accuracies that FedAvg can achieve in our experiments.
>
> |  | CIFAR-10 |  | CelebA |  |
> | --- | --- | --- | --- | --- |
> |  Algorithm | Accuracy (%) |  Communication Rounds for 50%  | Accuracy (%)  | Communication Rounds for 80%  |
> | FedAvg | 54.28±0.29 | 181 (1.0$\times$) | 85.92±0.89 | 339 (1.0$\times$) |
> | Cluster-based IS | 54.83±0.02  | 187 (0.91$\times$) | 86.77±0.11 | 303 (1.11$\times$) |
> | FedIS | 55.05±0.27 | 168 (1.07$\times$) | 88.12±0.71 | 261 (1.29$\times$) |
> | DELTA  | 55.20 ±0.26 |  151 (1.20$\times$) | 89.67 ±0.56 | 257 (1.32$\times$) |
>
> > On a related topic: Since for Table 2, 500 communication rounds were completed for all (dataset, algorithm) pairs why mean of maximum five accuracies? Are all these maximum values observed within the last x% of rounds? How do these accuracy profiles evolve? Have they saturated after 500 rounds or there is still room for improvements? I mean some short comment summarizing the full profile plots (similar to Figures 12a and 12b in the appendix) would be more than sufficient.
> >
> We appreciate your suggestion. We would like to include comments about Table 2 in the experiment section as follows:
>
> The maximum values reported in Table 2 are observed during the last 4% of rounds, where these algorithms have already reached convergence. The term 'maximum five accuracies' refers to the mean of the five highest accuracy values obtained within the plateau region of the accuracy curve. By computing the average of these values, we aim to minimize error and variance.
>
> > How do you implement a sampling strategy like that of Equation (16) for FedPracDELTA in the experiments. There are parameters like alpha's, sigma's and zeta's in those: it would be great to include or refer to brief hints on how you choose values for those for conducting the empirical evaluation.
> >
>
> Thank you for your constructive suggestion. We would like to clarify the implementation of sampling algorithms:
>
> We introduce the setting of parameters in Appendix H.2 in the submission. Specifically, for $\alpha$, the default value is 0.5, whereas $\sigma$ and $\zeta$ are not hyperparameters but rather can be implemented by computing the locally obtained gradients, as elaborated in L262, L263, and eq(16).
>
> We are willing to include brief hints on how to set these hyperparameters in the main paper of our revised version.

---

> > ### Comment · Reviewer_yPdX · 2023-08-20
> >
> > Thank you very much for the clarifications and the clearly stated plans for incorporating suggested changes. I will keep my score.

---

> > > ### Author Response · Authors · 2023-08-21
> > > **To reviewer yPdX**
> > >
> > > Dear reviewer yPdX,
> > >
> > > Thanks a lot for valuing our efforts. We sincerely appreciate your time and dedication in reviewing our paper.

---

### Official Review · Reviewer_stNZ · 2023-07-06

**Soundness:** 2 fair
**Presentation:** 1 poor
**Contribution:** 2 fair
**Rating:** 5
**Confidence:** 3

**Summary:**

This paper proposes a client sampling scheme in federated learning for faster convergence. The authors argue that the previous client sampling method based on importance sampling ignores gradient diversity. Convergence analysis is provided, as well as experiments on four image datasets.

**Strengths:**

The problem of client selection in federated learning is interesting and worth investigating.

**Weaknesses:**

[1] The paper overall is not well-written. Both organization and writing could be improved. There are multiple grammar errors even in the Introduction.

[2] Both theoretical and main method seems to rely on gradient information. However, gradient information is usually unavailable in a federated learning setting. It is unclear to me what's the contribution of this paper either theoretically or practically.

**Questions:**

See weakness.

**Limitations:**

The authors didn't really discuss limitation, even though it has a limitation session.

---

> ### Author Rebuttal · Authors · 2023-08-09
>
> Dear reviewer stNZ, thanks for your time in reviewing our paper, please find our responses to your raised questions:
>
> > Both theoretical and main method seems to rely on gradient information. However, gradient information is usually unavailable in a federated learning setting. It is unclear to me what's the contribution of this paper either theoretically or practically.
> >
>
> We have carefully checked our writing according to your suggestions.
>
> **Regarding gradient information in FL:**
>
> - This paper’s gradient information is the same as the model difference, differing only by a learning rate $\eta_L$ multiplier $\Delta_t^i=x_{t,K}^i-x_{t,0}^i=-\eta_L\sum_{k=0}^{K-1} g_{t,k}^i$. Our implementation aligns with the majority of works in FL community, which necessitate user-provided gradients (or model difference), such as [1,2,3,4,5,6,7,8].
> - Gradient privacy protection stands as a distinct research direction from the sampling algorithm studied in this paper. Furthermore, privacy protection techniques like secret sharing and encryption protocols [9] are orthogonal to our sampling algorithm and can be directly employed for information preservation.
>
> **Clarify our contribution:**
>
> Please see “Reply to All reviewers”, in which we clarify our contribution item by item.
>
> [1]Communication-efficient learning of deep networks from decentralized data. AIS 2017
>
> [2]Tackling the objective inconsistency problem in heterogeneous federated optimization. NIPS 2020
>
> [3]Federated learning under importance sampling. IEEE Transactions on Signal Processing 2022
>
> [4]Can 5th Generation Local Training Methods Support Client Sampling? Yes! ICAIS 2023
>
> [5]Optimal client sampling for federated learning. TMLR 2022
>
> [6Fast heterogeneous federated learning with hybrid client selection. UAI, 2023
>
> [7]Federated optimization in heterogeneous networks. MLS 2020
>
> [8]Scaffold: Stochastic controlled averaging for federated learning. ICML 2020
>
> [9]A privacy preserving federated learning scheme using homomorphic encryption and secret sharing. Telecommunication Systems, 2023

---

> > ### Comment · Reviewer_stNZ · 2023-08-14
> > **Rebuttal Reply**
> >
> > Thanks for the response.
> >
> > I was confused by Algorithm 1. Lines 7 and 8 give me the impression that each client only performs one gradient descent step. On the contrary, methods like Fedavg would let each client run for a couple of epochs.  Could the authors further clarify this, such as how many epochs did the authors use in the evaluation?
> >
> > I would be happy to increase my score once I make sure I understand the work correctly.

---

> > > ### Author Response · Authors · 2023-08-15
> > > **To reviewer stNZ**
> > >
> > > Thank you for your response. We would like to clarify that each client performs multiple local epochs of gradient descents, similar to the FedAvg.
> > >
> > > - In Algorithm 1, Lines 7 and 8 denote the local model update during local epoch $k$, with a total of $K$ ($k \in [0,K-1]$) local epochs. Therefore, Algorithm 1 lets each client run for $K$ epochs.\
> > > As demonstrated in Line 263, the used gradient of DELTA is $\hat{g}^t_i= \sum_{k=0}^{K-1} \nabla F_i(x_{k,t}^i,\xi_{k,t}^i)=\frac{1}{\eta_L}(x_{t,0}^i-x_{t,K}^i)$, which represents the cumulative gradient descent from multiple local updates.
> > > - Similar algorithmic procedures, akin to those presented in Lines 7 and 8, are used in various FL works [1,2,3,4], serving to represent the multiple local updates of FL.
> > > - In our experiments, each algorithm executes 5 local epochs (5 gradient descent steps), as detailed in the experiment setup of Appendix H.2.
> > > - In addition, FedIS (Algorithm 2) also uses gradient $\hat{g}_i^t$, the cumulative gradient descent from multiple local updates, as demonstrated in Line 147.
> > >
> > > We would like to know if our responses have addressed your concerns and look forward to discussing with you. We would greatly appreciate it if you could reconsider the review score.
> > >
> > > [1]Adaptive federated optimization. ICLR 2021.
> > >
> > > [2]Achieving linear speedup with partial worker participation in non-iid federated learning. ICLR 2021.
> > >
> > > [3]Scaffold: Stochastic controlled averaging for federated learning. ICML, 2020.
> > >
> > > [4]Optimal client sampling for federated learning. TMLR, 2022.

---

> ### Author Response · Authors · 2023-08-18
> **Thank you for raising our score.**
>
> Dear Reviewer stNZ,
>
> We sincerely thank you for raising our score from 3 to 5.
>
> Thank you again for the time and effort you have invested in reviewing our paper.

---

### Official Review · Reviewer_1Yft · 2023-07-06

**Soundness:** 2 fair
**Presentation:** 3 good
**Contribution:** 2 fair
**Rating:** 5
**Confidence:** 4

**Summary:**

The paper proposes a novel unbiased client sampling scheme called DELTA for Federated Learning (FL). The authors address the issue of unrepresentative client subsets in FL, which can lead to  significant variance in model updates and slow convergence. They show that existing unbiased sampling methods have limitations in terms of convergence speed and redundancy. In contrast, DELTA selects diverse clients  with significant gradients without clustering operations. The authors also propose practical algorithms for DELTA and Importance Sampling (IS) that rely on accessible client information. The results are validated through experiments  on both synthetic and real-world datasets.

**Strengths:**

1. The paper presents a novel unbiased sampling scheme for FL that addresses the limitations of existing methods. The approach of selecting diverse clients based on gradient diversity without  clustering operations appears to be new and promising.
2. The paper provides theoretical analyses and experimental results to support the proposed sampling schemes. The convergence rates of the practical algorithms are shown to attain the same  order as the theoretical optimal sampling probabilities.
3. The paper is well-organized and clearly presents the motivation, methodology, and results of the study. The figures effectively illustrate the concepts and comparisons between different sampling methods.

**Weaknesses:**

1. In Corollary 4.2, the convergence rate of the practical algorithm depends on a term $\tilde{U}$, which can be very large if $\left\|\nabla F_i\left(x_s\right)-\nabla f\left(x_s\right)\right\|$ is small.

2. Can the proposed client selection framework be applied to federated learning algorithms other than vanilla SGD. I believe a unified theoretical analysis would show the effectiveness of the proposed method.


**Questions:**

See the weakness part above.

**Limitations:**

The paper does not explicitly address the limitations and potential negative societal impacts of the proposed work, but I do not think that this is a major issue.

---

> ### Author Rebuttal · Authors · 2023-08-09
>
> Dear Reviewer 1Yft, we sincerely appreciate your thorough review of our paper. We have diligently incorporated your feedback to enhance the quality of our work. Kindly find our comprehensive responses to your questions outlined below:
>
>
> > In Corollary 4.2, the convergence rate of the practical algorithm depends on a term $\tilde{U}$, which can be very large if $||∇F_i(x_s)−∇f(x_s)||$ is small.
> >
>
> We would like to clarify that $\tilde{U}$ will not be very large. $\tilde{U}$ are determined by $U_1=\frac{||\nabla F_i(x_t)-\nabla f(x_t)||}{||F_i(x_s)-\nabla f(x_s)||}$  and $U_2=\frac{||\sigma_{L,i,t}||}{||\sigma_{L,i,s}||}$, where $t$ represents the current training round and $s$ represent the last participated round of client $i$. Specifically,
>
> - $U_1=E(\frac{||\nabla F_i(x_t)-\nabla f(x_t)||}{||F_i(x_s)-\nabla f(x_s)||})$ represents the gradient diversity change ratio between rounds. As training progresses, gradient diversity typically diminishes, leading to a change ratio smaller than 1.
> - Additionally, Figure 2 in the one-page response PDF illustrates the minimum and maximum norms of gradient diversity per round across all clients. Even in the most adverse scenario where $||\nabla F_i(x_t) - \nabla f(x_t)||$ is chosen as its maximum and $||\nabla F_i(x_s) - \nabla f(x_s)||$ is chosen as its minimum, $U_1$ is constrained. The minimum value across all rounds exceeds 0.5, while the maximum value remains below 8.
> - In addition to $U_1$, the local variance-related term $U_2$ also contributes to the overall value of $\tilde{U}$. The precise expression for $\tilde{U}$ is $E\left|\left|\frac{R_i^t}{R_i^s}\right|\right|$, where $R_i^s = \sqrt{\alpha_1||\nabla F_i(x_s) - \nabla f(x_s)|| + \alpha_2\sigma_{L,s}}$. Therefore, due to the presence of $\sigma_{L,s}$, the denominator remains sufficiently nontrivial, preventing an excessive increase in the value of $\tilde{U}$.
>
> The above discussion regarding $\tilde{U}$ is provided in Remark G.1 and eq (89) of Appendix G, and we refer to the discussion in the paper L283.
>
> > Can the proposed client selection framework be applied to federated learning algorithms other than vanilla SGD. I believe a unified theoretical analysis would show the effectiveness of the proposed method.
> >
>
> We appreciate your suggestion. In addition to the results already presented in Table 2, which include momentum, prox, and VARP, we are willing to provide additional experimental results.
>
> Specifically, we conduct experiments on LEAF (FEMNIST and CelebA) with other non-vanilla SGD algorithms, namely Adagrad and Adam. The results are shown in the below Table:
>
> | Algorithm on FEMNIST | Adagrad  |  | Adam |  |
> | --- | --- | --- | --- | --- |
> |  | Accuracy (%) | Rounds for 80% | Accuracy (%) | Rounds for 80%  |
> | FedAvg | 80.93 ± 0.08 | 893 (1.0$\times$) | 80.04 ± 0.15  |  882  (1.0$\times$) |
> | Cluster-based IS | 80.69±0.12 | 760 (1.17$\times$)  | 79.11± 0.18 | - |
> | FedIS | 80.96 ± 0.03 | 723 (1.24$\times$) | 80.10 ±0.25 | 787 (1.12$\times$) |
> | DELTA | 81.79  ± 0.09 | 612 (1.46$\times$) | 80.92 ±0.07 | 600(1.47$\times$) |
>
> | Algorithm on CelebA | Adagrad  |  | Adam |  |
> | --- | --- | --- | --- | --- |
> |  | Accuracy (%) | Rounds for 80% | Accuracy (%) | Rounds for 80%  |
> | FedAvg | 88.92 ± 0.08 | 329 (1.0$\times$) | 89.04 ± 0.22  |  244  (1.0$\times$) |
> | Cluster-based IS | 89.71±0.10 | 329 (1.0$\times$)  | 89.26± 0.19 | 164 (1.49$\times$) |
> | FedIS | 90.14 ± 0.01 | 243 (1.35$\times$) | 89.92 ±0.05 | 140 (1.74$\times$) |
> | DELTA | 90.38  ± 0.02 | 214 (1.54$\times$) | 90.58 ±0.07 | 109 (2.24) |
>
> Analogous to [1], our analysis seamlessly extends to FedAdagrad and FedAdam. The divergence between these algorithms and SGD originates from the learning rate adaptation, which doesn't alter the convergence analysis steps.
>
> Specifically, the difference in analysis arises with $x_{t+1}=x_t+\eta \frac{\Delta_t}{\sqrt{v_t}+\tau}$ replacing $x_{t+1}=x_t+\eta \Delta_t$, where $v_t=\beta v_{t-1}+\left(1-\beta\right) \Delta_t^2$. As demonstrated in Corollary 1 and Corollary 2 of [1], following the very similar steps of vanilla SGD, Adam's convergence rate order is the same as that of SGD.
>
> [1] Adaptive federated optimization. ICLR 2021.

---

> ### Author Response · Authors · 2023-08-21
> **To reviewer 1Yft**
>
> Dear reviewer 1Yft,
>
> Thank you for your invaluable review that significantly improved our manuscript's quality. In addition, we have taken into account the feedback provided by other reviewers to further refine our paper. For a summary of these revisions, please refer to the "Author Rebuttal by Authors" global comment.
>
> As the discussion period concludes, we would like to ensure that we have adequately addressed all your concerns. Please inform us of any further clarifications or experimental evaluations that could enhance our work.
>
> We deeply appreciate your expertise and dedicated time reviewing our manuscript.
>
> Best regards,
>
> Authors

---

### Official Review · Reviewer_71Y7 · 2023-07-09

**Soundness:** 3 good
**Presentation:** 3 good
**Contribution:** 3 good
**Rating:** 5
**Confidence:** 4

**Summary:**

This paper studies the sampling schemes in federated learning. In particular, the authors first develop a new analysis method for important sampling (IS) strategy, which achieves a better convergence rate and then they propose a new sampling approach called DELTA, which can outperform the IS scheme. In addition, the authors also proposed a practically implementable version of DELTA and the performance is evaluated using experiments.

**Strengths:**

1. The authors develop a new convergence analysis for IS for nonconvex objective functions.

2. The authors propose a new sampling approach called DELTA and propose a practically implementable version of DELTA.

3. The proposed algorithm is evaluated using experiments.

**Weaknesses:**

1. The novelty of the analysis is unclear. It seems a standard "by round" analysis. Since the authors claim the analysis is one of the major contributions of this paper. I suggest the authors to elaborate this point in detail.

2. The Assumption 4 seems strong and can significantly simplify the analysis in general. In addition, it  can be violated easily in practice.

3. The wring of the paper is not careful enough. Many parameters are presented without explanation although it is not very hard to figure them out. For example, in Theorem 3.1, many parameters such as K and T are not explained. In addition, instead of just mentioning some lemmas and assumptions in the appendix or other papers, it will be much better to state them in the main paper to make the reading easier.

**Questions:**

1. The improvement of the proposed analytical approach is unclear. It seems all the approach achieve a $O(1/\epsilon^2)$ of the communication complexity.

2. When IS is analyzed, Assumption 3 is used, whereas DELTA is analyzed, it seems that Assumption 3 is not used while the gradient diversion $\zeta_{G,i,t}$ is used, since $\sigma_G$ is not part of Theorem 3.4. From Corollary 3.7 and compare it to Theorem 3.1, it seems the advantage of DELTA is to eliminate the term of $\sigma_G$. If it is correct, it seems that it is a little bit unfair to use different assumptions to analyze the two different approaches.

3. The novelty of the analysis has to be more elaborated. Please clearly state the main differences between the proposed analysis and other analysis in the literature such as IS and many works related to FedAvg. To me, it is a pretty standard "by round" analysis for nonconvex objective functions.

**Limitations:**

The authors discussed the limitations of the work in the end.

---

> ### Author Rebuttal · Authors · 2023-08-09
>
> Dear reviewer 71Y7 , we would like to thank you for your time spent reviewing our paper and for providing constructive comments. Please kindly find our responses to your raised questions below:
>
> > The novelty of the analysis is unclear. Please clearly state the main differences between the proposed analysis and other analysis in the literature such as IS and many works related to FedAvg.
> >
>
> Thanks for your suggestion, we would like to elaborate on the novelty of our analysis. Please see “Reply to All Reviewers” for details.
>
> > Assumption 4 seems strong and can significantly simplify the analysis in general. In addition, it can be violated easily in practice.
> >
>
> Thank you for your comment. We would like to explain the soundness of Assumption 4:
>
> - This assumption is essential for IS methods [1,2,3]. Furthermore, it finds common usage in the FL community for convergence analysis [4,5,6,7].
> In particular, according to the definition of IS, $E_{q(x)}[h(x)]=E_{p(x)}\left[\frac{q(x)}{p(x)} h(x)\right]$, where $q(x)$ is the given sampling distribution, $p(x)$  is our proposed sampling distribution and $h(x)$  is the value function. With a little work (similar to our Corollary F.1), one can prove the variance is minimized when $p^*(x) \propto q(x)\|h(x)\|$. If $\|h(x)\|$ is not consistently bounded, $p(x)$ is meaningless.
> - We would like to clarify that in our practical algorithm of DELTA, the used assumption is gradient heterogeneity bound: $E|| \nabla F_i(x_t)-\nabla f(x_t)||^2 \leq G^2$ instead of Assumption 4, as used in eq (89) of the convergence analysis of the practical algorithm. This is a looser assumption than Assumption 4.
> - Figure 1 in the one-page response PDF demonstrates the gradient norm of FedIS on MNIST and FashionMNIST datasets, suggesting gradient can be bounded.
>
> > The writing of the paper is not careful enough.
> >
>
> Sorry for any confusion arising from the simplicity of the symbols.
>
> - While we define the terms in Algorithm 1, we will also add the meanings of K and T in Theorem 3.1: T is the total communication round and K is the total local epoch times.
> - Due to the page limit, we present the lemma details in the Appendix. We will include the lemmas in the main paper along with streamlined explanations.
>
> > The improvement of the proposed analytical approach is unclear. It seems all the approach achieve a $\mathcal{O}(\frac{1}{\epsilon^2})$ of the communication complexity.
> >
>
> We would like to clarify the concise communication complexity of these approaches is $\mathcal{O}(\frac{1}{\epsilon^2}+\frac{1}{\epsilon})$ or $\mathcal{O}(\frac{1}{\epsilon^2}+\frac{1}{\epsilon}+\frac{1}{\epsilon^{3/2}})$, as shown in Table 2 of [8] and Table 1 of our paper. The relevance of the additional terms becomes significant when the dominant term $\mathcal{O}(\frac{1}{\epsilon^2})$ is the same for all approaches.
>
> We provide a convergence rate table for comparison, highlighting the improvement in our approach when all these algorithms share a dominant communication complexity of $\mathcal{O}(\frac{1}{\epsilon^2})$ ($\mathcal{O}(\frac{1}{\sqrt{T}})$).
>
> | Algorithm | Convergence upper bound $\min_{t \in[T]} E\left[\|\|\nabla f\left(x_{t}\right)\|\|_{2}^{2}\right] \leq$ | Rate improvement |
> | --- | --- | --- |
> | FedAvg[9] | $\frac{1}{c}\left(\frac{f^{0}-f^{*}}{\sqrt{nKT}}+\frac{\sigma_L^2+3K\sigma_G^2}{2\sqrt{nKT}}+\frac{5(\sigma_L^2+6K\sigma_G^2)^2}{2KT}+\frac{15(\sigma_L^2+6K\sigma_G^2)}{2\sqrt{nKT^3}}\right)$ | -- |
> | FedIS(others)[7] | $\frac{1}{c}\left(\frac{(f^{0}-f^{*})B^2}{\sqrt{n K T}}+\frac{2F \sigma_{L}^{2}+2F(1-n/m)K\sigma_G^2}{2\sqrt{n K T}}+\frac{B^2F}{T}+\frac{F^{2/3}\sigma_G}{T^{2/3}}\right)$ | reduce the coefficient of $\sigma_G$ |
> | FedIS(ours) | $\frac{1}{c}\left(\frac{f^{0}-f^{*}}{\sqrt{n K T}}+\frac{\sigma_{L}^{2}+K \sigma_G^{2}}{2\sqrt{n K T}}+\frac{5(\sigma_L^2+4K\sigma_G^2)}{2T}\right)$ | remove $\mathcal{O}(\frac{1}{T^{2/3}})$ and reduce the coefficient of $\sigma_G$ |
> | DELTA | $\frac{1}{c}\left(\frac{f^{0}-f^{*}}{\sqrt{n K T}}+\frac{\sigma_{L}^{2}}{2\sqrt{n K T}}+\frac{5(\sigma_{L}^{2}+4 K \zeta_{G}^{{2}})}{2K T}\right)$ | further improve the coefficient of $\mathcal{O}(\frac{1}{T^{1/2}})$ |
>
> > When IS is analyzed, Assumption 3 is used, whereas DELTA is analyzed, it seems that Assumption 3 is not used while the gradient diversion $\zeta_G$ is used,it seems that it is a little bit unfair to use different assumptions to analyze the two different approaches.
> >
>
> Thank you for your suggestion. We would like to clarify that the term $\zeta_G$ of DELTA can be easily transformed into a  $\sigma_G$ related term, thus it is fair.
>
> In particular, by taking the expectation on $\zeta_G$, it equates to $E||\nabla F_i(x_t) - \nabla f(x_t)||^2$. As demonstrated in [10], one can derive $E||\nabla F_i(x_t)-\nabla f(x_t)||^2 = E||\nabla F_i(x_t)||^2 -||\nabla f(x_t)||^2 \leq A||\nabla f(x_t)||^2 + \sigma_G^2$. Shifting $A\|\nabla f(x_t)\|^2$ to the left side of the convergence result, $\zeta_G$ can be directly transformed into $\sigma_G$.
>
> [1]Stochastic optimization with importance sampling for regularized loss minimization, ICML 2015
>
> [2]Not all samples are created equal: Deep learning with importance sampling. ICML 2018
>
> [3]Low-Cost Lipschitz-Independent Adaptive Importance Sampling of Stochastic Gradients. ICPR, 2021
>
> [4]Diverse client selection for federated learning: Submodularity and convergence analysis. ICML 2021
>
> [5]On the effectiveness of partial variance reduction in federated learning with heterogeneous data. CVPR. 2023
>
> [6]Sharper convergence guarantees for asynchronous sgd for distributed and federated learning. NIPS, 2022
>
> [7]Optimal client sampling for federated learning. TMLR, 2022.
>
> [8]Scaffold: Stochastic controlled averaging for federated learning. ICML 2020.
>
> [9]Achieving linear speedup with partial worker participation in non-iid federated learning. ICLR 2021.
>
> [10]A unified theory of decentralized sgd with changing topology and local updates. ICML 2020.

---

> ### Author Response · Authors · 2023-08-18
> **To Reviewer 71Y7**
>
> Dear Reviewer 71Y7,
>
> I hope this message finds you well.
>
> We would like to first express our profound gratitude for the time and expertise you've dedicated to the assessment of our submission. We appreciate your insights and have found your comments to be extremely beneficial in refining our work.
>
> Regarding your concerns, we hope that our rebuttal was able to shed light on them. To summarize briefly,  we clearly compare our analysis with existing literature, such as FedAvg and IS, showcasing the advancements in our approach. Meanwhile, we demonstrate that the practical DELTA  does not need Assumption 4 but a looser assumption, whereas the practical FedIS relies on Assumption 4 due to its necessity as a condition for the importance sampling employed in deep learning.
>
> Your constructive feedback has recommended that we refine our expression and pinpoint the limitations of our work. We deeply appreciate your expertise and we have refined our work based on your feedback. Thank you once again for your time and consideration.
>
> Best Regards,
>
> authors

---

> > ### Comment · Reviewer_71Y7 · 2023-08-20
> >
> > I would like to thank the authors for replying my questions. Most of my concerns were resolved. I increased my score to 5.

---

> > > ### Author Response · Authors · 2023-08-20
> > > **Thank you for raising the score**
> > >
> > > Dear reviewer 71Y7,
> > >
> > > Thank you very much for raising our scores. We have incorporated the discussion of the theoretical analysis novelty and the soundness of Assumption 4 in our revised version. We appreciate your time and efforts in reviewing our rebuttal.

---

### Author Rebuttal · Authors · 2023-08-10

We thank all the reviewers for their time and efforts in reviewing our paper.
We are encouraged they found our motivation and idea to be interesting(stNZ, yPdX), novel (1Yft, yPdX, 9Bi1), and promising (1Yft, yPdX).  We have carefully considered their suggestions and incorporated them into our revised manuscript.

For convenience, we have prepared a one-page response PDF, containing two Figures and three Tables of ***new experiments***.

1. Figure 1 and 2 show the norm of gradient and gradient diversity on MNIST and FashionMNIST.
2. Table 1 and 2 are the results of sampling algorithms integrated with the non-vanilla SGD algorithms (Adagrad and Adam) on FEMNIST and CelebA.
3. Table 3 shows the results of sampling algorithms under common thresholds, i.e., 50% accuracy for CIFAR-10 and 80% accuracy for CelebA.

Our ***contributions*** are concluded as follows：

1. We are the first to propose DELTA, the diverse gradient-based unbiased sampling for FL.  Our analysis shows  DELTA surpasses the SOTA FedAvg in convergence rate by eliminating $\mathcal{O}\left(1 / T^{2 / 3}\right)$ term and a $\sigma_G^2$-related term of $\mathcal{O}\left(1 / T^{1 / 2}\right)$.
2. Our nonconvex FL convergence analysis with IS outperforms existing FedIS analysis, employing a more lenient assumption. Notably, we eliminate the $\mathcal{O}\left(1 / T^{2 / 3}\right)$ term from the convergence rate compared to existing unbiased sampling analysis, including FedIS and FedAvg.
3. We present a practical DELTA algorithm to mitigate the reliance on full gradients, along with a theoretical convergence guarantee.
4. Extensive experiments across datasets confirm DELTA's superiority over existing FL unbiased sampling methods and its compatibility with other optimization algorithms.

### ***Elaborate on the novelty and contribution of our analysis below:***

1. Regarding FedIS analysis,
    1. **Existing challenges:** Despite the existence of existing convergence analysis of partial participant FL [1,2,3], including FedIS that builds on this analysis [4, 5], none of them take full advantage of the nature of unbiased sampling, and thus yield an imprecise upper bound on convergence.
    2. **Our solution:**  To tighten the FedIS upper bound, we first derive a tighter convergence upper bound for unbiased sampling FL. By adopting uniform sampling for unbiased probability, we achieve a tighter FedAvg convergence rate. Leveraging this derived bound, we optimize convergence variance using IS.

    *Compare with existing unbiased sampling FL works, including FedAvg and FedIS (others), our analysis on FedIS entails:*

    - **A tighter Local Update Bound Lemma:** We establish Lemma C.3 using Assumption 3, diverging from the stronger assumption $||\nabla F_i(x_t))-\nabla f(x_t)||^2 \leq \sigma_G^2$ (used in [1,2]), and the derived Lemma C.3 achieves a tighter upper bound than other works (Lemma 4 in [1], Lemma 2 in [2]).
    - **A tighter upper bound on aggregated model updates $E||\Delta_t||^2$:** By fully utilizing the nature of unbiased sampling, we convert the bound analysis of $A_2=E||\Delta_t||^2$ equally to a bound analysis of participant variance $V\left(\frac{1}{m p_i^t} \hat{g}_i^t\right)$ and aggregated model update with full user participation (Eq. (33) and Eq. (34)). In contrast, instead of exploring the property the unbiased sampling,  [1] repeats to use Lemma 4 and [2] uses Lemma 2 for bound $A_2$. This inequality transform imposes a loose upper bound for $A_2$,  resulting in a convergence variance term determined by $\eta_L^3$, which reacts to the rate order being $\mathcal{O}(T^{-\frac{ 2}{3}})$.
    - **Relying on a more lenient assumption:** Beyond the aforementioned analytical improvement, our IS analysis obviates the necessity for unusual assumptions in other FedIS analysis such as Mix Participation [3] and $\rho$-Assumption [4].
2. Regarding DELTA analysis,
    1. **Existing challenge:** IS focuses on minimizing $V\left(\frac{1}{m p_i^t} \hat{g}_i^t\right)$ in convergence variance $\Phi$ (Eq. (4)), while leaving other terms like $\sigma_L$ and $\sigma_G$ unreduced.
    2. **Our solution:** Unlike IS roles to reduce the update gap [5], we propose analyzing the surrogate objective for additional variance reduction.

    *Compared with FedIS, our analysis of DELTA entails:*

    - **Focusing on surrogate objective, introducing a novel Lemma and bound:**
        1. We decompose global objective convergence into surrogate objective and update gap (Eq. (6)). For surrogate objective analysis, we novelly introduce Lemma E.8 to bound local updates.
        2. Leveraging the unique surrogate objective expression and Lemma E.8, we link sampling probability with local variance and gradient diversity, deriving novel upper bounds for $A_1$ and $A_2$ (Eq. (57), Eq. (59)).
        3. By connecting update gap's convergence behavior to surrogate objective through Definition E.1 and Lemma C.2, along with (Eq. 6), we establish $\tilde{\Phi}$ (Eq. (10),(11))) as the new global objective convergence variance.
    - **Optimizing convergence variance through novel $\tilde{\Phi}$:**
    FedIS aims to reduce the update variance term $V(\frac{1}{(mp_i^t)}\hat{g}_i^t)$ in $\Phi$, while FedDELTA aims to minimize the entire convergence variance $\tilde{\Phi}$, which is composed of both gradient diversity and local variance. By minimizing $\tilde{\Phi}$, we get the sampling method DELTA, which further reduces the variance terms of $\Phi$ that cannot be minimized through IS.

[1]Adaptive federated optimization. ICLR 2021.

[2]Achieving linear speedup with partial worker participation in non-iid federated learning. ICLR 2021.

[3]Scaffold: Stochastic controlled averaging for federated learning. ICML, 2020.

[4]Tackling system and statistical heterogeneity for federated learning with adaptive client sampling. IEEE INFOCOM 2022.

[5]Optimal client sampling for federated learning. TMLR, 2022.

---

### Decision · Program_Chairs · 2023-09-21

**Decision:**

Accept (poster)

**Comment:**

The paper introduces a new unbiased client sampling technique for Federated Learning (FL) called DELTA. Motivated by the need to consider both variance and gradient diversity, the authors propose this method as a means to achieve improved convergence rates for non-convex objective functions in FL. To support their proposal, they offer theoretical proofs and evidence from numerical experiments. The paper aims to show that by considering gradient diversity in client sampling, a significant impact can be made on federated learning, addressing a crucial challenge in client sampling.

The authors have adequately addressed concerns from reviews, resulting in reviewers increasing their scores. At this point, I believe that any remaining minor issues can be addressed in the final camera ready version of the paper.